# Learning Locally, Revising Globally:
# Global Reviser for Federated Learning with Noisy Labels

**Yuxin Tian** [1 2]  **Mouxing Yang** [1 2]  **Yuhao Zhou** [1 2]  **Jian Wang** [1 2]  **Qing Ye** [1 2]  **Tongliang Liu** [3]  **Gang Niu** [4]
**Jiancheng Lv** [1 2]

## Abstract

Conventional federated learning (FL) heavily depends on high-quality labels, which are often impractical in the real world, leading to the federated label-noise (F-LN) problem. Worse still, the F-LN problem is exacerbated by the heterogeneity of FL, whereas clients experience different label-noise types, ratios, and data distribution. In this study, we first observe an intriguing phenomenon that the global model of FL exhibits a slow memorization of noisy labels, suggesting its ability to maintain reliable predictions and robust representations in FL. Motivated by this, we propose a novel method termed Federated Global Reviser (FedGR), a straightforward yet effective method comprising three modules that collaboratively rectify noisy labels and regularize local training. By exploiting this inherent property, FedGR improves the label-noise robustness of FL in a self-contained manner. Extensive experiments on three widely used F-LN benchmarks demonstrate the superior performance of FedGR, consistently outperforming eight state-of-the-art baselines even in severe label-noise and data heterogeneity. Code: `https://github.com/cs-yuxintian/FedGR-ICML26`.

## 1. Introduction

Federated learning (FL) facilitates privacy-preserving collaborative training across clients (Zhou et al., 2024; 2025a;b) for applications like healthcare (Kaissis et al., 2020; Huang et al., 2026), recommendation systems (Sun et al., 2024),

and graph learning (Fu et al., 2025a; Huang et al., 2025; Fu et al., 2025b). Despite promising performance (McMahan et al., 2017; Li et al., 2020b; Meng et al., 2024), FL heavily relies on high-quality annotated data. However, precisely annotating decentralized datasets is impractical (Irvin et al., 2019), inevitably leading to the federated label noise (F-LN) problem (Yang et al., 2022b; Xu et al., 2022). Unlike centralized label-noise (C-LN) problem (Han et al., 2018; Li et al., 2020a), F-LN is more challenging due to the label-noise and data heterogeneity (Li et al., 2026b;a), encompassing diverse noise patterns (*e.g.*, varying ratios/types) and data heterogeneity causing class imbalance (Li et al., 2022; Qi et al., 2023; Wu et al., 2023; Qi et al., 2025). This heterogeneity significantly hinders the direct application of centralized learning with noisy labels (C-LNL) methods (Li et al., 2022; Wei et al., 2021). Thus, it is highly desirable to develop a federated learning with noisy labels (F-LNL) approach to tackle the F-LN problem.

Existing F-LNL approaches typically treat the F-LN problem as a distributed extension of learning with noisy labels, focusing on refining client-side training algorithms (Jiang et al., 2022; Wang et al., 2022; Xu et al., 2022; Ji et al., 2024; Kim et al.) or detecting and isolating noisy clients (Xu et al., 2022; Lu et al., 2024) to mitigate negative impacts. However, the dual heterogeneity inherent in F-LNL often renders these methods ineffective. While recent studies (Yang et al., 2022b; Kim et al., 2022; Wu et al., 2023; Tam et al., 2023; Li et al., 2024) have attempted to address this challenge by constructing consensus among clients, such mechanisms frequently involve local data statistics, thereby posing significant risks of privacy leakage. In contrast to these methods, this work tackles the F-LN problem from a novel perspective. Specifically, as illustrated in Figure 1, we anatomize the memorization phenomenon of FL and observe that the global model exhibits significantly slower overfitting to label noise compared to centralized training—a phenomenon we term the intrinsic label-noise robustness of FL. Harnessing this previously not well-explored characteristic enables us to enhance the label-noise robustness of FL in a self-contained and privacy-preserving manner.

In other words, we propose a novel method termed Feder-

[1]College of Computer Science, Sichuan University, Chengdu, China [2]Engineering Research Center of Machine Learning and Industry Intelligence, Ministry of Education, China [3]University of Sydney, Sydney, Australia [4]Southeast University, Nanjing, China. Correspondence to: Jiancheng Lv <lvjiancheng@scu.edu.cn>.

*Proceedings of the 43rd International Conference on Machine Learning*, Seoul, South Korea. PMLR 306, 2026. Copyright 2026 by the author(s).

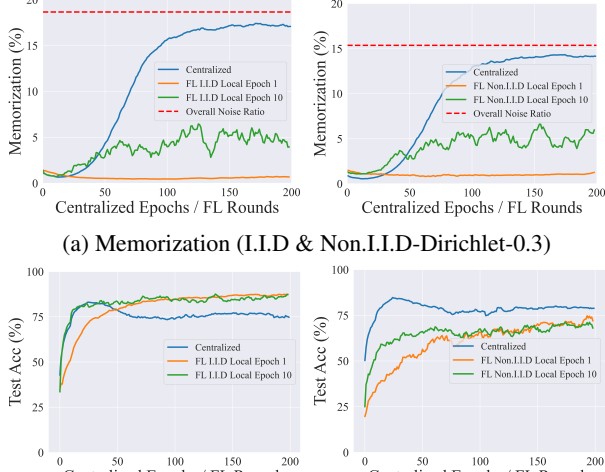

(a) Memorization (I.I.D & Non.I.I.D-Dirichlet-0.3)

(b) Test Acc (I.I.D & Non.I.I.D-Dirichlet-0.3)

*Figure 1.* **(a) Slower Memorization Effect**: On CIFAR-10, the global FL model memorizes $\leq 30\%$ of noisy labels, while significantly lower than that of centralized training. **(b) Preservation of Test Performance**: The global model in FL avoids the test performance degradation typically observed in centralized training under noisy labels. Please see `appendix A.3` for more results and discussions, which indicates that such a phenomenon is non-trivial.

ated Global Reviser (FedGR) to mitigate the adverse effect of the F-LN problem. To be specific, FedGR comprises three modules and takes advantage of the robust global model in two aspects: noisy label correction and local model regularization. First, FedGR introduces a sieving-and-refining module to partition clean and noisy samples for each client and subsequently rectify noisy labels. To address the quality and quantity issues of refined labels under the dual-heterogeneity of the F-LN problem, FedGR introduces a globally revised exponential moving average (EMA) distillation module and a global representation regularization module to further regularize the local training. To sum up, the contributions of this study are outlined as follows:

- This study provides an insightful observation that the global model of FL has a slower tendency to overfit noisy labels, which we refer to as the intrinsic label-noise robustness of FL. To the best of our knowledge, this phenomenon has not been thoroughly explored in previous works, and it motivates us to enhance the label-noise robustness of FL in a self-contained and privacy-preserving manner.

- Motivated by this insight, we introduce FedGR, a novel method designed to enhance robustness in a self-contained and privacy-preserving manner. FedGR employs three modules: sieving-and-refining for sample selection, globally revised EMA distillation for robust pseudo labeling and regularization, and global

representation regularization to further prevent label-noise overfitting. Additionally, we provide a theoretical analysis to guarantee the convergence (see `appendix A.7`).

- Comprehensive experiments on three public F-LN benchmarks, under diverse noise levels and distribution settings, show that FedGR consistently surpasses seven state-of-the-art baselines, delivering substantial gains in both accuracy and robustness.

## 2. Related Work

**Centralized Learning with Noisy Labels.** To address the C-LN problem, most existing C-LNL studies leverage the *memorization effect* (Arpit et al., 2017) to design robust training strategies for sample selection (Han et al., 2018; Yu et al., 2019; Yang et al., 2022a; 2024) and noisy label correction (Berthelot et al., 2019; Li et al., 2020a; Xiao et al., 2023; Zhang et al., 2024). However, due to the following two reasons, it is undesirable for FL to directly adopt these C-LNL methods to tackle the F-LN problem. On the one hand, the data heterogeneity (Li et al., 2022; Huang et al., 2022) of FL makes the class-balanced assumption used by almost all the C-LNL approaches unattainable (Li et al., 2020a; Wei et al., 2021). On the other hand, these methods induce sophisticated learning algorithms, such as two peer networks (Han et al., 2018; Yu et al., 2019; Li et al., 2020a), which involve high computation and communication overhead for FL. In contrast, FedGR performs sample sieving on the server instead of each client, which mitigates the adverse impacts of the dual heterogeneity and introduces moderate computation and communication overhead.[1]

**Federated Learning with Noisy Labels.** By treating the F-LN problem as a distributed extension of learning with noisy labels, existing F-LNL methods often adopt client-side independent sample selection (Wang et al., 2022; Xu et al., 2022; Ji et al., 2024; Jiang et al., 2024; Jiang & Zhang, 2025) or noisy client detection (Xu et al., 2022; Lu et al., 2024; Jiang et al., 2024; Morafah et al.) to alleviate the negative effects of noisy labels. Additionally, some works even straightforwardly employ regularization techniques to regularize the local training of client (Jiang et al., 2022; Kim et al.; Zhou & Wang, 2024). However, these methods often struggle to handle the dual heterogeneity of the F-LN problem effectively. Thus, most recent studies (Yang et al., 2022b; Kim et al., 2022; Wu et al., 2023; Tam et al., 2023; Li et al., 2024) attempt to construct consensus among clients to address the dual heterogeneity of the F-LN problem. Even so, these F-LNL approaches exhibit several limitations. To be specific, both client-side independent sample selection and noisy client detection, which rely on the memoriza-

---

[1]Please see `appendix A.8` for analysis.

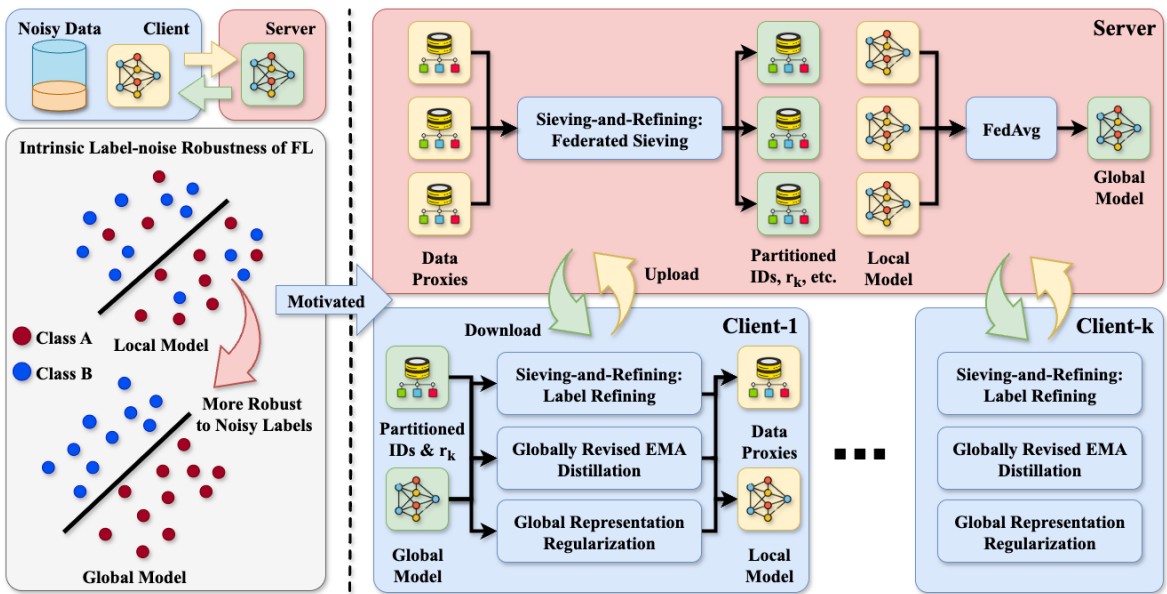

*Figure 2.* FedGR operates in three modules (see `appendix` A.5 for pseudo codes). Initially, the Sieving-and-Refining module employs Federated Sieving (FS) to model global noise patterns via client-side data proxies, followed by Label Refining (LR) to correct noisy labels using the robust global model. Second, to tackle the quality and quantity issues of refined labels under the dual-heterogeneity, Globally Revised EMA Distillation is introduced to distills knowledge from a local EMA model revised by the global parameters. Lastly, Global Representation Regularization is proposed to further prevent local overfitting to cumulative EMA errors by enforcing global-to-local representation consistency.

tion effect (Arpit et al., 2017), prove unreliable for clients presenting dual heterogeneity. Moreover, techniques that construct consensus based on local data statistics pose potential privacy risks, as they necessitate the transmission of sensitive information (Yang et al., 2022b; Kim et al., 2022; Tam et al., 2023). In contrast, we exploit the intrinsic label-noise robustness of the global model to promote the performance of FL faced with noisy labels, which is irrelevant to data distribution and thus privacy-preserving. And the effectiveness of the proposed FedGR is substantiated through comprehensive experimental evaluations.

## 3. Method

### 3.1. Problem Definition

A typical FL system (McMahan et al., 2017; Xu et al., 2022) maintains a server for model parameter aggregation and $K$ clients that train their local models on their local low-quality datasets $\hat{\mathcal{D}}_k = \{(\mathbf{x}_i, \hat{y}_i)\}_{i=1}^{n_k}$, where $\hat{y}_i$ and $n_k$ represent the one-hot vector of the label and the size of the local dataset on client $k$, respectively. The local objective of client $k$ with a loss function $\ell(\cdot, \cdot)$ on $\hat{\mathcal{D}}_k$ at the $t$-th round can be:

$$\mathcal{L}_k = \mathbb{E}_{\hat{\mathcal{D}}_k} \left[ \ell(\mathbf{p}_i^l, \hat{y}_i) \right] \text{ s.t. } \mathbf{p}_i^l = (h \circ f)(\mathbf{x}_i; \mathbf{w}_k^t), k \in \mathcal{S}(t), \tag{1}$$

where $\mathbf{p}_i^l$ is the logits output by the head $h(\cdot; \mathbf{w}_{k,h}^t)$ and backbone $f(\cdot; \mathbf{w}_{k,f}^t)$ with the local model parameters $\mathbf{w}_k^t = \{\mathbf{w}_{k,h}^t, \mathbf{w}_{k,f}^t\}$. The global model parameters $\mathbf{w}_g^t$ at commu-

nication round[2] $t$ are computed as the importance-weighted average of the aggregated local model parameters:

$$\mathbf{w}_g^t = \sum_{k \in \mathcal{S}(t)} a_k \mathbf{w}_k^t \quad \text{s.t.} \quad \sum_{k \in \mathcal{S}(t)} a_k = 1, \tag{2}$$

where $\mathcal{S}(t)$ is the set of selected clients at round $t$ and $a_k = n_k / \sum_{i \in \mathcal{S}(t)} n_i$ is the corresponding importance weight. Finally, the global objective $\mathcal{L}$ of F-LNL can be formulated as:

$$\min_{\mathbf{w}_g} \mathcal{L}(\mathbf{w}_g) = \sum_{k \in \mathcal{S}} a_k \mathcal{L}_k(\mathbf{w}_k) \quad \text{s.t.} \quad \sum_{k \in \mathcal{S}} a_k = 1, \tag{3}$$

where $\mathcal{S}$ is the set of all clients and $\|\mathcal{S}\| = K$.

### 3.2. Overview of FedGR

As shown in Figure 1, the global model of FL exhibits a slower propensity to overfit noisy labels, indicating its ability to maintain reliable predictions and robust representations during training. Building upon this observation, we propose FedGR, which incorporates three specialized modules for local training in FL to enhance the label-noise robustness, as shown in Figure 2. In brief, FedGR first leverages the label-noise-robust characteristic of the global model to sieve and refine the noisy labels of each client with the sieving-and-refining module. It then regularizes local

---

[2]Communication round and round are used interchangeably.

model training through globally revised EMA distillation module and global representation regularization module, with the help of the global model. By combining the objectives of these three modules, the local learning objective of the FedGR can be:

$$\mathcal{L}_k = \mathcal{L}_k^{SR} + \lambda_{\mathcal{B}}\mathcal{B}_k + \lambda_{\mathcal{R}}\mathcal{R}_k, \tag{4}$$

where $\mathcal{L}_k^{SR}$, $\mathcal{B}_k$, and $\mathcal{R}_k$ correspond to the sieving-and-refining objective, globally revised EMA distillation, and global representation regularization, respectively. The hyperparameters $\lambda_{\mathcal{B}}$ and $\lambda_{\mathcal{R}}$ control the relative importance of each term. Theoretical convergence analysis is presented in `appendix` A.7. In the following, we elaborate on each module.

### 3.3. Sieving-and-Refining

Briefly, the sieving-and-refining module consists of two components: Federated Sieving (FS) and Label Refining (LR). In FS, the server employs a Gaussian Mixture Model (GMM) to model the label-noise patterns using aggregated instance-level data proxies (*e.g.,* loss). Based on this modeling, FS partitions each client's data proxies into clean and noisy subsets and estimates the client's label-noise ratio $r_k$. Then, these partitioning results are transmitted back to the clients, where LR is adopted to refine the noisy samples identified by FS, guided by the estimated $r_k$. In the following, we will introduce the objective of the sieving-and-refining module and then elaborate on the FS and LR.

According to the memorization effect (Arpit et al., 2017), the global model will undergo an $\alpha$-round warm-up phase before LR is activated. Thus, the local learning objective of sieving-and-refining could be divided into two phases. For the first $\alpha$ rounds, the local objective of client $k$ is to perform vanilla supervised learning on its local dataset $\hat{\mathcal{D}}_k$, *i.e.,*

$$\mathcal{L}_k^{SR} = \mathbb{E}_{\hat{\mathcal{D}}_k} \left[ \mathcal{H}\left(\mathbf{p}_i^l, \hat{y}_i\right) \right], \text{ if } t < \alpha, \tag{5}$$

where $\mathbf{p}_i^l$ and $\mathcal{H}(\cdot)$ are the output logits and cross entropy loss. To resist the overfitting to noisy labels (Nishi et al., 2021), we adopt strong data augmentation on the input to get the logits, *i.e.,*

$$\mathbf{p}_i^l \rightarrow \mathbf{p}_i^{l,s} = (h \circ f)\left(\mathbf{x}_i^s; \mathbf{w}_k^t\right). \tag{6}$$

Next, after $\alpha$ rounds, client $k$ would adopt LR to refine the noisy labels detected by FS and the refined dataset $\tilde{\mathcal{D}}_k$ will be subsequently used for the local training, *i.e.,*

$$\mathcal{L}_k^{SR} = \mathbb{E}_{\tilde{\mathcal{D}}_k} \left[ \mathcal{H}\left(\mathbf{p}_i^{l,s}, \tilde{y}_i\right) \right], \text{ if } t \geq \alpha. \tag{7}$$

To sum up, the objective of sieving-and-refining module is

$$\mathcal{L}_k^{SR} = \begin{cases} \mathbb{E}_{\hat{\mathcal{D}}_k}\left[\mathcal{H}\left(\mathbf{p}_i^{l,s}, \hat{y}_i\right)\right], & t < \alpha \\ \mathbb{E}_{\tilde{\mathcal{D}}_k}\left[\mathcal{H}\left(\mathbf{p}_i^{l,s}, \tilde{y}_i\right)\right], & t \geq \alpha \end{cases}. \tag{8}$$

**Federated Sieving.** The FS comprises two steps: client-side instance-level data proxy computation and server-side noisy sample partitioning. To be specific, for the first step, client $k \in \mathcal{S}(t)$ would adopt the label-noise robust global model $\mathbf{w}_g^{t-1}$ to compute a noise-distinguishable data proxy for each sample $\mathbf{x}_i \in \hat{\mathcal{D}}_k$ at the beginning of local training in each round. In order to preserve privacy, we define the data proxy of sample $\mathbf{x}_i \in \hat{\mathcal{D}}_k$ as its mean inference loss in the previous $t$ rounds. To compute it, client $k$ first maintains a loss observation set for each sample, and the loss observation set $L_i^t = \{\ell_{i,p}\}_{p=1}^{T_k}$ of sample $\mathbf{x}_i$ at round $t$ is updated as follows:

$$\ell_{i,T_k} = \mathcal{H}\left(\mathbf{p}_i^g, \hat{y}_i\right) \text{ s.t. } \mathbf{p}_i^g = (h \circ f)\left(\mathbf{x}_i; \mathbf{w}_g^{t-1}\right), \tag{9}$$

where $T_k$ represents the number of times client $k$ has been selected in the previous $t$ rounds. Subsequently, the mean inference loss of sample $\mathbf{x}_i$ in the previous $t$ rounds can be obtained as follows:

$$\bar{\ell}_i^t = \frac{1}{T_k} \sum_{p=1}^{T_k} \ell_{i,p}. \tag{10}$$

Then, client $k$ uploads its local parameters $\mathbf{w}_k^t$ and instance-level data proxies $\{(d_{i,k}, \bar{\ell}_i^t)\}_{i=1}^{n_k}$ to the server at the end of local training, where $d_{i,k}$ denotes a globally unique identifier for sample $\mathbf{x}_i$ on client $k$. In the second step of FS, the server would aggregate these data proxies from all selected clients $\mathcal{S}(t)$ and model their distribution using a two-component GMM. By setting a partitioning threshold for the posterior probability $q_{i,k}$ (Li et al., 2020a) of sample $\mathbf{x}_i \in \hat{\mathcal{D}}_k$ belonging to the "clean" GMM component, the server can partition the identifiers into clean/noisy subsets. Subsequently, the client's noise ratio $r_k$ can also be derived from the above partition. Finally, the partitioning results of client $k$ and the aggregated global model parameters will be returned to it when it is selected for collaborative training in subsequent rounds. To obtain the partitioning results of all clients, FS follows (Xu et al., 2022) to adopt random sampling without replacement in the first $\alpha$ rounds (Xu et al., 2022), deviating from the standard FL setup. After that, the standard FL client sampling (McMahan et al., 2017) is adopted.

The benefits of FS can be twofold. On the one hand, the proposed FS is a more reliable sample sieving under the dual heterogeneity of the F-LN problem, as it leverages larger-scale training dynamics (*e.g.,* loss) than any single client can provide for label-noise modeling. On the other hand, FS is privacy-preserving, as it only transmits information irrelevant to the data distribution.[3]

**Label Refining.** At the beginning of local training on client $k \in \mathcal{S}(t)$ during the $t$-*th* round, the client can divide its

---

[3]Loss is irrelevant to the distribution of $(\mathbf{x}, \mathbf{y})$, thus it is privacy-preserving. See `appendix` A.6 for discussion.

local dataset $\hat{\mathcal{D}}_k$ into a clean subset $\hat{\mathcal{D}}_k^c$ and a noisy subset $\hat{\mathcal{D}}_k^n$, based on the returned partition results. Then, client $k$ can employ LR to generate pseudo labels that correct the labels of the noisy subset $\hat{\mathcal{D}}_k^n$. Notably, due to the dual heterogeneity of the F-LN problem, we propose a label refinement strategy that is conditioned on the estimated $r_k$ to obtain reliable refined labels. Specifically, the refined label $\tilde{y}_i$ of sample $\mathbf{x}_i \in \hat{\mathcal{D}}_k$ is defined as follows:

$$\tilde{y}_i = \begin{cases} \hat{y}_i, & r_k < \beta \text{ and } \mathbf{x}_i \in \hat{\mathcal{D}}_k^c \\ q_{i,k}\hat{y}_i + (1-q_{i,k})y_i^{pse}, & r_k < \beta \text{ and } \mathbf{x}_i \in \hat{\mathcal{D}}_k^n \\ y_i^{pse}, & r_k \geq \beta \end{cases}$$
(11)

where $y_i^{pse}$ is the pseudo label and $\beta$ is the label-noise ratio threshold. According to the observation on the global model, we adopt FixMatch (Sohn et al., 2020) on the global model to generate the reliable pseudo label $y_i^{pse}$ for sample $\mathbf{x}_i$.

The above strategy is predicated on the following two considerations. Firstly, for clients exhibiting simple label-noise patterns, the partitioning results are deemed relatively reliable. Therefore, we refine the noisy label set by leveraging the clean probability $q_i$ derived from the GMM, following the methodology outlined in (Li et al., 2020a). Secondly, for the clients suffering from high label-noise ratios (*e.g.,* $r_k \geq \beta$) or complex label-noise types (*e.g.,* asymmetric), we consider all their provided labels to be untrustworthy and only use the pseudo labels as the refined labels.

### 3.4. Globally Revised EMA Distillation

Although the global model is relatively robust to noisy labels, it struggles to fit the local data distribution of each client due to dual heterogeneity. Thus, it often fails to produce a sufficient number of reliable pseudo labels for each client. While the local model can fit clients' distributions, it is easily corrupted by noisy labels, especially on high-noise clients, resulting in unreliable predictions. To address this conflict between the global and local models under dual heterogeneity, we propose a globally revised EMA distillation module. This module resorts to two types of models, *i.e.,* the global model and the local EMA model, to regularize local learning via knowledge distillation.

To be specific, as EMA inherently has stability and early-training robustness in learning with noisy labels (Zhou et al., 2021; Morales-Brotons et al., 2024), each client will maintain a local EMA model $\mathbf{w}_{k,ema}^t$ during the local training. To mitigate the cumulative effect of noisy labels on the local EMA model, we propose revising the local EMA model with the global model before distillation, as the global model is more robust to noisy labels. Formally, for the $t$-th round, the revising and usual updating steps for the local EMA model $\mathbf{w}_{k,ema}^t$ on client $k$ at the $m_k$-th local training step

can be:

$$\mathbf{w}_{k,ema}^{t,m_k} = \begin{cases} \gamma_g \mathbf{w}_{k,ema}^{t-1,m_k} + (1-\gamma_g)\mathbf{w}_g^{t-1}, & m_k = 0 \\ \gamma_l \mathbf{w}_{k,ema}^{t,m_k-1} + (1-\gamma_l)\mathbf{w}_k^{t,m_k}, & m_k \geq 1 \end{cases}$$
(12)

where $m_k$ and $\gamma_{g/l}$ denote the local training step and the momentum decay for the global revised EMA step/local EMA step, respectively. Then, the proposed globally revised EMA distillation module would adopt the revised local EMA model $(h \circ f)\left(\cdot; \mathbf{w}_{k,ema}^{t,0}\right)$ as teacher to distill the knowledge to the local model $(h \circ f)(\cdot; \mathbf{w}_k^t)$ and the objective of it on client $k$ can be formulated as:

$$\mathcal{B}_k = \mathbb{E}_{\hat{\mathcal{D}}_k}\left[KL\left(\frac{\mathbf{p}_i^{le,w}}{\tau}, \frac{\mathbf{p}_i^{l,s}}{\tau}\right)\right],$$
(13)

where $\mathbf{p}_i^{le/l,w}$, $KL$, and $\tau$ denote the output logits of revised local EMA model/local model on weakly augmented data, the Kullback-Leibler divergence loss, and the temperature, respectively. Notably, the logits predicted by the revised local EMA model are

$$\mathbf{p}_i^{le,w} = (h \circ f)\left(\mathbf{x}_i^w; \mathbf{w}_{k,ema}^{t,0}\right),$$
(14)

where $\mathbf{x}_i^w$ refers to the weakly augmented $\mathbf{x}_i \in \hat{\mathcal{D}}_k$. The benefit of distilling logits from the revised EMA model instead of the online EMA model $(h \circ f)\left(\cdot; \mathbf{w}_{k,ema}^{t,m_k}\right)$ could be twofold. On the one hand, it lowers the forward computation cost, as the teacher's logits are computed only once at the beginning of local training. On the other hand, it improves the resilience of logits to the accumulated adverse effect of noisy labels and incorrect refined labels. The ablation in Table 4 also demonstrates that such a mechanism is more effective.

Similar to the label refinement strategy, $\gamma_g$ of each client should be conditioned on the $r_k$ and the results of sieving-and-refining module due to the dual heterogeneity of the F-LN problem. For instance, after the warm-up phase, if client $k \in \mathcal{S}(t)$ suffers from a high label-noise ratio ($r_k \geq \beta$) and fails to obtain a sufficient number of samples, the local EMA model is deemed unreliable and should be entirely replaced by the global model, *i.e.,* $\gamma_g = 0$. Otherwise, the local EMA model will be revised by the global model with momentum decay $\gamma_g$. Formally, $\gamma_g$ could be adjusted as follows:

$$\gamma_g = \begin{cases} \gamma_g, & t \geq \alpha \\ 0, & \left(r_k \geq \beta \text{ and } \dfrac{\left\|\tilde{\mathcal{D}}_k^r\right\|}{\left\|\hat{\mathcal{D}}_k\right\|} < \mu\right) \text{ or } t < \alpha \end{cases},$$
(15)

*Table 1.* Results on CIFAR-10. The 1st/2nd-best results are in a gray box w/. and w/o. boldface.

| Data Partition | I.I.D | | | | | | | | Non.I.I.D-Dirichlet (0.3) | | | | | | | |
|---|---|---|---|---|---|---|---|---|---|---|---|---|---|---|---|---|
| Noise Type | Clean | Sym | | Asym | | Mixed | | Avg | Clean | Sym | | Asym | | Mixed | | Avg |
| $\phi$ | 0.0 | 0.6 | 1.0 | 0.6 | 1.0 | 0.6 | 1.0 | | 0.0 | 0.6 | 1.0 | 0.6 | 1.0 | 0.6 | 1.0 | |
| $\mathcal{U}(\rho_{min}, \rho_{max})$ | 0.0-0.0 | 0.5-1.0 | 0.5-1.0 | 0.2-0.4 | 0.2-0.4 | 0.2-0.4 | 0.2-0.4 | | 0.0-0.0 | 0.5-1.0 | 0.5-1.0 | 0.2-0.4 | 0.2-0.4 | 0.2-0.4 | 0.2-0.4 | |
| FedAvg | 92.55 ±0.14 | 61.60 ±3.73 | 23.89 ±3.62 | 83.46 ±0.89 | 70.44 ±1.80 | 81.90 ±0.73 | 70.66 ±0.69 | 65.33 ±1.91 | **87.05** ±1.45 | 39.46 ±3.02 | 17.32 ±1.07 | 69.98 ±1.58 | 53.74 ±1.61 | 66.10 ±2.09 | 51.92 ±1.70 | 49.75 ±1.84 |
| FedProx | 90.75 ±0.16 | 56.57 ±4.27 | 23.02 ±2.90 | 79.78 ±1.00 | 64.44 ±1.64 | 77.94 ±0.61 | 65.23 ±0.83 | 61.16 ±1.88 | 86.86 ±0.58 | 36.94 ±2.42 | 16.69 ±1.32 | 69.19 ±0.98 | 52.68 ±0.60 | 64.08 ±1.50 | 49.77 ±1.00 | 48.22 ±1.30 |
| FL-Coteaching | - | 66.43 ±2.65 | 47.28 ±5.03 | 87.40 ±0.58 | 82.61 ±0.75 | 86.51 ±0.43 | 83.99 ±0.54 | 75.70 ±1.66 | - | 44.42 ±2.11 | 33.49 ±1.00 | 76.20 ±1.99 | 72.93 ±1.93 | 74.40 ±1.67 | 72.42 ±1.56 | 62.31 ±1.71 |
| FL-DivideMix | - | 76.54 ±0.42 | 68.47 ±3.00 | 85.46 ±0.21 | 86.23 ±0.36 | 84.76 ±0.31 | 85.19 ±0.48 | 81.11 ±0.80 | - | 58.94 ±1.19 | 38.35 ±4.45 | 73.13 ±1.71 | 71.45 ±1.47 | 70.18 ±1.37 | 68.86 ±1.29 | 63.49 ±1.91 |
| FedCorr [CVPR22] | 92.55 ±0.71 | 92.04 ±0.17 | 55.12 ±1.55 | 83.76 ±0.74 | 83.06 ±1.36 | 84.15 ±0.52 | 84.00 ±0.17 | 80.36 ±0.75 | 77.14 ±8.44 | 78.85 ±4.74 | 29.42 ±2.43 | 57.91 ±8.15 | 55.67 ±6.81 | 83.33 ±1.43 | 67.85 ±3.75 | 62.17 ±4.55 |
| FedNoRo [IJCAI23] | - | 63.30 ±0.93 | 33.98 ±4.71 | 71.83 ±0.33 | 63.29 ±1.12 | 71.07 ±0.32 | 63.24 ±0.30 | 61.12 ±1.29 | - | 51.64 ±2.07 | 18.60 ±1.12 | 58.99 ±2.07 | 41.18 ±2.92 | 57.09 ±1.57 | 43.99 ±1.30 | 45.25 ±1.84 |
| FedDiv [AAAI24] | - | 90.36 ±0.57 | 33.14 ±21.83 | **93.67** ±0.04 | **92.86** ±0.19 | **93.43** ±0.07 | **92.36** ±0.09 | 82.64 ±3.80 | - | 20.76 ±7.76 | 14.22 ±3.65 | 26.85 ±2.49 | 26.28 ±11.31 | 35.71 ±6.23 | 23.20 ±6.28 | 24.50 ±6.29 |
| FedFixer [AAAI24] | - | 66.49 ±1.03 | 30.18 ±2.92 | 83.29 ±0.28 | 70.65 ±1.11 | 85.52 ±0.95 | 77.50 ±1.15 | 68.94 ±1.24 | - | 52.37 ±3.43 | 22.53 ±2.54 | 72.24 ±2.18 | 57.30 ±1.20 | 74.83 ±3.02 | 63.72 ±2.30 | 57.17 ±2.45 |
| FedGR [Ours] | **93.95** ±0.10 | **92.09** ±0.25 | **83.91** ±1.32 | 93.38 ±0.30 | 91.64 ±0.38 | 93.13 ±0.32 | 92.27 ±0.18 | **91.07** ±0.46 | 86.22 ±1.97 | **82.04** ±2.23 | **63.64** ±5.39 | **86.79** ±2.68 | **83.67** ±5.02 | **86.50** ±2.36 | **84.65** ±2.38 | **81.22** ±3.43 |

*Table 2.* Results on CIFAR-100. The 1st/2nd-best results are in a gray box w/. and w/o. boldface.

| Data Partition | I.I.D | | | | | | | | Non.I.I.D-Dirichlet (0.3) | | | | | | | |
|---|---|---|---|---|---|---|---|---|---|---|---|---|---|---|---|---|
| Noise Type | Clean | Sym | | Asym | | Mixed | | Avg | Clean | Sym | | Asym | | Mixed | | Avg |
| $\phi$ | 0.0 | 0.6 | 1.0 | 0.6 | 1.0 | 0.6 | 1.0 | | 0.0 | 0.6 | 1.0 | 0.6 | 1.0 | 0.6 | 1.0 | |
| $\mathcal{U}(\rho_{min}, \rho_{max})$ | 0.0-0.0 | 0.5-1.0 | 0.5-1.0 | 0.2-0.4 | 0.2-0.4 | 0.2-0.4 | 0.2-0.4 | | 0.0-0.0 | 0.5-1.0 | 0.5-1.0 | 0.2-0.4 | 0.2-0.4 | 0.2-0.4 | 0.2-0.4 | |
| FedAvg | 64.85 ±0.33 | 33.97 ±1.82 | 13.00 ±1.85 | 54.51 ±0.83 | 44.18 ±1.72 | 52.37 ±0.40 | 43.02 ±0.44 | 40.18 ±1.18 | 63.86 ±0.61 | 29.04 ±1.64 | 10.23 ±1.27 | 51.74 ±0.60 | 41.36 ±1.16 | 49.29 ±0.68 | 39.16 ±1.60 | 36.80 ±1.16 |
| FedProx | 56.85 ±0.55 | 30.22 ±1.68 | 11.94 ±1.91 | 45.97 ±0.92 | 37.83 ±0.76 | 45.08 ±0.50 | 36.82 ±0.47 | 34.64 ±1.04 | 58.83 ±0.63 | 26.19 ±1.52 | 9.31 ±1.27 | 46.12 ±0.87 | 36.86 ±1.16 | 44.26 ±0.80 | 35.40 ±1.23 | 37.77 ±0.98 |
| FL-Coteaching | - | 41.19 ±1.04 | 25.98 ±1.87 | 58.84 ±0.63 | 50.56 ±1.26 | 58.13 ±0.36 | 53.42 ±0.68 | 48.02 ±1.02 | - | 36.64 ±1.60 | 24.81 ±1.03 | 57.70 ±0.54 | 50.71 ±0.95 | 56.80 ±0.40 | 52.35 ±1.21 | 46.50 ±0.96 |
| FL-DivideMix | - | 49.48 ±0.52 | **35.35** ±1.15 | 59.26 ±0.25 | 55.12 ±0.31 | 59.40 ±0.25 | 57.53 ±0.20 | 52.69 ±0.45 | - | 44.25 ±0.81 | 27.29 ±2.50 | 57.93 ±0.35 | 52.53 ±1.33 | 57.70 ±0.23 | 55.47 ±0.95 | 49.20 ±1.03 |
| FedCorr [CVPR22] | 70.77 ±2.11 | 58.29 ±1.30 | 27.54 ±2.04 | 67.04 ±0.49 | 59.58 ±0.70 | 66.56 ±0.67 | 61.41 ±1.03 | 56.74 ±1.04 | 64.46 ±3.54 | 55.83 ±0.40 | 19.02 ±1.52 | 61.29 ±0.90 | 52.18 ±1.61 | 61.45 ±0.90 | 53.94 ±1.61 | 50.62 ±1.17 |
| FedNoRo [IJCAI23] | - | 29.61 ±0.40 | 16.00 ±0.39 | 35.48 ±0.27 | 30.57 ±0.52 | 34.56 ±0.36 | 30.81 ±0.98 | 29.50 ±0.49 | - | 26.66 ±0.57 | 12.43 ±0.61 | 34.00 ±0.37 | 27.00 ±0.61 | 32.64 ±0.22 | 27.02 ±0.33 | 26.63 ±0.45 |
| FedDiv [AAAI24] | - | 37.14 ±4.20 | 4.13 ±3.25 | **69.37** ±0.42 | 60.26 ±1.56 | 66.65 ±0.66 | 62.64 ±0.42 | 59.21 ±1.25 | - | 10.18 ±1.05 | 1.11 ±0.18 | 39.49 ±7.41 | 39.59 ±5.80 | 42.32 ±9.67 | 33.20 ±5.33 | 27.19 ±0.42 |
| FedFixer [AAAI24] | - | 34.46 ±1.03 | 13.93 ±0.60 | 53.17 ±0.56 | 44.11 ±0.33 | 53.86 ±0.98 | 46.08 ±0.07 | 40.94 ±0.60 | - | 29.26 ±0.22 | 11.61 ±0.34 | 51.43 ±0.71 | 43.01 ±0.58 | 51.67 ±1.30 | 43.63 ±0.38 | 38.44 ±0.59 |
| FedGR [Ours] | **71.64** ±0.22 | **63.19** ±0.91 | 35.28 ±1.47 | 69.10 ±0.20 | **62.97** ±0.44 | **68.73** ±0.46 | **64.56** ±0.32 | **60.64** ±0.63 | **69.38** ±0.52 | **57.76** ±0.54 | **30.30** ±0.96 | **65.57** ±0.65 | **56.49** ±0.49 | **65.57** ±0.64 | **59.68** ±0.40 | **55.90** ±0.61 |

where $\tilde{\mathcal{D}}_k^r$ denotes the data subset with relatively reliable hard labels after LR which is initialized to $\emptyset$, *i.e.,*

$$\tilde{\mathcal{D}}_k^r = \tilde{\mathcal{D}}_k^r \cup \hat{\mathcal{D}}_k^c \cup \{(\mathbf{x}_i, y_i^{pse})|y_i^{pse} \neq \mathbf{0} \quad \forall i = 1, \cdots, n_k\}, \tag{16}$$

$\left\|\tilde{\mathcal{D}}_k^r\right\|/\left\|\hat{\mathcal{D}}_k\right\|$ refers to the proportion of the samples with reliable labels and $\mu$ is the corresponding threshold. Additionally, in order not to affect the quality of FS via regularization, globally revised EMA distillation is activated after $\alpha$ rounds, *i.e.,* $\lambda_\mathcal{B} = 0$ if $t < \alpha$.

### 3.5. Global Representation Regularization

Though the globally revised EMA distillation module can effectively regularize local training, the local model on a high label-noise-ratio client still inevitably overfits noisy labels. In light of the representation learning (Chen & He, 2021; Lubana et al., 2022) and our observation, we further introduce a global representation regularization module to regularize the local learning of the local model. To be specific, we adopt an instance-discriminative-like task, *i.e.,* the global representation of a weakly augmented image

should be consistent with the local representation of the strongly augmented image, as the regularization objective. Formally, the objective of regularization on $k$-th client at $t$-th round could be

$$\mathcal{R}_k = \mathbb{E}_{\hat{\mathcal{D}}_k} \left[ KL \left( \frac{f\left(\mathbf{x}_i^w; \mathbf{w}_{g,f}^{t-1}\right)}{\tau}, \frac{f\left(\mathbf{x}_i^s; \mathbf{w}_{k,f}^t\right)}{\tau} \right) \right]. \tag{17}$$

## 4. Experiments

We conduct experiments against eight baselines under I.I.D. and Non-I.I.D. FL settings with various label-noise levels and types to evaluate the effectiveness of the proposed FedGR. Then we perform ablations on the three main modules to investigate their effects and further analysis to show the superiority of the proposed FS. Please refer to the appendix A.5 for more experimental details, results, analysis, discussion, and data visualization.

**Datasets & F-LNL setups.** The experiments are conducted

*Table 3.* Results on Clothing1M. The 1st/2nd-best results are in a gray box w/. and w/o. boldface.

| Methods | FedAvg | FedProx | FL-Coteaching | FL-DivideMix | FedCorr [CVPR22] | FedDiv [AAAI24] | FedFixer [AAAI24] | FedGR [Ours] |
|---|---|---|---|---|---|---|---|---|
| I.I.D | $69.60_{\pm0.27}$ | $69.69_{\pm0.25}$ | $69.48_{\pm0.20}$ | $69.13_{\pm0.22}$ | $69.84_{\pm0.20}$ | $67.71_{\pm0.30}$ | $70.61_{\pm0.28}$ | $\mathbf{71.19}_{\pm\mathbf{0.42}}$ |
| Non.I.I.D-Dirichlet (0.3) | $67.90_{\pm1.31}$ | $68.18_{\pm1.13}$ | $68.16_{\pm0.82}$ | $63.88_{\pm1.21}$ | $60.42_{\pm7.23}$ | $65.79_{\pm2.77}$ | $65.16_{\pm0.95}$ | $\mathbf{68.52}_{\pm\mathbf{1.11}}$ |

*Table 4.* Ablation studies.

| Data Partition | I.I.D | | | Non.I.I.D-Dirichlet (0.3) | | |
|---|---|---|---|---|---|---|
| Noise Type | Sym | Asym | Mixed | Sym | Asym | Mixed |
| $\phi$ | 1.0 | 1.0 | 1.0 | 1.0 | 1.0 | 1.0 |
| $\mathcal{U}(\rho_{min},\rho_{max})$ | 0.5-1.0 | 0.2-0.4 | 0.2-0.4 | 0.5-1.0 | 0.2-0.4 | 0.2-0.4 |
| FedGR | 83.91 $\pm1.32$ | 91.64 $\pm0.38$ | 92.27 $\pm0.18$ | 63.64 $\pm5.39$ | 83.67 $\pm5.02$ | 84.65 $\pm2.38$ |
| w/o. FS | 54.59 $\pm3.04$ | 87.49 $\pm0.12$ | 91.71 $\pm0.13$ | 45.48 $\pm7.81$ | 83.45 $\pm4.49$ | 84.01 $\pm0.31$ |
| w/o. LR | 75.23 $\pm2.86$ | 90.42 $\pm0.13$ | 90.46 $\pm0.10$ | 59.48 $\pm5.85$ | 82.92 $\pm3.43$ | 83.21 $\pm1.89$ |
| w/o. $\mathcal{R}_k$ | 81.49 $\pm1.07$ | 91.22 $\pm0.34$ | 91.84 $\pm0.30$ | 58.23 $\pm4.08$ | 81.80 $\pm2.32$ | 82.70 $\pm0.63$ |
| w/o. $\mathcal{B}_k$ | 78.14 $\pm1.48$ | 91.54 $\pm0.24$ | 91.24 $\pm0.13$ | 51.07 $\pm5.16$ | 80.07 $\pm3.36$ | 79.44 $\pm2.79$ |

on CIFAR-10/100 and the real-world label-noise benchmark Clothing1M (Xiao et al., 2015) under various F-LNL settings. To simulate label-noise heterogeneity, we first partition CIFAR-10, CIFAR-100, and Clothing1M into 100, 100, and 500 FL clients under I.I.D. and extreme Dirichlet Non-I.I.D. (Li et al., 2022) settings, respectively. Subsequently, we perform a noisy-label synthesis process. Specifically, $\phi$ is introduced to control the proportion of clients affected by noisy labels. Next, the parameters $\rho_{min}$ and $\rho_{max}$ are used to bound a uniform distribution $\mathcal{U}(\rho_{min},\rho_{max})$, from which client $k$'s label-noise ratio is sampled. In addition to the label-noise ratio, the label-noise type (Song et al., 2023), *i.e.,* symmetric, asymmetric, or a mixture thereof, is also controlled. For instance, the *Mixed* setting in Table 1, 2, and 4 denotes that the noise type of each client is randomly assigned as either *Sym* or *Asym*. Consequently, I.I.D. and Non-I.I.D. in all tables in this study also denote label-noise heterogeneity and dual heterogeneity involving both label noise and data distribution, respectively.

**Baselines.** The experimental comparison employs eight baselines, which are divided into three groups: 1) the classic FL methods, *i.e.,* FedAvg (McMahan et al., 2017) and FedProx (Li et al., 2020b); 2) the typical C-LNL methods implemented in FL, *i.e.,* FL-Coteaching (Han et al., 2018) and FL-DivideMix (Li et al., 2020a); 3) the most recent F-LNL approaches, *i.e.,* FedCorr (Xu et al., 2022), Fed-NoRo (Wu et al., 2023), FedFixer (Ji et al., 2024), and FedDiv (Li et al., 2024).

**Implementation Details.** We report the mean test accuracy over the last 10 rounds instead of the best test accuracy, demonstrating the capability for preventing the overfitting of noisy labels and mitigating the substantial fluctuations. All the experiments are conducted three times with different random seeds and the mean and standard deviation are

reported. For Non.I.I.D. data partition, we follow (Li et al., 2022) to use the Dirichlet distribution to partition the data where $\alpha_{dirichlet} = 0.3$. As for the data augmentation, we adopt RandAugmentation (Cubuk et al., 2020) and the augmentation in (Xu et al., 2022) as the strong and weak data augmentation, respectively. Following (Xu et al., 2022), we adopt SGD as optimizer with a constant learning rate and use ResNet-18, ResNet-34, and pre-trained ResNet-50 as the backbone for CIFAR-10, CIFAR-100, and Clothing1M, respectively. The local epochs are set to 10 and 2 for CIFAR-10/100 and Clothing1M, respectively. $\lambda_{\mathcal{B}}$ and $\lambda_{\mathcal{R}}$ are set to 1.0 and 0.2 as default, respectively. For CIFAR-10, we decrease $\lambda_{\mathcal{R}}$ to 0.1, as the dataset is relatively simple.

*Table 5.* Further analysis.

| Data Partition | I.I.D | | | Non.I.I.D-Dirichlet (0.3) | | |
|---|---|---|---|---|---|---|
| Noise Type | Sym | Asym | Mixed | Sym | Asym | Mixed |
| $\phi$ | 1.0 | 1.0 | 1.0 | 1.0 | 1.0 | 1.0 |
| $\mathcal{U}(\rho_{min},\rho_{max})$ | 0.5-1.0 | 0.2-0.4 | 0.2-0.4 | 0.5-1.0 | 0.2-0.4 | 0.2-0.4 |
| FedGR | 83.91 $\pm1.32$ | 91.64 $\pm0.38$ | 92.27 $\pm0.18$ | 63.64 $\pm5.39$ | 83.67 $\pm5.02$ | 84.65 $\pm2.38$ |
| w/o. weak aug | 55.73 $\pm12.97$ | 89.93 $\pm0.29$ | 90.37 $\pm0.20$ | 25.32 $\pm4.36$ | 72.48 $\pm6.03$ | 78.54 $\pm4.22$ |
| w/o. strong aug | 34.51 $\pm3.26$ | 80.53 $\pm0.41$ | 78.72 $\pm0.68$ | 18.43 $\pm1.47$ | 59.74 $\pm2.01$ | 58.60 $\pm1.09$ |
| Online EMA distill | 82.80 $\pm1.44$ | 91.34 $\pm0.27$ | 92.04 $\pm0.29$ | 62.18 $\pm4.13$ | 82.50 $\pm5.79$ | 83.56 $\pm3.63$ |
| Randomly sample clients w/. replacement in warmup | 82.56 $\pm1.15$ | 91.21 $\pm0.46$ | 92.08 $\pm0.22$ | 62.03 $\pm5.85$ | 82.91 $\pm2.23$ | 83.19 $\pm2.52$ |

*Table 6.* Hyperparameter analysis for $\lambda_{\mathcal{B}}$ & $\lambda_{\mathcal{R}}$.

| Data Partition | I.I.D | | | Non.I.I.D-Dirichlet (0.3) | | |
|---|---|---|---|---|---|---|
| Noise Type | Sym | Asym | Mixed | Sym | Asym | Mixed |
| $\phi$ | 1.0 | 1.0 | 1.0 | 1.0 | 1.0 | 1.0 |
| $\mathcal{U}(\rho_{min},\rho_{max})$ | 0.5-1.0 | 0.2-0.4 | 0.2-0.4 | 0.5-1.0 | 0.2-0.4 | 0.2-0.4 |
| $\lambda_{\mathcal{B}}=1.0$ $\lambda_{\mathcal{R}}=0.1$ | 83.91 $\pm1.32$ | 91.64 $\pm0.38$ | 92.27 $\pm0.18$ | 63.64 $\pm5.39$ | 83.67 $\pm5.02$ | 84.65 $\pm2.38$ |
| $\lambda_{\mathcal{B}}=1.0$ $\lambda_{\mathcal{R}}=0.2$ | 73.16 $\pm11.26$ | 91.32 $\pm0.40$ | 92.26 $\pm0.17$ | 24.49 $\pm4.04$ | 82.44 $\pm4.33$ | 83.84 $\pm3.39$ |
| $\lambda_{\mathcal{B}}=1.0$ $\lambda_{\mathcal{R}}=0.5$ | 52.74 $\pm9.33$ | 91.87 $\pm0.20$ | 92.53 $\pm0.17$ | 32.94 $\pm5.57$ | 82.10 $\pm4.83$ | 84.47 $\pm2.93$ |
| $\lambda_{\mathcal{B}}=0.5$ $\lambda_{\mathcal{R}}=0.1$ | 83.18 $\pm0.80$ | 90.93 $\pm2.40$ | 91.73 $\pm0.20$ | 62.22 $\pm4.32$ | 79.60 $\pm4.95$ | 79.59 $\pm3.41$ |
| $\lambda_{\mathcal{B}}=1.0$ $\lambda_{\mathcal{R}}=0.1$ | 83.91 $\pm1.32$ | 91.64 $\pm0.38$ | 91.73 $\pm0.20$ | 63.64 $\pm5.39$ | 83.67 $\pm5.02$ | 84.65 $\pm2.38$ |
| $\lambda_{\mathcal{B}}=2.0$ $\lambda_{\mathcal{R}}=0.1$ | 83.68 $\pm0.45$ | 91.08 $\pm0.30$ | 92.13 $\pm0.14$ | 49.62 $\pm6.36$ | 83.66 $\pm5.87$ | 85.23 $\pm3.13$ |

**Comparison with State-Of-The-Arts.** We compared FedGR against eight baselines under diverse label-noise and data heterogeneity setups (Kim et al., 2022; Li et al., 2022) (Table 1– 3). In brief, the proposed FedGR achieves eye-catching performance and outperforms all the baselines by a considerable margin. Specifically, on CIFAR-10/-100, whenever the data is I.I.D. or Non.I.I.D., the pro-

*Table 7.* Hyperparameter analysis for $\gamma_g$ & $\gamma_l$.

| Data Partition | I.I.D | | | Non.I.I.D-Dirichlet (0.3) | | |
|---|---|---|---|---|---|---|
| Noise Type | Sym | Asym | Mixed | Sym | Asym | Mixed |
| $\phi$ | 1.0 | 1.0 | 1.0 | 1.0 | 1.0 | 1.0 |
| $\mathcal{U}(\rho_{min}, \rho_{max})$ | 0.5-1.0 | 0.2-0.4 | 0.2-0.4 | 0.5-1.0 | 0.2-0.4 | 0.2-0.4 |
| $\gamma_g = 0.9$ $\gamma_l = 0.99$ | 83.91 ±1.32 | 91.64 ±0.38 | 92.27 ±0.18 | 63.64 ±5.39 | 83.67 ±5.02 | 84.65 ±2.38 |
| $\gamma_g = 0.95$ $\gamma_l = 0.99$ | 80.97 ±0.49 | 90.82 ±0.57 | 91.79 ±0.22 | 62.18 ±4.51 | 82.07 ±4.90 | 81.82 ±3.92 |
| $\gamma_g = 0.99$ $\gamma_l = 0.99$ | 79.87 ±0.56 | 90.61 ±0.38 | 91.60 ±0.17 | 61.71 ±4.93 | 81.65 ±4.79 | 80.57 ±4.03 |
| $\gamma_g = 0.9$ $\gamma_l = 0.9$ | 80.13 ±0.38 | 90.94 ±0.38 | 91.75 ±0.25 | 61.83 ±4.72 | 81.19 ±4.57 | 80.09 ±3.85 |
| $\gamma_g = 0.9$ $\gamma_l = 0.99$ | 83.91 ±1.32 | 91.64 ±0.38 | 91.73 ±0.20 | 63.64 ±5.39 | 83.67 ±5.02 | 84.65 ±2.38 |
| $\gamma_g = 0.9$ $\gamma_l = 0.999$ | 84.85 ±0.57 | 92.39 ±0.23 | 92.14 ±0.14 | 50.86 ±4.13 | 79.00 ±3.87 | 81.28 ±3.34 |

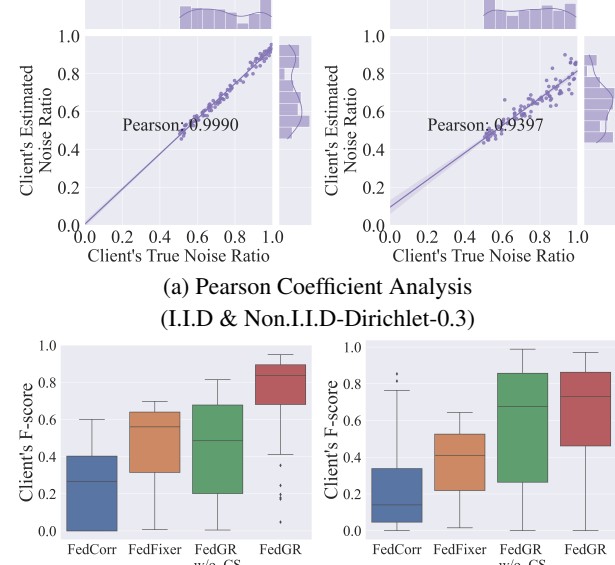

(a) Pearson Coefficient Analysis
(I.I.D & Non.I.I.D-Dirichlet-0.3)

(b) Clients' F-score Distributions
(I.I.D & Non.I.I.D-Dirichlet-0.3)

*Figure 3.* The (a) is the pearson coefficient analysis of the $\{r_k | k \in \mathcal{S}\}$. The (b) is the clients' F-score distributions of different methods. The F-LNL setup: CIFAR-10, Sym, $\phi = 1.0$, and $\mathcal{U}(0.5, 1.0)$.

posed FedGR could achieve the smallest performance gap to clean-data-trained models, if there are clean clients in a FL system ($\phi = 0.6$). Crucially, it remains robust in high-noise settings (*i.e.*, Sym, $\phi = 1.0$, and $\mathcal{U}(0.5, 1.0)$) where baselines like FedNoRo, FedDiv, and FedFixer fail. Notably, under the most challenging dual-heterogeneity setups, the proposed FedGR outperforms baselines by a considerable margin. Surprisingly, the proposed FedGR even outperforms FedAvg trained with clean data in some setups (*e.g.*, Mixed, $\phi = 0.6$, and $\mathcal{U}(0.2, 0.4)$) due to the regularization. Nevertheless, the magnitude of this effect is relatively small in the clean setting, whereas on noisy labels, the gains of FedGR over FedAvg are much more pronounced. This indicates that the primary benefit of FedGR indeed comes from its ability to handle label noise rather than regularization. Going beyond, the results in Table 3 verify the effectiveness of the FedGR on a large-scale real-world label-noise dataset Clothing1M [4]. The F-LNL-oriented baselines (*e.g.*, FedCorr, FedDiv, and FedFixer) do not achieve superior results to the proposed FedGR, especially under dual-heterogeneity setups. In conclusion, the proposed FedGR achieves a new state-of-the-art level of label-noise robustness.

The following experiments are conducted on CIFAR-10 unless otherwise specified.

**Ablations.** We evaluate the efficacy of FS by comparing it against a baseline that uses local model $\mathbf{w}_k^t$ for loss observation inference and perform GMM independently on each client. Then we compare LR against a vanilla strategy that trains solely on estimated local clean sets. We further set $\lambda_k = 0$ and $\lambda_b = 0$ to examine the contributions of $\mathcal{R}_k$ and $\mathcal{B}_k$, respectively. As reported in Table 4, FedGR achieves the best overall performance. The quality of sample sieving, *i.e.*, the use of FS, is particularly critical in challenging F-LNL settings (*e.g.*, Sym with $\phi = 1.0$ and $\mathcal{U}(0.5, 1.0)$). The LR component further improves performance by rectifying noisy labels. Additionally, both $\mathcal{R}_k$ and $\mathcal{B}_k$ yield further

---
[4]As discussed in `appendix A.5`, the Clothing1M is less faithful to the realistic F-LN problem than CIFAR setups.

performance gains.

**Further Analysis.** We also conduct further analysis to study the effects of data augmentation, the online EMA distillation for the globally revised EMA distillation module, and the client sampling strategy during warm-up (*i.e.*, whether all clients are guaranteed to be sampled at least once). As shown in Table 5, weak-strong augmentation substantially mitigates overfitting to noisy labels in FL. Additionally, replacing our globally revised EMA with online EMA distillation, or removing the warm-up sampling guarantee, results in slight performance degradation.

**Hyperparameters Analysis.** From Table 6, one can find $\lambda_{\mathcal{R}}$ should not be too large, as excessively strong regularization leads to performance degradation, while $\lambda_{\mathcal{B}}$ must balance regularization strength against overfitting to label noise. As for $\gamma_g$ and $\gamma_l$, it should satisfy $\gamma_g < \gamma_l$ as it is designed to resolve the conflict between the global and local models under dual heterogeneity. The analysis about them are shown in Table 7, and FedGR is quite robust to the choice of $\gamma_l$ when $\gamma_l > 0.9$, while using a larger $\gamma_g$ tends to degrade performance.

**FS Quality.** Figure 3 shows that FedGR can more accurately capture the relative noise ratios across clients (Pearson correlation $> 0.9$) and perform effective sample selection compared with FedCorr and FedFixer.

## 5. Conclusion

This study introduces FedGR, a novel approach for addressing the F-LN problem, inspired by the observation that the global model in FL exhibits a reduced propensity for label-noise overfitting, which has not been explored to the best of our knowledge. By strategically leveraging the global model through our proposed sieving-and-refining, globally revised EMA distillation, and global representation regularization modules, FedGR effectively enhances label-noise robustness while respecting the privacy constraints of FL. The comprehensive experiments across diverse and realistic F-LNL scenarios underscore the significant effectiveness of FedGR compared to existing state-of-the-art methods. We leave the theoretical analysis of this phenomenon for future investigation and intend to leverage this phenomenon to address the C-LN problem.

## Acknowledgements

We sincerely thank the anonymous reviewers for their insightful comments and constructive suggestions, which have significantly improved the quality of this paper. We also thank Xi Peng and Peng Hu for their valuable support. This work is supported by National Natural Science Foundation of China under Grant 62427820; in part by National Natural Science Foundation of China under Grant 62306198 & 624B2099, the Sichuan Science and Technology Program under Grant 2025ZDZX0125, the Science Fund for Creative Research Groups of Sichuan Province Natural Science Foundation under Grant 2024NSFTD0035, and the Ministry of Education Engineering Research Center Guiding Project for Machine Learning and Industrial Intelligence Applications under Grant SCU2024D013.

## Impact Statement

This paper presents work whose goal is to advance the field of Machine Learning. Overall, the main contribution concerns enhancing the robustness of FL systems against noisy labels without compromising user privacy. The central objective pertains to developing a self-contained, privacy-preserving method that allows decentralized clients to collaboratively train reliable models even when their local data annotations are flawed or of low quality.

From a broader societal perspective, our work has positive implications for privacy-sensitive domains such as healthcare, finance, and personal recommendation systems. By enabling robust learning from noisy, decentralized data, our approach lowers the barrier to deploying effective and fair machine learning models in real-world environments while strictly adhering to data minimization and privacy principles. There are many potential societal consequences of our work, none of which we feel must be specifically highlighted here as posing negative ethical impacts or dual-use risks.

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

# A. Appendix

## A.1. Overview of Appendix

In the following sections, we present

- the summary of mathematical notations,

- more observation experimental results on CIFAR-10 and CIFAR-100,

- the experimental results under the best test accuracy metric,

- the implementation details of the proposed FedGR,

- the hyperparameter configurations of the proposed FedGR and the baselines,

- the privacy risk statement of the proposed FedGR,

- the complexity analysis of the proposed FedGR,

- and the visualization of the FL data partition.

## A.2. Notations

For your reference, we summarize all the mathematical notations in Table 8 and 9.

*Table 8.* Table of Notations: Part I.

| Symbols | Section | Definition |
|---|---|---|
| $\mathcal{S}$ | 3.1 | The set of clients. |
| $K$ | 3.1 | The number of clients. |
| $t$ | 3.1 | $t$-th communication round. |
| $k$ | 3.1 | $k$-th client. |
| $\mathcal{S}(t)$ | 3.1 | The set of selected clients at $t$-th round. |
| $\ell(\cdot,\cdot)$ | 3.1 | Loss function. |
| $\mathcal{L}$ | 3.1 | Global training objective of F-LNL. |
| $\mathcal{L}_k$ | 3.1 | Local training objective of $k$-th client. |
| $\hat{\mathcal{D}}_k = \{(\mathbf{x}_i, \hat{y}_i)\}_{i=1}^{n_k}$ | 3.1 | The local dataset of $k$-th client, where $\mathbf{x}_i$, $\hat{y}_i$ and $n_k$ represents the data, the one-hot noisy label, and the size of the local dataset. |
| $\mathbf{p}_i^l$ | 3.1 | Output logits w.r.t the $\mathbf{x}_i$ and the local model parameters $\mathbf{w}_k^t$ of $k$-th client. |
| $f(\cdot; \mathbf{w}_{k,f}^t)$ | 3.1 | The backbone network of $k$-th client at $t$-th round, where $\mathbf{w}_{k,f}^t$ denotes the parameters. |
| $h(\cdot; \mathbf{w}_{k,h}^t)$ | 3.1 | The classification head network of $k$-th client at $t$-th round, where $\mathbf{w}_{k,h}^t$ denotes the parameters. |
| $\mathbf{w}_k^t = \{\mathbf{w}_{k,h}^t, \mathbf{w}_{k,f}^t\}$ | 3.1 | The local model parameters of $k$-th client at $t$-th round. |
| $\mathbf{w}_g^t$ | 3.1 | The global model parameters at $t$-th round. |
| $a_k = n_k / \sum_{i \in \mathcal{S}(t)} n_i$ | 3.1 | The corresponding importance weight for FedAvg at $t$-th round. |

*Table 9.* Table of Notations: Part II.

| Symbols | Section | Definition |
|---|---|---|
| $\mathcal{L}_k^{SR}$ | 3.2 | The objective of sieving-and-refining module on $k$-th client. |
| $\mathcal{B}_k$ | 3.2 | The objective globally revised EMA distillation module on $k$-th client. |
| $\mathcal{R}_k$ | 3.2 | The objective global representation regularization module on $k$-th client. |
| $\lambda_{\mathcal{B}}$ | 3.2 | The coefficient of $\mathcal{B}_k$. |
| $\lambda_{\mathcal{R}}$ | 3.2 | The coefficient of $\mathcal{R}_k$. |
| $\alpha$ | 3.3 | The number warmup rounds. |
| $\mathbf{p}_i^{l,s}$ | 3.3 | The output logits of strong augmented data $\mathbf{x}_i^s$ on $k$-th client at $t$-th round. |
| $\tilde{y}_i, \tilde{y}_i^{pse}$ | 3.3 | The refined and pseudo label of data $\mathbf{x}_i$ on $k$-th client. |
| $\ell_{i,T_k}$ | 3.3 | The loss observation of data $\mathbf{x}_i$ on $k$-th client at $t$-th round, where $T_k$ represents the number of selected times for $k$-th client in last $t$ rounds. |
| $L_i^t$ | 3.3 | The mean inference loss of data $\mathbf{x}_i$ on $k$-th client at $t$-th round. |
| $d_{i,k}$ | 3.3 | The global unique identifier of data $\mathbf{x}_i$ on $k$-th client. |
| $q_{i,k}$ | 3.3 | The "clean" GMM posterior probability of data $\mathbf{x}_i$ on $k$-th client. |
| $r_k$ | 3.3 | The estimated noise ratio of $k$-th client. |
| $\hat{\mathcal{D}}^c, \hat{\mathcal{D}}^n$ | 3.3 | The partitioned clean and noisy data subset on $k$-th client. |
| $\beta$ | 3.3 | The label-noise ratio threshold. |
| $\gamma_g, \gamma_l$ | 3.4 | The momentum coefficient for global-model-revised EMA step and usual local EMA step. |
| $m_k$ | 3.4 | The number of local training steps on $k$-th client. |
| $\mathbf{w}_{k,ema}^t$ | 3.4 | The parameters of local EMA on $k$-th client at $t$-th round. |
| $\mathbf{p}_i^{le/l,w}$ | 3.4 | The local EMA model/local model output logits of weak augmented data $\mathbf{x}_i^w$ on $k$-th client at $t$-th round. |
| $KL(\cdot, \cdot)$ | 3.4 | The Kullback–Leibler divergence loss. |
| $\tau$ | 3.4 | The temperature of knowledge distillation. |
| $\tilde{\mathcal{D}}_k^r$ | 3.4 | The data subset with relatively reliable hard labels after LR. |
| $\mu$ | 3.4 | The threshold for the data subset with relatively reliable hard labels. |

## A.3. Observation Experiment

In this section, we provides additional observation results on CIFAR-10 and CIFAR-100 to support that **the global model of FL memorizes noisy labels slowly and is capable to maintain reliable predictions and robust representation**. In addition to guarantee the reproduction, we also provide the corresponding implementation details of these observation experiments. Specifically, to investigate the memorization effect (Arpit et al., 2017) of the global model of FL, we first analyze the difference in noisy labels overfitting between the centralized trained model and the global model of FL under all possible F-LNL setups, including dual heterogeneity under extreme data heterogeneity (Dirichlet-0.3). The results are shown in Figure 5, 6, 7, 8, 9, and 10. Then, to show this phenomenon in more detail, we report the difference in the degree of overfitting between the client's local model and the global model of FL on the noisy labels of each client, showing in Figure 12, 14, 16, 13, 15, and 17. Additionally, we also provide the observation results under different noise ratios (*i.e.,* 50% and 80%) and different client scales (*i.e.,* 20 and 100), showing in Figure 11.

**Implementation details of the observations.** The observation experiments are conducted on CIFAR-10 and CIFAR-100 with ResNet18 (He et al., 2016) as the backbone. For federated learning (FL) (McMahan et al., 2017), we partition the data into 10 clients in I.I.D and Non.I.I.D. manner (Dirichlet (Li et al., 2022), $\alpha_{dirirchlet} = 0.3$). And we adopt an SGD optimizer with a constant learning rate of 0.01, weight decay of 5e-4, and momentum of 0.5. Then, we set the epochs of centralized training and the communication rounds of FL to 200 and the random seed to 1. For hyperparameter configurations of the observation experiments, please refer to Table 10 to 13.

**Discussions.** The above observation experiments prove that such a phenomenon is reusable and non-trivial. Specifically, we have the following observations:

- Showing from Figure 5 to 10, under moderate noisy ratio, unlike the centralized trained model, the global model of FL memorizes less noisy labels throughout training and does not suffer from a drop in test performance across different local epochs, client sample ratios, and extreme data heterogeneity.

- Additionally, as shown from Figure 12 to 17, the global model of FL memorizes fewer label errors than the client's local model under diverse client sampling ratios.

- As shown in Figure 11, when the noisy ratio is high (*e.g.,* $\geq 50\%$), the global model of FL still memorizes noisy labels more slowly than the centralized trained model and does not suffer from a drop in test performance. However, the global model of FL may not achieve the peak test performance of the centralized trained model.

Going beyond the F-LN problem, we believe this intriguing phenomenon could also inspire future research about learning with noisy labels in centralized learning. In other words, **can we design training algorithms in centralized learning that can mimic the implicit regularization effects of FL to resist label noise?** We leave this as an interesting open problem for future work.

Kim et al. is the most relevant work to this study. They observe that the local model of FL memorizes noisy labels faster than the global model of FL and subsequently propose a regularization method to alleviate this issue. This study goes beyond their observations in two aspects. First, this study conducts more comprehensive experimental analysis to validate the phenomenon. We comprehensively investigate the memorization effect of the global model of FL under all possible F-LNL setups on two widely-used benchmarks, including dual heterogeneity under extreme data heterogeneity (Dirichlet-0.3). These results further confirm such a phenomenon is not an artifact of specific experimental settings. Second, we further analyze the difference in the degree of label-noise overfitting between the global model of FL and the model trained in centralized learning, revealing that the global model of FL memorizes noisy labels more slowly than the centrally trained model. This phenomenon draws an intriguing connection between FL and learning with noisy labels in centralized learning, which could inspire future research in both fields.

*Table 10.* Shared hyperparameter configurations of the observation experiments.

| Scenario | Centralized Learning | Federated Learning | |
|---|---|---|---|
| | | I.I.D | Non.I.I.D |
| Data Partition | - | I.I.D | Non.I.I.D |
| Dataset | CIFAR-10/100 | CIFAR-10/100 | CIFAR-10/100 |
| # of client | - | 10 | 10 |
| $\alpha_{dirichlet}$ | - | - | 0.3 |
| Optimizer | SGD | SGD | SGD |
| SGD Momentum | 0.5 | 0.5 | 0.5 |
| Weight Decay | 5e-4 | 5e-4 | 5e-4 |
| Network | ResNet18 | ResNet18 | ResNet18 |
| Learning Rate | 0.01 | 0.01 | 0.01 |
| Epochs/Rounds | 200 | 200 | 200 |
| Random Seed | 1 | 1 | 1 |

*Table 11.* Other hyperparameter configurations of the observation experiments.

| Scenario | Centralized Learning | Federated Learning | | | |
|---|---|---|---|---|---|
| | | I.I.D | | Non.I.I.D | |
| Data Partition | - | I.I.D | | Non.I.I.D | |
| Local Epoch | - | 1 | 10 | 1 | 10 |
| Sample Ratio | - | 0.2 | 0.2 | 0.2 | 0.2 |
| Batch Size | 64 | 32 | 32 | 32 | 32 |
| Noise Ratio (%) | 18.66/15.36 | 18.66 | 18.66 | 15.36 | 15.36 |
| Local Epoch | - | 1 | 10 | 1 | 10 |
| Sample Ratio | - | 0.5 | 0.5 | 0.5 | 0.5 |
| Batch Size | 160 | 32 | 32 | 32 | 32 |
| Noise Ratio (%) | 18.66/15.36 | 18.66 | 18.66 | 15.36 | 15.36 |
| Local Epoch | - | 1 | 10 | 1 | 10 |
| Sample Ratio | - | 1.0 | 1.0 | 1.0 | 1.0 |
| Batch Size | 320 | 32 | 32 | 32 | 32 |
| Noise Ratio (%) | 18.66/15.36 | 18.66 | 18.66 | 15.36 | 15.36 |

*Table 12.* Shared hyperparameter configurations of the observation experiments under different label-noise ratios and client scales.

| Scenario | Centralized Learning | Federated Learning | |
|---|---|---|---|
| | | I.I.D | Non.I.I.D |
| Data Partition | - | I.I.D | Non.I.I.D |
| Dataset | CIFAR-10 | CIFAR-10 | CIFAR-10 |
| # of client | - | 20/100 | 20/100 |
| $\alpha_{dirichlet}$ | - | - | 0.3 |
| Optimizer | SGD | SGD | SGD |
| SGD Momentum | 0.5 | 0.5 | 0.5 |
| Weight Decay | 5e-4 | 5e-4 | 5e-4 |
| Network | ResNet18 | ResNet18 | ResNet18 |
| Learning Rate | 0.01 | 0.01 | 0.01 |
| Epochs/Rounds | 200 | 200 | 200 |
| Random Seed | 1 | 1 | 1 |

*Table 13.* Other hyperparameter configurations of the observation experiments under different label-noise ratios and client scales.

| Scenario | Centralized Learning | Federated Learning | |
|---|---|---|---|
| | | I.I.D | Non.I.I.D |
| Data Partition | - | I.I.D | Non.I.I.D |
| Local Epoch | - | 1 | 1 |
| Sample Ratio | - | 0.2 | 0.2 |
| Batch Size | 64 | 32 | 32 |
| Noise Ratio (%) | 50/80 | 50/80 | 50/80 |

One possible explanation for this intriguing phenomenon is as follows. In both centralized learning and FL, models first learn the pattern of clean data from the whole training set (Arpit et al., 2017). As training proceeds, sooner or later, both will fit all noisy labels. However, in FL, the vanilla FedAvg can be interpreted as a weight-space ensemble over a collection of client-specific optimization trajectories that repeatedly emanate from, and are re-synchronized at, a shared global model. With moderate, heterogeneous noisy labels across clients, the gradients driven by the clean patterns tend to be reasonably well aligned across clients, whereas the gradients induced by noisy labels are highly client-specific and exhibit low cross-client correlation. **Consequently, when the server aggregates local updates, the coherent signal updates are systematically reinforced, while the incoherent noise updates tend to cancel out in expectation.** Moreover, the temporal dynamics of FedAvg—periodically restarting local optimization from the current global model and constraining the number of local update steps per communication round—induces an implicit early-stopping mechanism: local models spend most of their training in the early-learning regime, where deep networks preferentially fit shared, simple and clean data, and their subsequent tendency to memorize idiosyncratic noisy labels is repeatedly truncated and smoothed by aggregation. **Consequently, FL seems to extend the early learning period, which memorizes less noisy labels.**

## A.4. Extra Experimental Comparison Results

To better reflect label-noise robustness and the ability to avoid memorizing noisy labels, we report the average test accuracy over the last 10 rounds as the metric in the main paper. The reasons are as follows.

- In practice, however, one cannot rely on oracle early stopping based on the test set to halt exactly at that single best round, as no one knows whether continuing the training would actually improve or worsen the performance.

- In F-LNL, the test accuracy curves of many methods exhibit substantial fluctuations, as shown in Fig.4. And some baselines reach a very sharp peak at an intermediate epoch and then deteriorate as they start memorizing noisy labels.

In this section, we still report the best test accuracy for reference. From Table 14 to 16, it can be seen that the proposed FedGR still achieves strong performance on CIFAR-10/-100 and Clothing1M. This confirms that the performance gains of FedGR are not an artifact of the particular evaluation protocol.

*Table 14.* The mean of the best results, averaged across three trials on CIFAR-10. The 1st/2nd-best results are in a gray box w/. and w/o. boldface.

| Data Partition | I.I.D | | | | | | | | Non.I.I.D-Dirichlet (0.3) | | | | | | | |
| --- | --- | --- | --- | --- | --- | --- | --- | --- | --- | --- | --- | --- | --- | --- | --- | --- |
| Noise Type | Clean | Sym | | Asym | | Mixed | | | Clean | Sym | | Asym | | Mixed | | |
| $\phi$ | 0.0 | 0.6 | 1.0 | 0.6 | 1.0 | 0.6 | 1.0 | Avg | 0.0 | 0.6 | 1.0 | 0.6 | 1.0 | 0.6 | 1.0 | Avg |
| $\mathcal{U}(\rho_{min}, \rho_{max})$ | 0.0-0.0 | 0.5-1.0 | 0.5-1.0 | 0.2-0.4 | 0.2-0.4 | 0.2-0.4 | 0.2-0.4 | | 0.0-0.0 | 0.5-1.0 | 0.5-1.0 | 0.2-0.4 | 0.2-0.4 | 0.2-0.4 | 0.2-0.4 | |
| FedAvg | 92.69 ± 0.08 | 70.93 ± 0.40 | 32.92 ± 0.86 | 85.98 ± 0.21 | 73.89 ± 0.40 | 82.98 ± 0.69 | 72.36 ± 0.26 | 73.11 ± 0.41 | 89.57 ± 0.39 | 50.64 ± 1.67 | 22.97 ± 1.55 | 75.05 ± 0.15 | 57.49 ± 0.95 | 70.70 ± 0.34 | 55.03 ± 1.01 | 60.21 ± 0.87 |
| FedProx | 90.80 ± 0.19 | 65.68 ± 0.71 | 35.25 ± 2.14 | 82.05 ± 0.49 | 68.29 ± 0.64 | 79.31 ± 0.31 | 67.51 ± 0.03 | 69.84 ± 0.64 | 87.69 ± 0.53 | 48.99 ± 2.03 | 24.24 ± 2.06 | 73.94 ± 0.53 | 56.73 ± 0.04 | 67.51 ± 0.45 | 54.19 ± 0.21 | 59.04 ± 0.84 |
| FL-Coteaching | - | 70.90 ± 0.49 | 51.54 ± 2.83 | 88.38 ± 0.27 | 84.37 ± 0.18 | 87.12 ± 0.16 | 84.65 ± 0.33 | 77.83 ± 0.71 | - | 51.30 ± 0.54 | 38.10 ± 1.27 | 80.45 ± 0.23 | 75.78 ± 1.00 | 77.80 ± 0.62 | 74.28 ± 0.89 | 66.29 ± 0.76 |
| FL-DivideMix | - | 77.20 ± 0.27 | 69.85 ± 2.12 | 85.78 ± 0.19 | 86.59 ± 0.51 | 84.87 ± 0.36 | 85.49 ± 0.39 | 81.63 ± 0.64 | - | 61.78 ± 0.52 | 42.72 ± 3.06 | 74.99 ± 1.60 | 73.23 ± 0.89 | 72.09 ± 0.31 | 70.94 ± 0.54 | 65.96 ± 1.15 |
| FedCorr [CVPR22] | 92.73 ± 0.54 | 92.43 ± 0.12 | 59.00 ± 1.18 | 86.18 ± 0.21 | 86.38 ± 1.52 | 85.77 ± 1.11 | 86.15 ± 0.31 | 82.65 ± 0.74 | | 84.35 ± 0.51 | 33.76 ± 2.79 | 64.11 ± 8.25 | 63.11 ± 6.57 | 86.34 ± 1.08 | 74.87 ± 0.50 | 67.76 ± 3.28 |
| FedNoRo [IJCAI23] | - | 63.75 ± 0.96 | 36.90 ± 4.22 | 72.51 ± 0.10 | 65.11 ± 1.19 | 71.47 ± 0.19 | 64.11 ± 0.35 | 62.31 ± 1.17 | | 53.57 ± 1.62 | 21.44 ± 1.49 | 61.22 ± 1.51 | 44.32 ± 2.32 | 59.41 ± 1.06 | 47.23 ± 1.04 | 47.87 ± 1.51 |
| FedDiv [AAAI24] | - | 90.62 ± 0.65 | 37.17 ± 19.55 | 93.93 ± 0.10 | 93.07 ± 0.17 | 93.63 ± 0.13 | 92.56 ± 0.09 | 83.50 ± 3.45 | | 46.61 ± 14.98 | 22.12 ± 7.61 | 80.83 ± 0.94 | 72.87 ± 6.65 | 76.05 ± 5.58 | 52.71 ± 13.17 | 58.53 ± 8.16 |
| FedFixer [AAAI24] | - | 73.94 ± 0.68 | 47.17 ± 2.06 | 86.01 ± 0.39 | 75.03 ± 1.29 | 86.60 ± 0.51 | 79.65 ± 0.68 | 74.73 ± 0.94 | | 63.04 ± 1.62 | 34.78 ± 2.88 | 77.52 ± 1.57 | 63.43 ± 2.01 | 79.09 ± 1.79 | 68.71 ± 1.92 | 64.43 ± 1.97 |
| FedGR [Ours] | **94.21** ± 0.08 | 92.40 ± 0.12 | **84.66** ± 1.03 | 93.62 ± 0.29 | 92.06 ± 0.23 | 93.34 ± 0.23 | 92.55 ± 0.20 | **91.83** ± 0.31 | 88.93 ± 1.49 | **85.68** ± 0.49 | **65.70** ± 4.34 | **89.71** ± 0.26 | **87.77** ± 1.12 | **89.58** ± 0.21 | **87.65** ± 0.56 | **85.00** ± 1.21 |

*Table 15.* The mean of the best results, averaged across three trials on CIFAR-100. The 1st/2nd-best results are in a gray box w/. and w/o. boldface.

| Data Partition | I.I.D | | | | | | | | Non.I.I.D-Dirichlet (0.3) | | | | | | | |
| --- | --- | --- | --- | --- | --- | --- | --- | --- | --- | --- | --- | --- | --- | --- | --- | --- |
| Noise Type | Clean | Sym | | Asym | | Mixed | | | Clean | Sym | | Asym | | Mixed | | |
| $\phi$ | 0.0 | 0.6 | 1.0 | 0.6 | 1.0 | 0.6 | 1.0 | Avg | 0.0 | 0.6 | 1.0 | 0.6 | 1.0 | 0.6 | 1.0 | Avg |
| $\mathcal{U}(\rho_{min}, \rho_{max})$ | 0.0-0.0 | 0.5-1.0 | 0.5-1.0 | 0.2-0.4 | 0.2-0.4 | 0.2-0.4 | 0.2-0.4 | | 0.0-0.0 | 0.5-1.0 | 0.5-1.0 | 0.2-0.4 | 0.2-0.4 | 0.2-0.4 | 0.2-0.4 | |
| FedAvg | 65.17 ± 0.20 | 37.57 ± 0.74 | 16.48 ± 0.65 | 55.79 ± 0.75 | 46.03 ± 0.41 | 53.17 ± 0.11 | 44.12 ± 0.27 | 45.48 ± 0.45 | 64.51 ± 0.48 | 33.85 ± 0.46 | 12.70 ± 0.65 | 53.70 ± 0.71 | 42.28 ± 0.69 | 50.73 ± 0.63 | 40.14 ± 1.18 | 42.56 ± 0.69 |
| FedProx | 57.24 ± 0.29 | 33.10 ± 0.82 | 15.37 ± 0.47 | 47.20 ± 0.38 | 39.31 ± 0.46 | 45.80 ± 0.29 | 37.74 ± 0.40 | 39.39 ± 0.40 | 59.38 ± 0.38 | 30.47 ± 0.33 | 11.38 ± 0.96 | 47.97 ± 0.68 | 37.84 ± 0.85 | 45.52 ± 0.86 | 36.28 ± 0.86 | 38.41 ± 0.67 |
| FL-Coteaching | - | 42.87 ± 0.31 | 27.76 ± 1.23 | 59.74 ± 0.25 | 52.15 ± 0.93 | 58.78 ± 0.14 | 53.91 ± 0.47 | 49.20 ± 0.56 | - | 39.15 ± 0.60 | 24.90 ± 1.11 | 59.02 ± 0.43 | 50.78 ± 1.39 | 57.48 ± 0.42 | 52.42 ± 0.82 | 47.29 ± 0.80 |
| FL-DivideMix | - | 49.61 ± 1.18 | **36.30** ± 1.60 | 58.17 ± 1.42 | 54.29 ± 0.90 | 58.02 ± 1.40 | 56.54 ± 1.49 | 52.16 ± 1.33 | - | 46.29 ± 0.39 | 29.28 ± 1.22 | 57.42 ± 0.75 | 51.47 ± 0.70 | 57.11 ± 0.34 | 54.26 ± 1.22 | 49.31 ± 0.77 |
| FedCorr [CVPR22] | 71.24 ± 1.45 | 60.10 ± 1.32 | 29.94 ± 1.55 | 67.83 ± 0.45 | 61.05 ± 0.67 | 67.29 ± 0.85 | 62.13 ± 0.89 | 58.06 ± 0.96 | 65.38 ± 3.78 | 57.53 ± 0.52 | 21.55 ± 1.95 | 62.91 ± 0.98 | 54.28 ± 1.81 | 63.01 ± 1.95 | 55.33 ± 0.70 | 52.44 ± 1.32 |
| FedNoRo [IJCAI23] | - | 30.03 ± 0.40 | 17.08 ± 0.53 | 35.95 ± 0.35 | 31.00 ± 0.56 | 35.10 ± 0.36 | 31.69 ± 0.67 | 30.14 ± 0.48 | - | 27.43 ± 0.41 | 13.17 ± 0.85 | 34.34 ± 0.30 | 27.46 ± 0.54 | 33.16 ± 0.10 | 27.55 ± 0.32 | 27.19 ± 0.42 |
| FedDiv [AAAI24] | - | 40.70 ± 2.47 | 9.01 ± 2.67 | **69.90** ± 0.31 | 60.72 ± 1.59 | 67.12 ± 0.81 | 63.16 ± 0.39 | 51.77 ± 1.37 | - | 22.50 ± 1.41 | 5.62 ± 1.42 | 57.12 ± 2.69 | 49.90 ± 1.37 | 55.74 ± 1.16 | 47.79 ± 1.43 | 39.78 ± 1.58 |
| FedFixer [AAAI24] | - | 37.96 ± 0.57 | 19.22 ± 0.41 | 55.15 ± 0.59 | 46.06 ± 0.55 | 54.70 ± 1.03 | 47.20 ± 0.24 | 43.38 ± 0.57 | - | 34.09 ± 0.13 | 16.06 ± 0.24 | 53.92 ± 1.19 | 44.11 ± 0.61 | 52.82 ± 1.24 | 44.79 ± 0.36 | 40.97 ± 0.63 |
| FedGR [Ours] | **72.03** ± 0.12 | **64.01** ± 0.46 | 35.82 ± 1.43 | 69.48 ± 0.22 | **64.12** ± 0.10 | **69.21** ± 0.16 | **65.15** ± 0.16 | **62.83** ± 0.38 | **70.36** ± 0.06 | **59.07** ± 0.58 | **31.89** ± 0.49 | **66.34** ± 0.38 | **58.00** ± 0.67 | **66.54** ± 0.40 | **61.01** ± 0.42 | **59.03** ± 0.43 |

*Table 16.* The mean of the best results, averaged across three trials on Clothing1M. The 1st/2nd-best results are in a gray box w/. and w/o. boldface.

| Methods | FedAvg | FedProx | FL-Coteaching | FL-DivideMix | FedCorr [CVPR22] | FedDiv [AAAI24] | FedFixer [AAAI24] | FedGR [Ours] |
|---|---|---|---|---|---|---|---|---|
| I.I.D | $70.24 \pm 0.11$ | $70.21 \pm 0.06$ | $69.73 \pm 0.15$ | $69.48 \pm 0.15$ | $70.11 \pm 0.25$ | $67.97 \pm 0.35$ | $71.18 \pm 0.33$ | $\mathbf{71.71 \pm 0.16}$ |
| Non.I.I.D-Dirichlet (0.3) | $70.47 \pm 0.55$ | $70.23 \pm 0.39$ | $69.17 \pm 0.26$ | $64.42 \pm 1.33$ | $64.47 \pm 8.73$ | $66.62 \pm 2.48$ | $69.65 \pm 1.27$ | $\mathbf{70.98 \pm 0.23}$ |

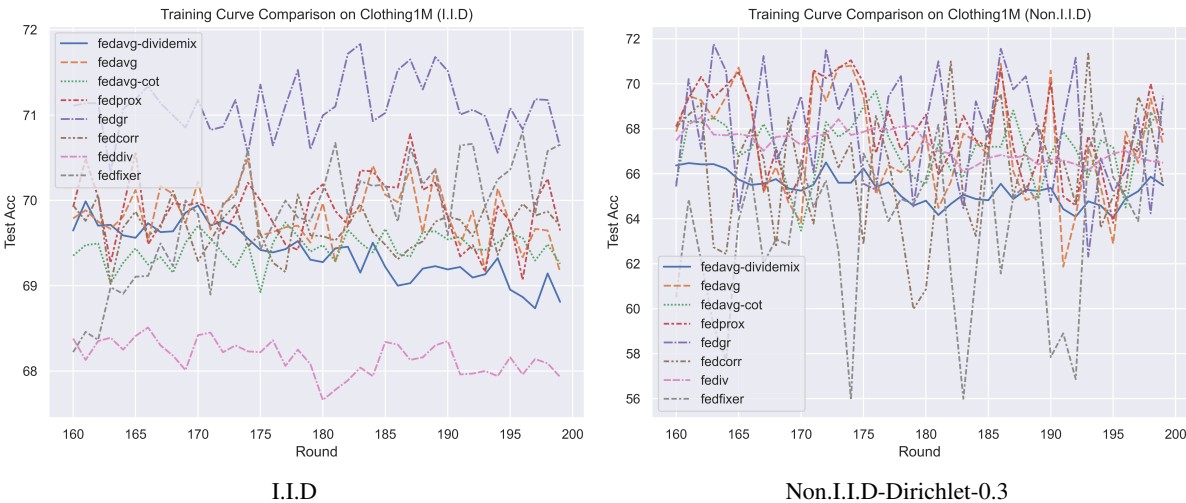

I.I.D        Non.I.I.D-Dirichlet-0.3

*Figure 4.* The (a) and (b) are the test accuracy curve of different methods on Clothing1M.

## A.5. Implementation Details and Baselines

**Implementation Details of FedGR .** This study adopts a model consisting of a network backbone $f(\cdot)$ and a linear classification head $h(\cdot)$ for all experiments. Due to the randomness of the data partition, the experiments on CIFAR-10, CIFAR-100, and Clothing1M are carried out three times with various random seeds (1, 13, and 42). The entire hyperparameter configuration that the proposed FedGR adopted is listed in Table 17. These hyperparameters are divided into three groups, namely the parameters for FL setups, the optimization configurations of the client's local training, and the specific hyperparameters for the proposed FedGR . All the experiments have a batch size of 32 and are supported by NVIDIA RTX 3090. The pseudo code of the proposed FedGR is described in Algorithm 1 and Algorithm 2.

**Hyperparameters of Baselines.** Our code repository implements the typical centralized learning with noisy labels (C-LNL) approaches (FL-Coteaching (Han et al., 2018), FL-DivideMix (Li et al., 2020a)). For the robust label-noise F-LNL methods, we use their official implementations for reproduction of results. Some common hyperparameters, such as the optimization configurations, are shown in Table 17. We will briefly describe the specific hyperparameters these baselines use by adopting the same mathematical notations used in their papers. Notably, all the baselines adopt the same optimization configurations as FedGR.

- **FedProx** (Li et al., 2020b) is proposed to tackle the data heterogeneity (Li et al., 2022) by introducing a model parameter proximal term to the local learning objective with hyperparameter $\mu_{prox}$. In this study, we follow FedCorr (Xu et al., 2022) to set $\mu_{prox} = 1$ for all experiments.

- **FL-Coteaching.** Following the original settings (Han et al., 2018), we let the decay of the client-specific sample drop rate be $R(T) = 1 - \tau \cdot \{T^c/T_k, 1\}$ with $c = 1$, $T_k = 25$, where $T$ denotes the $T$-th communication round and the client-specific noise level $\tau$ is known to be the expectation of the uniform distribution. Formally, we let $\tau = \mathbb{E}[\rho_k | \rho_k \sim \mathcal{U}(\rho_{min}, \rho_{max})]$ for the experiments on CIFAR-10/100. For Clothing1M (Xiao et al., 2015), we set $\tau = 0.39$, as it contains natural label noise in a ratio of 39.46%.

- **FL-DivideMix.** The DivideMix (Li et al., 2020a) is another milestone C-LNL method. Following the original settings (Li et al., 2020a), we let the sharpen temperature $\tau_{dividemix}$ be 0.5, the Gaussian mixture model (GMM)

posterior probability threshold be 0.5, the weight of the unsupervised loss $\lambda_u$ be 25, the hyperparameter of Mixup's beta distribution $\beta_{mixup}$ be 4, and the warm-up rounds be 100 for all experiments.

- **FedCorr** (Xu et al., 2022) is one of the early efforts to achieve label-noise robustness in FL. We use its official-released implementation and hyperparameter configurations to carry out all the experiments except for the F-LNL setups, learning rate, batch size, local epochs, SGD weight decay, and SGD momentum.

- **FedNoRo** (Wu et al., 2023) is another label-noise robust F-LNL approach, which detects the clients with noisy labels and rectifies them with soft labels. We also follow its official implementation to perform all the experiments on CIFAR-10/100. Specifically, we let the warm-up rounds be 100 and the F-LNL scenarios be the same as the proposed FedGR .

- **FedFixer** (Ji et al., 2024). With the official implementation, we carry out all the experiments with the originally reported hyperparameters (Ji et al., 2024) except for the F-LNL setups, learning rate, batch size, local epochs, SGD weight decay, and SGD momentum.

- **FedDiv** (Li et al., 2024). We carry out all the experiments with the originally reported hyperparameters (Ji et al., 2024) except for the F-LNL setups, learning rate, batch size, local epochs, SGD weight decay, and SGD momentum.

**Limitations of the Clothing1M.** To be specific, we have to partition Clothing1M using the given noisy labels. This implicitly imposes a strong and somewhat unrealistic assumption: all clients share the same label-noise pattern, i.e., the same noise transition matrix. Thus, this scenario underestimates the potential benefits of the proposed FedGR for modeling heterogeneous label noise.

*Table 17.* List of the hyperparameters used for FedGR.

| Dataset | CIFAR-10 | CIFAR-100 | Clothing1M | Remark |
|---|---|---|---|---|
| Size of $\mathcal{D}_{train}$ | 50,000 | 50,000 | 1,000,000 | - |
| # of classes | 10 | 100 | 14 | - |
| # of clients | 100 | 100 | 500 | - |
| # of rounds | 500 | 500 | 200 | - |
| Sample Ratio | 0.1 | 0.1 | 0.02 | Fraction of the participated clients of each round |
| $\alpha_{dirichlet}$ | 0.3 | 0.3 | 0.3 | The hyperparameter of the Dirichlet distribution |
| Backbone | ResNet-18 | ResNet-34 | pre-trained ResNet-50 | - |
| Optimizer | SGD | SGD | SGD | - |
| LR | 0.01 | 0.01 | 0.01 | Learning rate for the SGD optimizer |
| SGD Momentum | 0.5 | 0.5 | 0.5 | Momentum for the SGD optimizer |
| Weight Decay | 5e-4 | 5e-4 | 5e-4 | - |
| Batch Size | 32 | 32 | 32 | - |
| Local Epoch | 10 | 10 | 2 | - |
| $\lambda_{\mathcal{B}}$ | 1.0 | 1.0 | 1.0 | Weight of the global revised EMA distillation $\mathcal{B}_k$ |
| $\lambda_{\mathcal{R}}$ | 0.1 | 0.2 | 0.2 | Weight of the global representation regularization $\mathcal{R}_k$ |
| $\alpha$ | 100 | 100 | 50 | Rounds of the label-noise sniffing |
| $\epsilon$ | 0.9 | 0.9 | 0.9 | Pseudo labeling confidence threshold |
| $\beta$ | 0.8 | 0.8 | 0.8 | The threshold of the client's label-noise rate |
| $\gamma_l$ | 0.99 | 0.99 | 0.99 | Momentum decay of the standard local EMA update |
| $\gamma_g$ | 0.9 | 0.9 | 0.9 | Momentum decay of the global revised EMA update |
| $\tau$ | 0.5 | 0.5 | 0.5 | Distillation temperature of the $\mathcal{B}_k$ and $\mathcal{R}_k$ |
| $\mu$ | 0.5 | 0.5 | 0.5 | The threshold of the proportion of successfully refined samples |

### A.6. Privacy Preservation Statement

The transmitted information of FedGR is similar to FedAvg. The additional transmitted information in FedGR only contains two scalar values per local example: the running mean of its cross-entropy loss, together with an opaque identifier,

---

**Algorithm 1** Algorithm of FedGR

---

**Input:** Total round $R$
**Output:** $\mathbf{w}_g$
 1: **for** round $= 0 : R - 1$ **do**
 2:     Sample a set of clients $S(t)$;
 3:     **for** $k \in S(t)$ **do**
 4:         $\left(n_k, \mathbf{w}_k^t, \{(d_{i,k}, \bar{\ell}_i^t)\}_i^{n_k}\right) \leftarrow \text{LocalTraining}\left(\mathbf{w}_g^{t-1}, r_k, \{d_{i,k}\} \text{ of } \hat{\mathcal{D}}_k^c \text{ and } \hat{\mathcal{D}}_k^n\right)$;
 5:     **end for**
 6:     Perform *Central Sieving* to obtain $r_k$ and the $\{d_{i,k}\}$ of $\hat{\mathcal{D}}_k^c$ and $\hat{\mathcal{D}}_k^n$;
 7:     $\mathbf{w}_g^t \leftarrow \sum_{k \in S(t)} \frac{n_k}{\sum_{k \in S(t)} n_k} \mathbf{w}_k^t$;                                                 *// FedAvg Aggregation*
 8: **end for**
 9: $\mathbf{w}_g \leftarrow \mathbf{w}_g^R$
10: **Return** $\mathbf{w}_g$

---

**Algorithm 2** LocalTraining of FedGR

---

**Input:** $\mathbf{w}_g^{t-1}, r_k, \{d_{i,k}\}$ of $\hat{\mathcal{D}}_k^c$ and $\hat{\mathcal{D}}_k^n$, $\lambda_{\mathcal{B}}, \lambda_{\mathcal{R}}, \alpha, t, \beta, \mu, \tau, \epsilon, \gamma_g, \gamma_l$, and local training epochs $E$
**Output:** $n_k, \mathbf{w}_k^t, \mathbf{w}_{k,ema}^t$, and $\{(d_{i,k}, \bar{\ell}_i)\}_i^{n_k}$
 1: **Before local training:**
 2: Receive $\mathbf{w}_g^{t-1}$, estimated noise ratio $r_k$, and the $\{d_{i,k}\}$ of $\hat{\mathcal{D}}_k^c$ and $\hat{\mathcal{D}}_k^n$ from the server;
 3: Initialize the local model with $\mathbf{w}_g^{t-1}$;
 4: Forward the model $(h \circ f)\left(\mathbf{x}_i; \mathbf{w}_g^{t-1}\right)$ on $\tilde{\mathcal{D}}_k$ to perform *Label Refining* by Eq.11 in main paper, update $L_i^t$ by Eq.9 in main paper, get representations $\left\{ f\left(\mathbf{x}_i^w; \mathbf{w}_{g,f}^{t-1}\right) \right\}_i^{n_k}$, and calculate the averaged loss $\{\bar{\ell}_i^t\}_i^{n_k}$ by Eq.10 in main paper;
 5: *Globally Revise* local EMA model by Eq.12 in main paper and get the local EMA logits $\{\mathbf{p}_i^{le,w}\}_i^{n_k}$ by Eq.14 in main paper;
 6: **Local training:**
 7: **for** epoch $= 0 : E - 1$ **do**
 8:     **if** $t <= \alpha$ **then**
 9:         $\mathcal{L}_k^{SR} = \mathbb{E}_{\hat{\mathcal{D}}_k}\left[\mathcal{H}\left(\mathbf{p}_i^{l,s}, \hat{y}_i\right)\right]$;
10:     **else**
11:         $\mathcal{L}_k^{SR} = \mathbb{E}_{\tilde{\mathcal{D}}_k}\left[\mathcal{H}\left(\mathbf{p}_i^{l,s}, \hat{y}_i\right)\right]$;
12:     **end if**
13:                                                                     *// Sieving-and-Refining*
14:     $\mathcal{B}_k = \mathbb{E}_{\hat{\mathcal{D}}_k}\left[KL\left(\frac{\mathbf{p}_i^{le,w}}{\tau}, \frac{\mathbf{p}_i^{l,s}}{\tau}\right)\right]$;               *// Global Revised EMA Distillation*
15:     $\mathcal{R}_k = \mathbb{E}_{\hat{\mathcal{D}}_k}\left[KL\left(\frac{f\left(\mathbf{x}_i^w; \mathbf{w}_{g,f}^{t-1}\right)}{\tau}, \frac{f\left(\mathbf{x}_i^s; \mathbf{w}_{k,f}^t\right)}{\tau}\right)\right]$;     *// Global Representation Regularization*
16:     $\mathcal{L}_k = \mathcal{L}_k^{SR} + \lambda_{\mathcal{B}}\mathcal{B}_k + \lambda_{\mathcal{R}}\mathcal{R}_k$;
17:     $\mathbf{w}_k^t \leftarrow \arg\min_{\mathbf{w}_k^t} \mathcal{L}_k$;                                              *// SGD Optimization*
18:     $\mathbf{w}_{k,ema}^t \leftarrow$ EMA update by Eq.12 in main paper;
19: **end for**
20: **Return** $n_k, \mathbf{w}_k^t$, and $\left\{(d_{i,k}, \bar{\ell}_i^t)\right\}_i^{n_k}$ to server;

---

*i.e.*, $\{(d_{i,k}, \bar{\ell}_i^t)\}_{i=1}^{n_k}$. The cross-entropy loss function is many-to-one, so even a server that knows the current model cannot uniquely infer either the raw input or the label from the loss value alone. Practically, existing "deep leakage from gradients" (Zhu et al.) attacks rely on access to full gradients or parameter deltas, not on single scalar losses. Consequently, FedGR maintains the same privacy-preserving properties as FedAvg.

### A.7. Convergence Analysis

FedGR performs standard FedAvg (McMahan et al., 2017) aggregation and uses gradients of Eq. 4. Hence, the only algorithmic difference with FedAvg is that each stochastic gradient is taken on a different but smooth objective, and the convergence of FedGR is similar to that of FedAvg. For clarity we first restate the optimisation problem induced by FedGR, list the additional assumptions that its extra losses require, and then adapt the classical FedAvg proof to this augmented objective.

**Problem Statement.** We analyze FedGR under the classical *client-averaging* update

$$w_{t+1} = \sum_{k \in \mathcal{S}(t)} a_k \, w_{k,t}^{(E)},$$

where each selected client performs $E$ local SGD steps (epochs) with step-size $\eta$ on its **FedGR objective**

$$G_k(w) := F_k(w) + \lambda_B \, B_k(w) + \lambda_R \, R_k(w).$$

Below we prove that FedGR enjoys the same non-convex convergence rate $\mathcal{O}(1/\sqrt{T})$ as FedAvg.

**Assumption A.1** (Smoothness). Each $G_k$ is $L$-smooth: $\|\nabla G_k(u) - \nabla G_k(v)\| \le L \|u - v\|$, $\forall u, v$.

**Assumption A.2** (Bounded stochastic variance). For any stochastic gradient $g_{k,t}^{(m)}$ computed on client $k$ at local step $m$ of round $t$,

$$
\begin{aligned}
\mathbb{E}\big[g_{k,t}^{(m)} \mid w_{k,t}^{(m)}\big] &= \nabla G_k\big(w_{k,t}^{(m)}\big), \\
\mathbb{E}\big\|g_{k,t}^{(m)} - \nabla G_k(w_{k,t}^{(m)})\big\|^2 &\le \sigma^2.
\end{aligned}
\tag{18}
$$

**Assumption A.3** (Client heterogeneity). Let $G(w) = \sum_k a_k G_k(w)$. Then $\mathbb{E}_k \|\nabla G_k(w) - \nabla G(w)\|^2 \le \zeta^2$, $\forall w$.

**Assumption A.4** (Regularizer gradients). $\|\nabla B_k(w)\| \le M_B$, $\quad \|\nabla R_k(w)\| \le M_R$, $\forall k, w$.

Assumption A.4 implies $G_k$ is still $L$-smooth for some $L$ that absorbs $\lambda_B M_B$ and $\lambda_R M_R$.

**Lemma A.5** (One-round descent). *Under Assumptions A.1–A.4 and choosing $\eta \le \frac{1}{2LE}$,*

$$
\begin{aligned}
\mathbb{E}\big[G(w_{t+1})\big] &\le \mathbb{E}\big[G(w_t)\big] - \tfrac{\eta E}{2} \, \mathbb{E}\big\|\nabla G(w_t)\big\|^2 \\
&\quad + \eta^2 L^2 E^2 \big(\sigma^2 + \zeta^2\big).
\end{aligned}
\tag{19}
$$

*Proof sketch.* Apply $L$-smoothness of $G$, expand $w_{t+1} - w_t$, take expectation conditioning on $w_t$, and bound the second moment by $\sigma^2 + \zeta^2$ exactly as in FedAvg (cf. Reddi et al., 2021). Regulariser gradients are already contained in $\nabla G$. $\square$

**Main Result.**

**Theorem A.6** (FedGR convergence). *Let Assumptions A.1–A.4 hold and choose $\eta = \Theta\big(1/\sqrt{ET}\big)$. Then after $T$ communication rounds*

$$\frac{1}{T} \sum_{t=0}^{T-1} \mathbb{E}\big\|\nabla G(w_t)\big\|^2 \;=\; \mathcal{O}\Big(\frac{1}{\sqrt{ET}} + \frac{\sigma}{\sqrt{K}} + \eta L \zeta\Big).$$

*Hence FedGR attains the same $\mathcal{O}(1/\sqrt{T})$ rate as FedAvg for non-convex objectives.*

*Proof.* Sum the inequality of Lemma A.5 over $t = 0, \dots, T-1$, telescoping the left-hand side. Rearrange and plug in $\eta = \Theta(1/\sqrt{ET})$. $\square$

**Discussion.** The regularization weights $\lambda_B, \lambda_R$ only modify the smoothness constant $L$ and the bound on $\|\nabla G(w)\|$, affecting *constants* but not the asymptotic rate. Therefore, FedGR is as communication-efficient as FedAvg while providing superior robustness to noisy labels. □

## A.8. Complexity Analysis

As we discussed in related work, directly adopting the C-LNL methods to the local training of the FL would introduce additional computational overhead. Here, we provide an intuitive computational analysis, shown in Table 18. In this comparison, the communication cost is split into two parts, "server → client" and "client → server", while the computation cost is split into three other parts, "forward", "backward", and "other cost". The "total cost" aggregates "forward" and "backward" costs, while "other cost" denotes the computation overhead induced by diverse label-noise-robust F-LNL methods.

*Table 18.* A qualitative analysis of communication and computational costs of each round. Specifically, $M$ represents the communication cost associated with model transmission and reception. Computational costs of each client are delineated as follows: $F$ denotes the cost of forward propagation, $B$ denotes the cost of backward propagation, $E$ denotes the number of local epochs per round, and $m$ denotes the times of data augmentations. The context of FL includes $K$ clients, $C$ classes, selected clients $\mathcal{S}(t)$ at round $t$, and a global dataset size of $D$. Beyond standard training and communication overhead, additional computational costs for label-noise robustness are quantified. These include the overhead incurred by the Gaussian Mixture Model (GMM) and a k-nearest neighbors (KNN)-like algorithm, denoted as GMM and KNN, respectively.

| Method | Communication | | | Computation of each client | | | |
|---|---|---|---|---|---|---|---|
| | Srv→Clt | Clt→Srv | Total Cost | Forward | Backward | Total Cost | Other Cost |
| FedAvg | $\mathbf{M}$ | $\mathbf{M}$ | $2\mathbf{M}$ | $E\mathbf{F}$ | $E\mathbf{B}$ | $E\mathbf{F} + E\mathbf{B}$ | - |
| FedProx | $\mathbf{M}$ | $\mathbf{M}$ | $2\mathbf{M}$ | $E\mathbf{F}$ | $E\mathbf{B}$ | $E\mathbf{F} + E\mathbf{B}$ | - |
| FL-Coteaching | $2\mathbf{M}$ | $2\mathbf{M}$ | $4\mathbf{M}$ | $2E\mathbf{F}$ | $2E\mathbf{B}$ | $2E\mathbf{F} + 2E\mathbf{B}$ | - |
| FL-DivideMix | $2\mathbf{M}$ | $2\mathbf{M}$ | $4\mathbf{M}$ | $(4m+2)E\mathbf{F}$ | $2E\mathbf{B}$ | $(4m+2)E\mathbf{F} + 2E\mathbf{B}$ | $2E\mathbf{GMM}$ |
| FedCorr | $\mathbf{M}+K$ | $\mathbf{M}$ | $K+2\mathbf{M}$ | $(E+1)\mathbf{F}$ | $E\mathbf{B}$ | $(E+1)\mathbf{F} + E\mathbf{B}$ | $(1+\mathcal{S}(t))\mathbf{GMM} + \mathcal{S}(t)\mathbf{KNN}$ |
| FedNoRo | $\mathbf{M}+C$ | $\mathbf{M}$ | $C+2\mathbf{M}$ | $(E+1)\mathbf{F}$ | $E\mathbf{B}$ | $(E+1)\mathbf{F} + E\mathbf{B}$ | $\mathbf{GMM}$ |
| FedFixer | $2\mathbf{M}$ | $2\mathbf{M}$ | $4\mathbf{M}$ | $2E\mathbf{F}$ | $2E\mathbf{B}$ | $2E\mathbf{F} + 2E\mathbf{B}$ | - |
| FedDiv | $\mathbf{M}+2$ | $\mathbf{M}+2$ | $\mathbf{M}+4$ | $(E+3)\mathbf{F}$ | $E\mathbf{B}$ | $(E+3)\mathbf{F} + E\mathbf{B}$ | $\mathcal{S}(t)\mathbf{GMM}$ |
| FedGR | $\mathbf{M}+D$ | $\mathbf{M}+D$ | $2D+2\mathbf{M}$ | $(E+3)\mathbf{F}$ | $E\mathbf{B}$ | $(E+3)\mathbf{F} + E\mathbf{B}$ | $\mathbf{GMM}$ |

**Discussions.** According to Table 18, both FL-Coteaching and FL-DivideMix, which are the typical C-LNL methods (Coteaching (Han et al., 2018) and DivideMix (Li et al., 2020a)) implemented under FL, increase the additional communication and computation overhead. Label-noise-robust FL methods, such as FedCorr (Xu et al., 2022), FedNoRo (Wu et al., 2023), FedFixer (Ji et al., 2024), and FedGR, still inevitably introduce extra communication and computation overhead. However, these extra costs are much less than the typical C-LNL methods implemented under FL. Specifically, the transmitted information and the number of forward and backward passes are much less. As for the proposed FedGR, although it slightly increases the communication burden due to $D > N$ and $D > C$ and requires two more forward passes compared with FedCorr (Xu et al., 2022) and FedNoRo (Wu et al., 2023), it achieves the best label-noise robustness among the mentioned baselines.

## A.9. Visualizations

### A.9.1. VISUALIZATIONS OF FL DATA PARTITION

In this section, we visualize the data partition used by the proposed FedGR in Figure 18 and 19, following (Li et al., 2022). It is worth noting that the previous study FedCorr (Xu et al., 2022) has proposed another data partition method for the Non.I.I.D. scenario and it is also adopted by (Ji et al., 2024; Li et al., 2024). Nevertheless, the Non.I.I.D. scenario used by FedGR is more practical and harder than that of FedCorr (Xu et al., 2022), according to the visualizations in Figure 19, 20, and 21.

A.9.2. VISUALIZATIONS OF FS PARTITION RESULTS

In this section, we visualize a case of the FS partition results of the proposed FedGR. As shown in Fig. 22, the FS can generally distinguish between noisy labels and clean data. In fact, the proposed FedGR does not require the GMM of FS to distinguish all four categories perfectly. It is sufficient that the GMM achieves a reasonably high F1 score when separating data; the misassigned samples are then handled by the LR (Label Refining), EMA distillation, and representation regularization modules, which are explicitly designed to reduce their adverse impact.

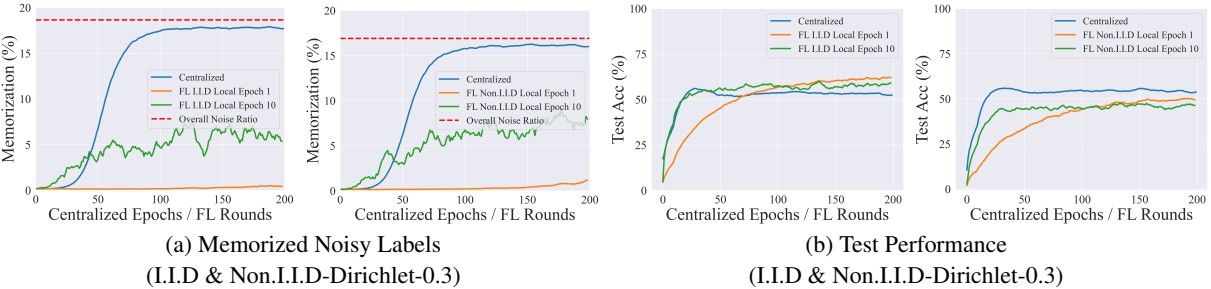

(a) Memorized Noisy Labels
(I.I.D & Non.I.I.D-Dirichlet-0.3)

(b) Test Performance
(I.I.D & Non.I.I.D-Dirichlet-0.3)

*Figure 5.* Memorization effect observation experimental results on CIFAR-10 with FL client sample ratio 0.2.

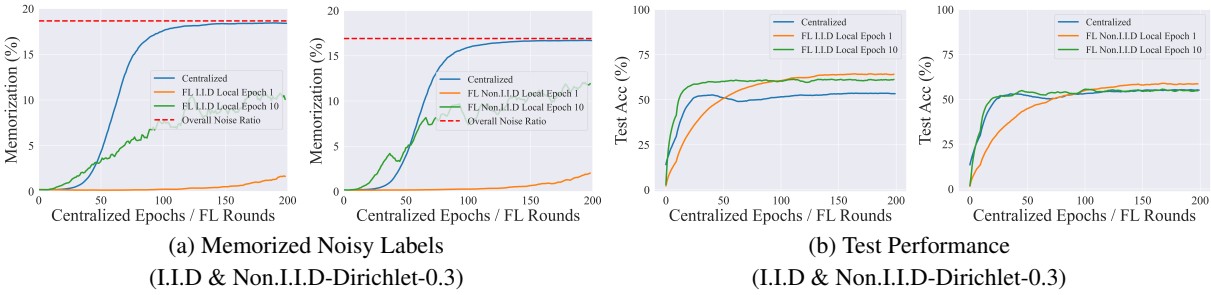

(a) Memorized Noisy Labels
(I.I.D & Non.I.I.D-Dirichlet-0.3)

(b) Test Performance
(I.I.D & Non.I.I.D-Dirichlet-0.3)

*Figure 6.* Memorization effect observation experimental results on CIFAR-10 with FL client sample ratio to 0.5.

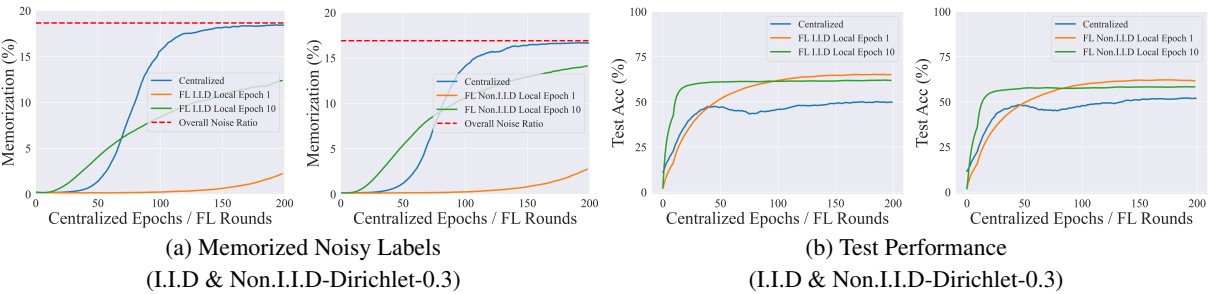

(a) Memorized Noisy Labels
(I.I.D & Non.I.I.D-Dirichlet-0.3)

(b) Test Performance
(I.I.D & Non.I.I.D-Dirichlet-0.3)

*Figure 7.* Memorization effect observation experimental results on CIFAR-10 with FL client sample ratio 1.0.

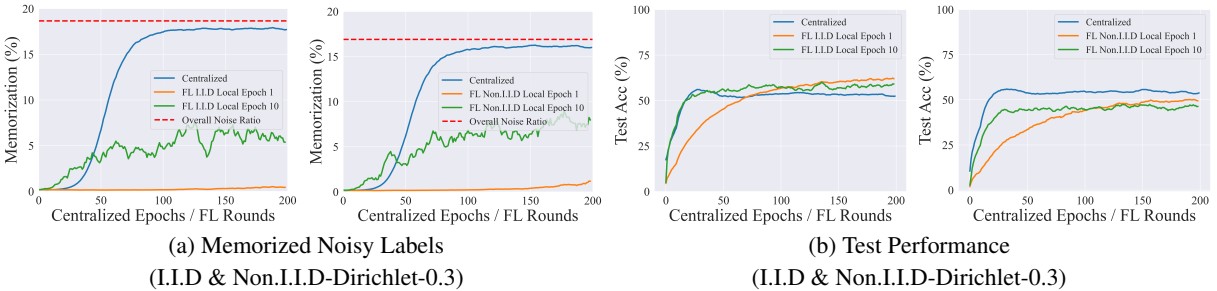

(a) Memorized Noisy Labels
(I.I.D & Non.I.I.D-Dirichlet-0.3)

(b) Test Performance
(I.I.D & Non.I.I.D-Dirichlet-0.3)

*Figure 8.* Memorization effect observation experimental results on CIFAR-100 with FL client sample ratio 0.2.

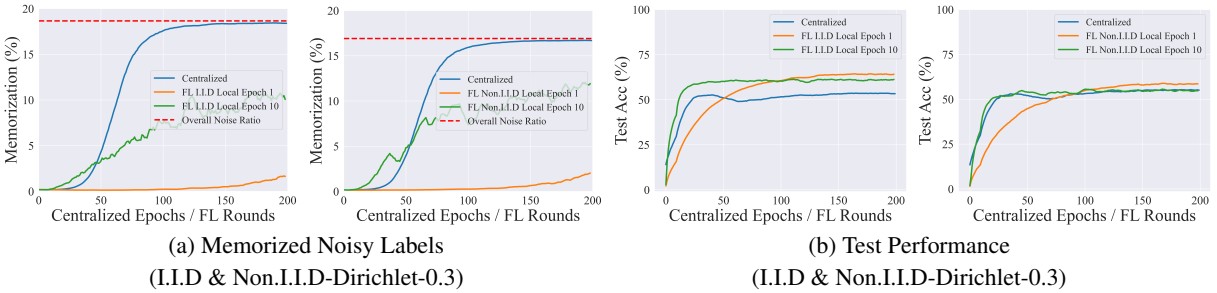

(a) Memorized Noisy Labels
(I.I.D & Non.I.I.D-Dirichlet-0.3)

(b) Test Performance
(I.I.D & Non.I.I.D-Dirichlet-0.3)

*Figure 9.* Memorization effect observation experimental results on CIFAR-100 with FL client sample ratio to 0.5.

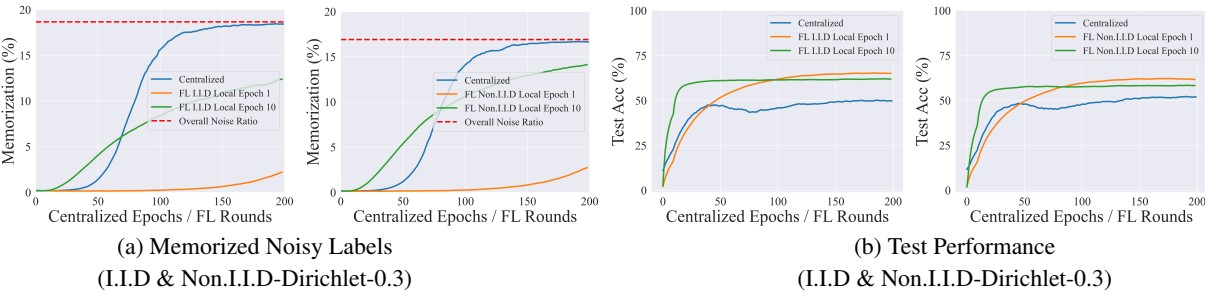

(a) Memorized Noisy Labels
(I.I.D & Non.I.I.D-Dirichlet-0.3)

(b) Test Performance
(I.I.D & Non.I.I.D-Dirichlet-0.3)

*Figure 10.* Memorization effect observation experimental results on CIFAR-100 with FL client sample ratio 1.0.

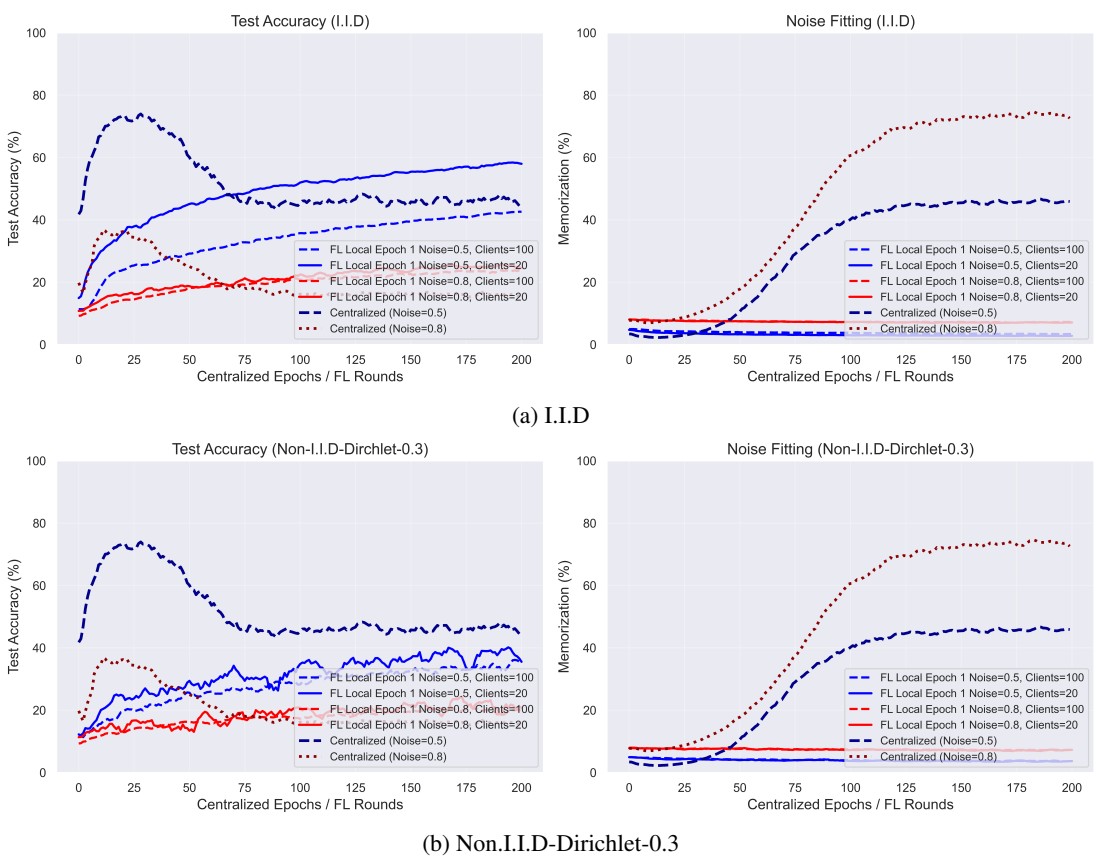

(a) I.I.D

(b) Non.I.I.D-Dirichlet-0.3

*Figure 11.* Observation of the memorization effect on CIFAR-10 under a federated learning client sampling ratio of 0.2 with different levels of label noise.

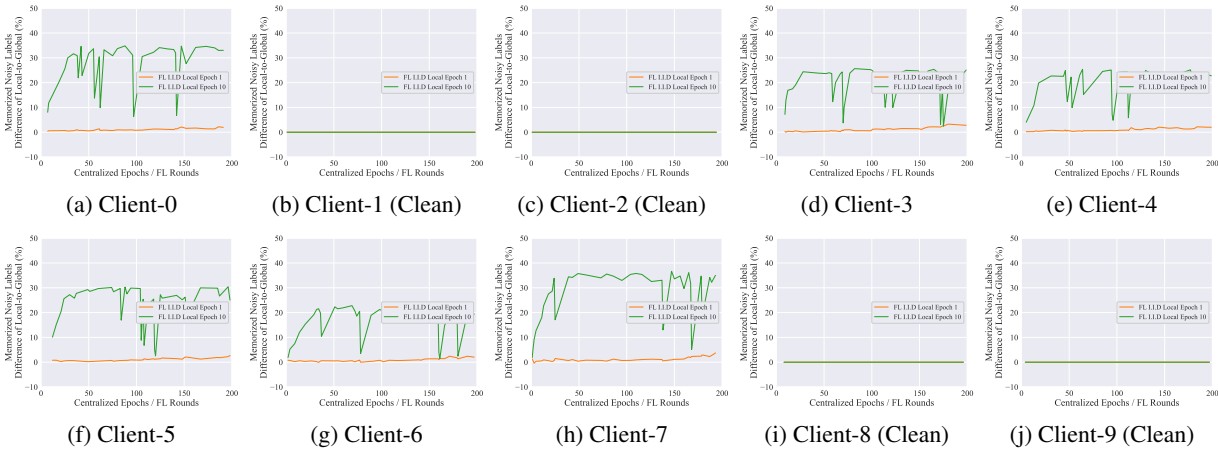

*Figure 12.* The difference between the overfitting degree of the client's local model on the client's noisy labels and that of the global model on the client's noisy labels. F-LNL setup: CIFAR-10, I.I.D, client sample ratio 0.2, Sym, $\phi = 0.6$, $\mathcal{U}(0.2, 0.4)$, and overall noise ratio=18.66%

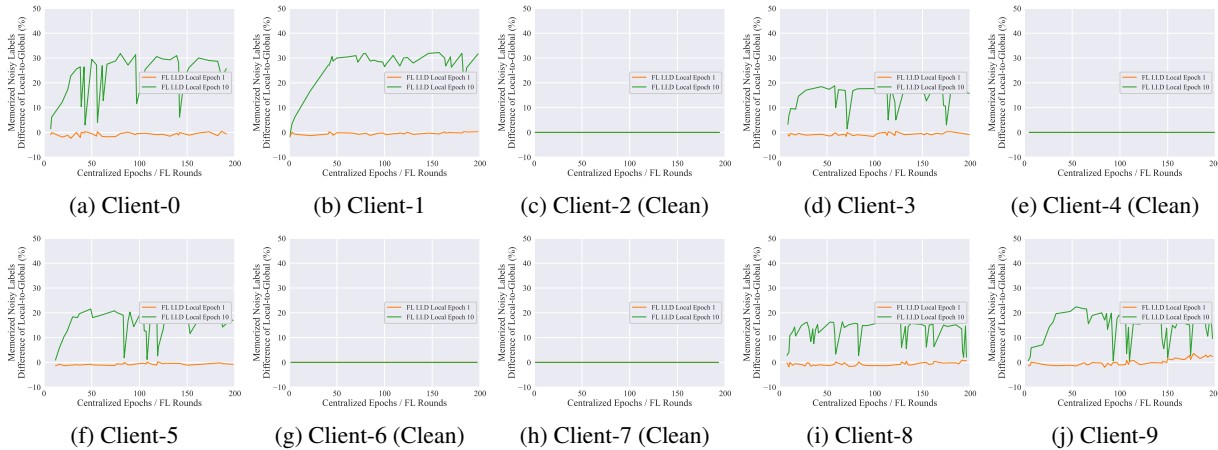

*Figure 13.* The difference between the overfitting degree of the client's local model on the client's noisy labels and that of the global model on the client's noisy labels. F-LNL setup: CIFAR-10, Non.I.I.D-Dirichlet (0.3), client sample ratio 0.2, Sym, $\phi = 0.6, \mathcal{U}(0.2, 0.4)$, and overall noise ratio=15.36%

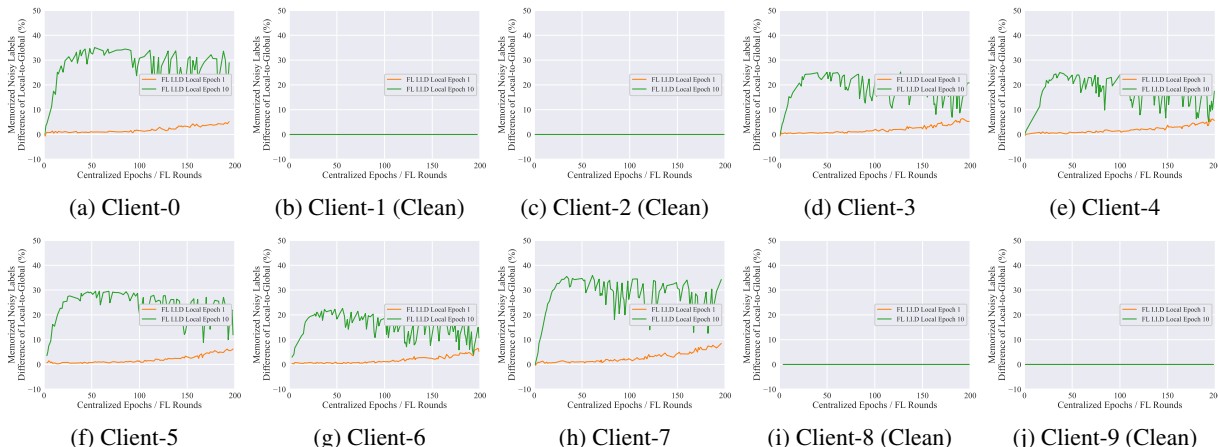

*Figure 14.* The difference between the overfitting degree of the client's local model on the client's noisy labels and that of the global model on the client's noisy labels. F-LNL setup: CIFAR-10, I.I.D, client sample ratio 0.5, Sym, $\phi = 0.6, \mathcal{U}(0.2, 0.4)$, and overall noise ratio=18.66%

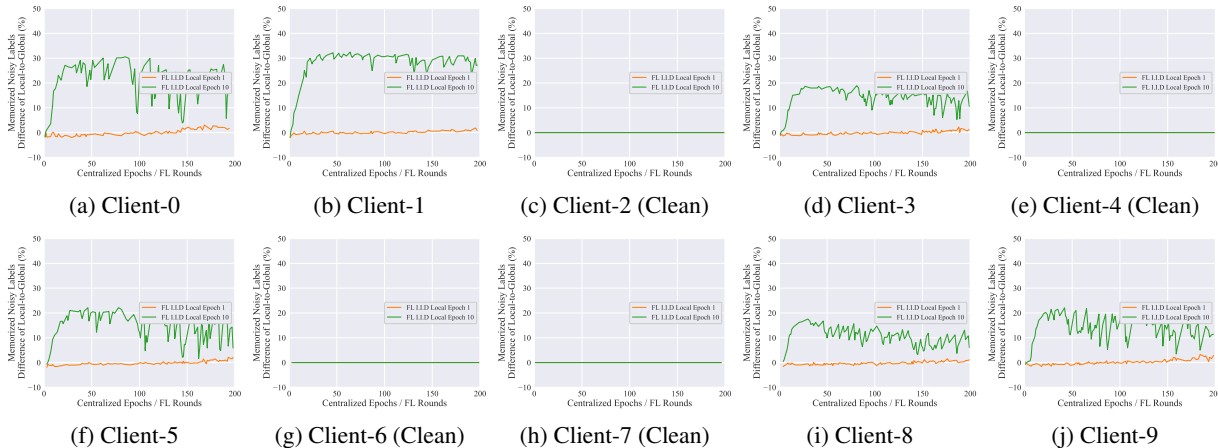

*Figure 15.* The difference between the overfitting degree of the client's local model on the client's noisy labels and that of the global model on the client's noisy labels. F-LNL setup: CIFAR-10, Non.I.I.D-Dirichlet (0.3), client sample ratio 0.5, Sym, $\phi = 0.6, \mathcal{U}(0.2, 0.4)$, and overall noise ratio=15.36%

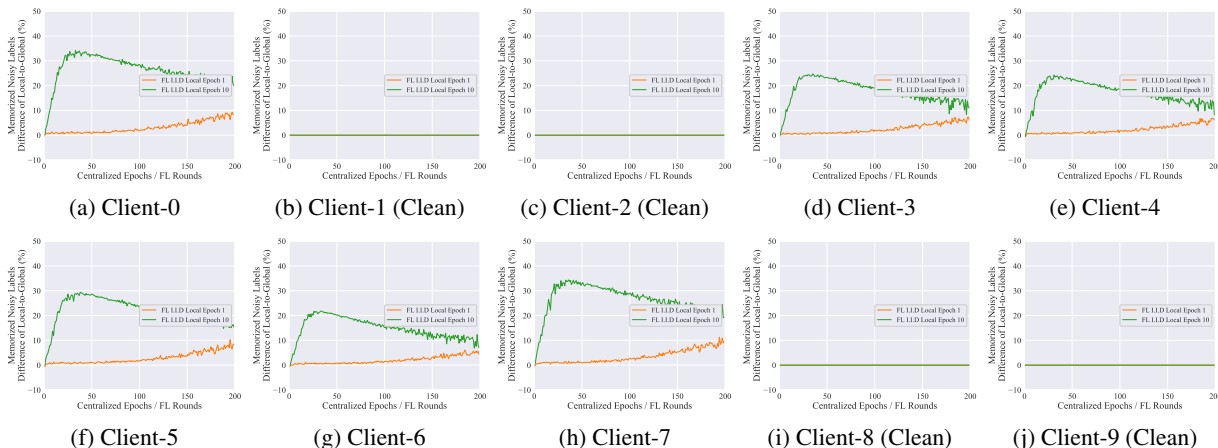

*Figure 16.* The difference between the overfitting degree of the client's local model on the client's noisy labels and that of the global model on the client's noisy labels. F-LNL setup: CIFAR-10, I.I.D, client sample ratio 1.0, Sym, $\phi = 0.6$, $\mathcal{U}(0.2, 0.4)$, and overall noise ratio=18.66%

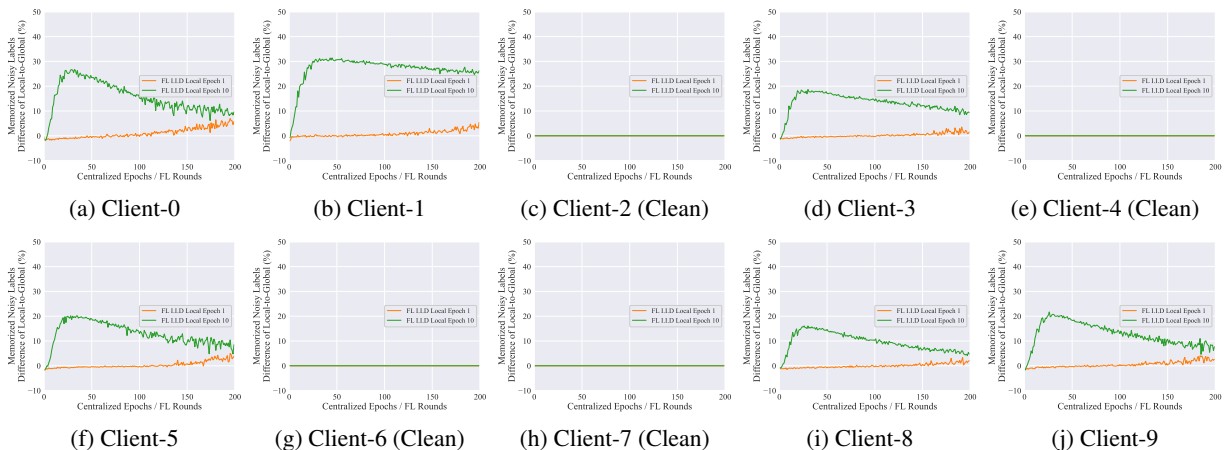

*Figure 17.* The difference between the overfitting degree of the client's local model on the client's noisy labels and that of the global model on the client's noisy labels. F-LNL setup: CIFAR-10, Non.I.I.D-Dirichlet (0.3), client sample ratio 1.0, Sym, $\phi = 0.6$, $\mathcal{U}(0.2, 0.4)$, and overall noise ratio=15.36%

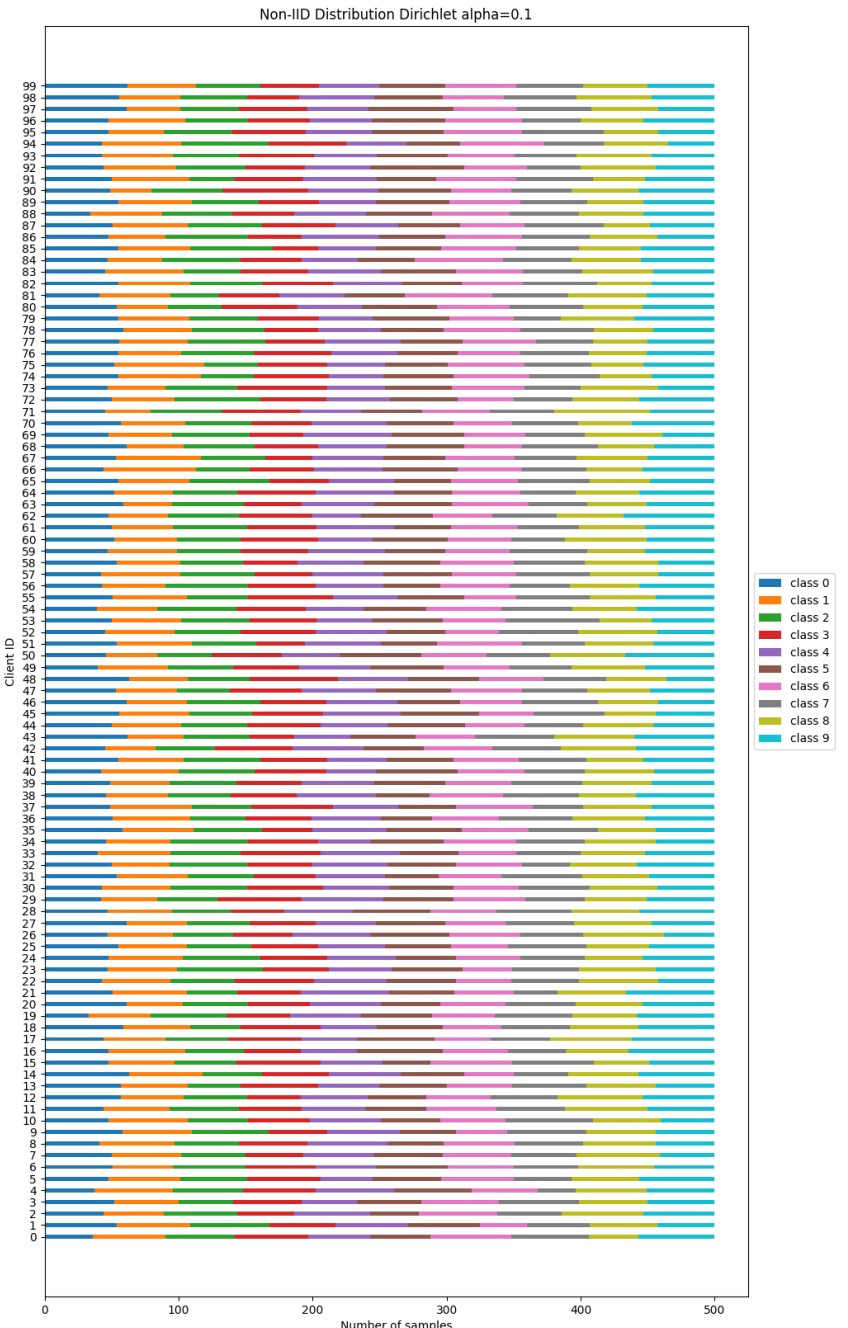

*Figure 18.* Visualization of the I.I.D data partition for FedGR.

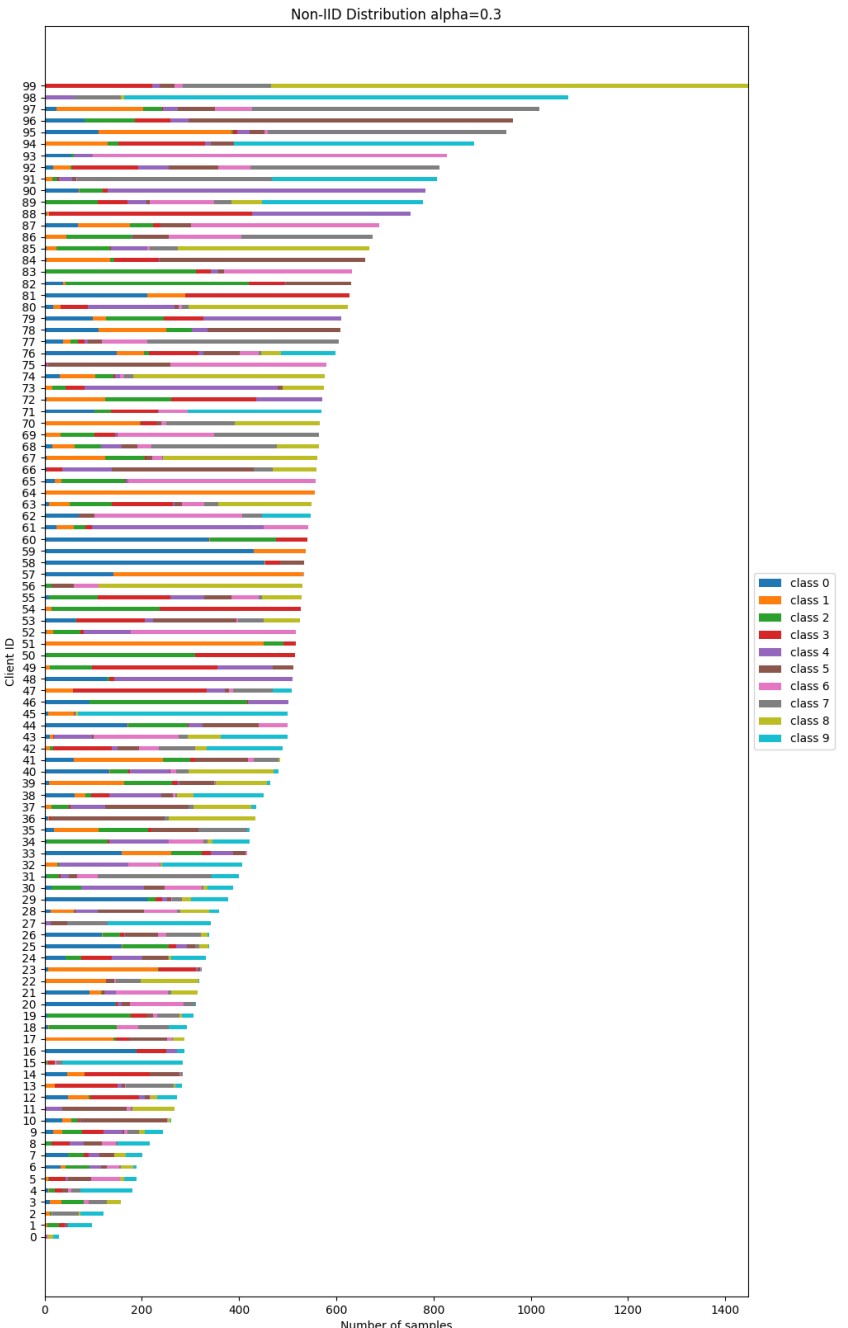

*Figure 19.* Visualization of the Non.I.I.D Dirichlet data partition for FedGR: $\alpha_{dirichlet} = 0.3$.

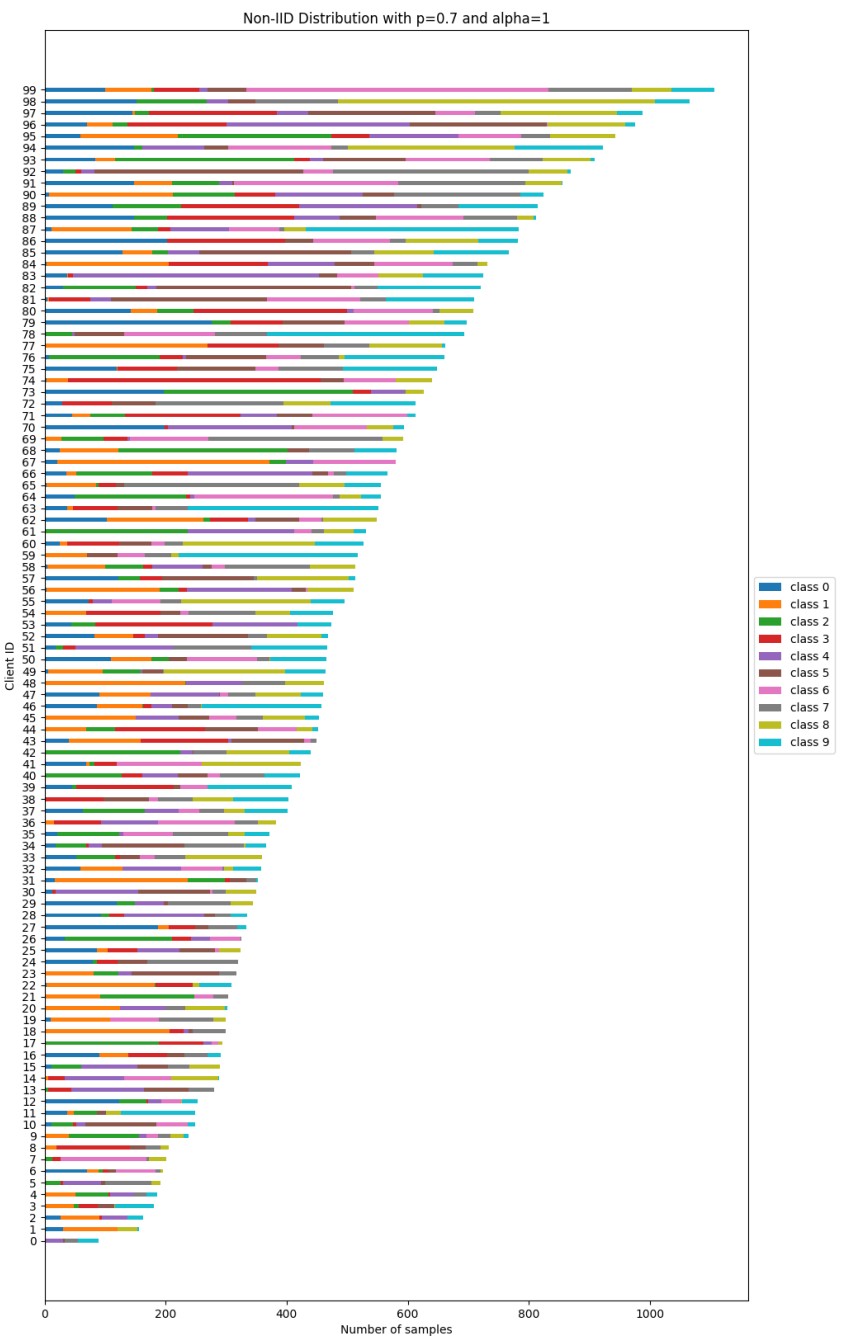

*Figure 20.* Visualization of the Non.I.I.D data partition used by FedCorr: $p = 0.7$ and $\alpha_{Dir} = 1$.

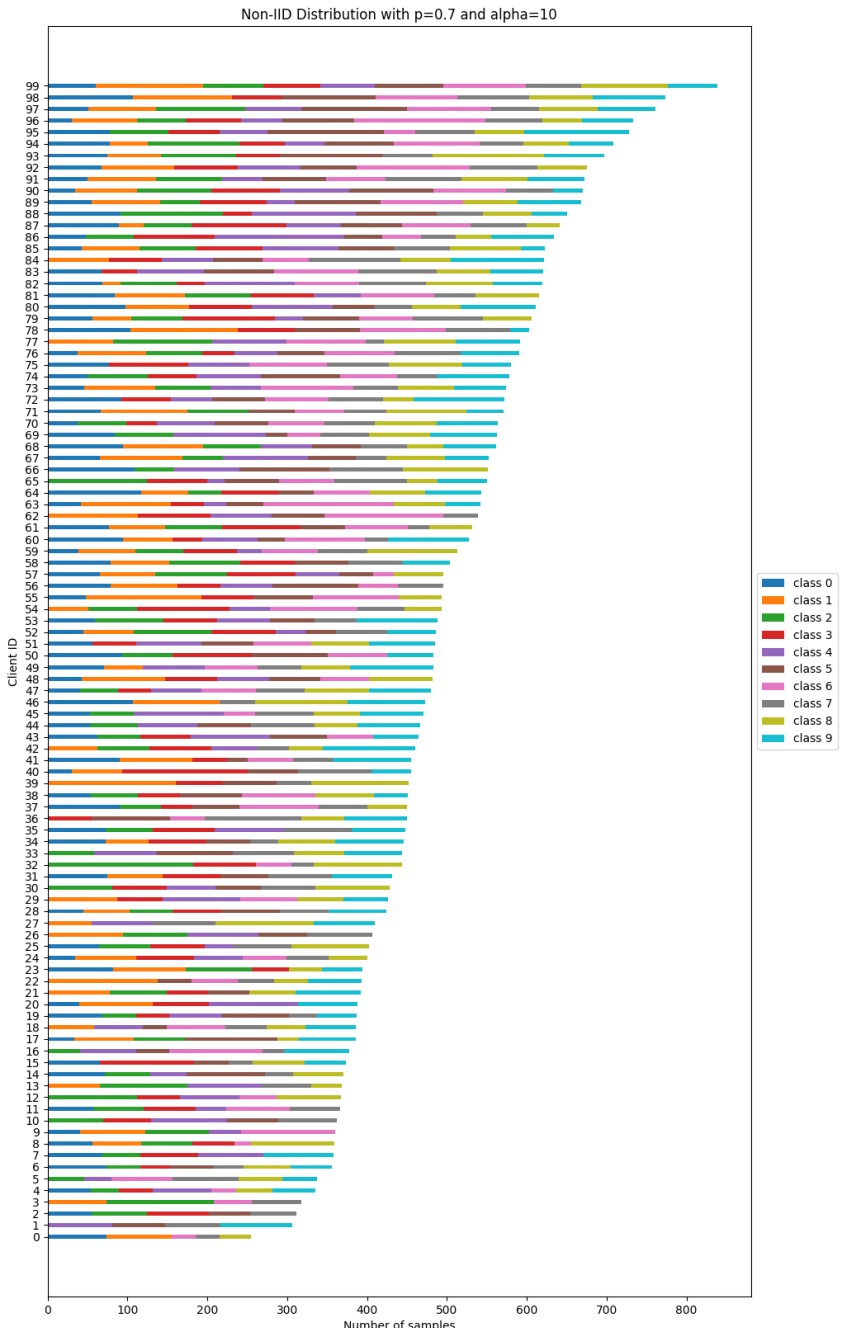

*Figure 21.* Visualization of the Non.I.I.D data partition used by FedCorr: $p = 0.7$ and $\alpha_{Dir} = 10$.

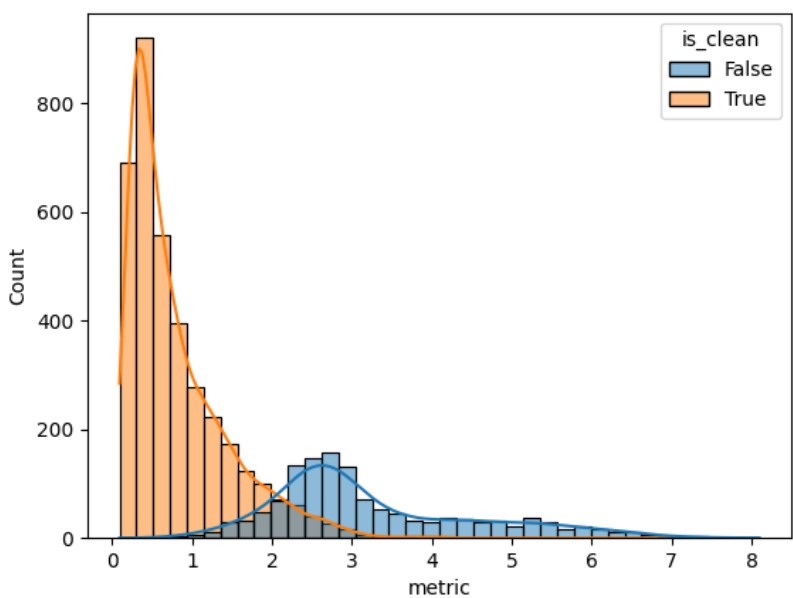

*Figure 22.* Visualization of the FS partition results of FedGR. The F-LNL setup: CIFAR-10, Mixed, $\phi = 1.0$, and $\mathcal{U}(0.2, 0.4)$.

