# Learning Locally, Revising Globally:
# Global Reviser for Federated Learning with Noisy Labels

## Abstract

Convantioanl federated learning (FL) heavily depends on high-quality labels, which are often impractical in the real world, leading to the federated label-noise (F-LN) problem. Worsely, the F-LN problem is exacerbated by the heterogeneity of FL, whereas clients experience different label-noise types, ratios, and data distribution. In this study, we first observe an intriguing phenomenon that the global model of FL exhibits a slow memorization of noisy labels, suggesting its ability to maintain reliable predictions and robust representations in FL. Motivated on this, we propose a novel method termed Federated Global Reviser (FedGR), a straightforward yet effective method comprising three modules that collaboratively rectify noisy labels and regularize local training. By exploiting above inherent property, FedGR improve the label-noise robustness of FL in a self-contained manner. Extensive experiments on three widely used F-LN benchmarks demonstrate the superior performance of FedGR, consistently outperforming seven state-of-the-art baselines even in severe label-noise and data heterogeneity. Code will be released upon acceptance.

## 1. Introduction

Federated learning (FL) facilitates privacy-preserving collaborative training across clients for applications like healthcare (Kaissis et al., 2020) and recommendation systems (Sun et al., 2024). Despite promising performance (McMahan et al., 2017; Li et al., 2020b; Meng et al., 2024), FL heavily relies on high-quality annotated data. However, precisely annotating decentralized datasets is impractical (Irvin et al., 2019), inevitably leading to the federated label noise (F-LN)

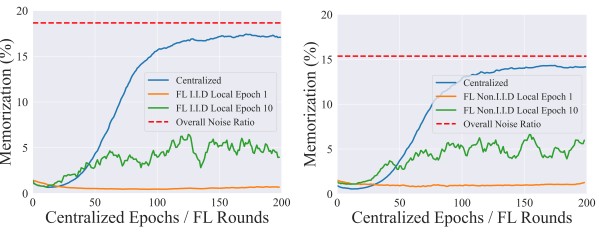

(a) Memorization (I.I.D & Non.I.I.D-Dirichlet-0.3)

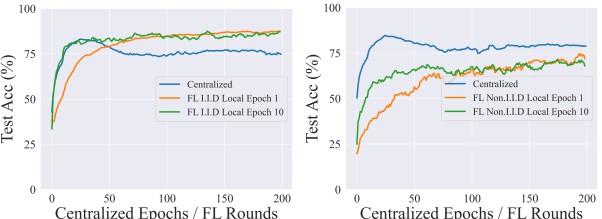

(b) Test Acc (I.I.D & Non.I.I.D-Dirichlet-0.3)

*Figure 1.* **(a) Slower Memorization Effect**: On CIFAR-10, the global FL model memorizes $\leq 30\%$ of noisy labels, while significantly lower than that of centralized training. **(b) Preservation of Test Performance**: The global model in FL avoids the test performance degradation typically observed in centralized training under noisy labels. Please see `appendix` for more results and discussions, which indicates that such a phenomenon is non-trivial.

problem (Yang et al., 2022; Xu et al., 2022). Unlike centralized label-noise (C-LN) problem (Han et al., 2018; Li et al., 2020a), F-LN is more challenging due to the label-noise and data heterogeneity, encompassing diverse noise patterns (*e.g.*, varying ratios/types) and data heterogeneity causing class imbalance (Li et al., 2022; Qi et al., 2023; Wu et al., 2023; Qi et al., 2025). This heterogeneity significantly hinders the direct application of centralized learning with noisy labels (C-LNL) methods (Li et al., 2022; Wei et al., 2021). Thus, it is highly expected to customize a federated learning with noisy labels (F-LNL) approach to tackle the F-LN problem.

Existing F-LNL approaches typically treat the F-LN problem as a distributed extension of learning with noisy labels, focusing on refining client-side training algorithms (Jiang et al., 2022; Wang et al., 2022; Xu et al., 2022; Ji et al., 2024; Kim et al.) or detecting and isolating noisy clients (Xu et al., 2022; Lu et al., 2024) to mitigate negative impacts. How-

[1]Anonymous Institution, Anonymous City, Anonymous Region, Anonymous Country. Correspondence to: Anonymous Author <anon.email@domain.com>.

Preliminary work. Under review by the International Conference on Machine Learning (ICML). Do not distribute.

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

| FedProx | **M** | **M** | 2**M** | $E$**F** | $E$**B** | $E$**F** + $E$**B** | - |
| FL-Coteaching | 2**M** | 2**M** | 4**M** | $2E$**F** | $2E$**B** | $2E$**F** + $2E$**B** | - |
| FL-DivideMix | 2**M** | 2**M** | 4**M** | $(4m+2)E$**F** | $2E$**B** | $(4m+2)E$**F** + $2E$**B** | $2E$**GMM** |
| FedCorr | **M** + $K$ | **M** | $K$ + 2**M** | $(E+1)$**F** | $E$**B** | $(E+1)$**F** + $E$**B** | $(1+\mathcal{S}(t))$**GMM** + $\mathcal{S}(t)$**KNN** |
| FedNoRo | **M** + $C$ | **M** | $C$ + 2**M** | $(E+1)$**F** | $E$**B** | $(E+1)$**F** + $E$**B** | **GMM** |
| FedFixer | 2**M** | 2**M** | 4**M** | $2E$**F** | $2E$**B** | $2E$**F** + $2E$**B** | - |
| FedDiv | **M** + 2 | **M** + 2 | **M** + 4 | $(E+3)$**F** | $E$**B** | $(E+3)$**F** + $E$**B** | $\mathcal{S}(t)$**GMM** |
| FedGR | **M** + $D$ | **M** + $D$ | $2D$ + 2**M** | $(E+3)$**F** | $E$**B** | $(E+3)$**F** + $E$**B** | **GMM** |

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

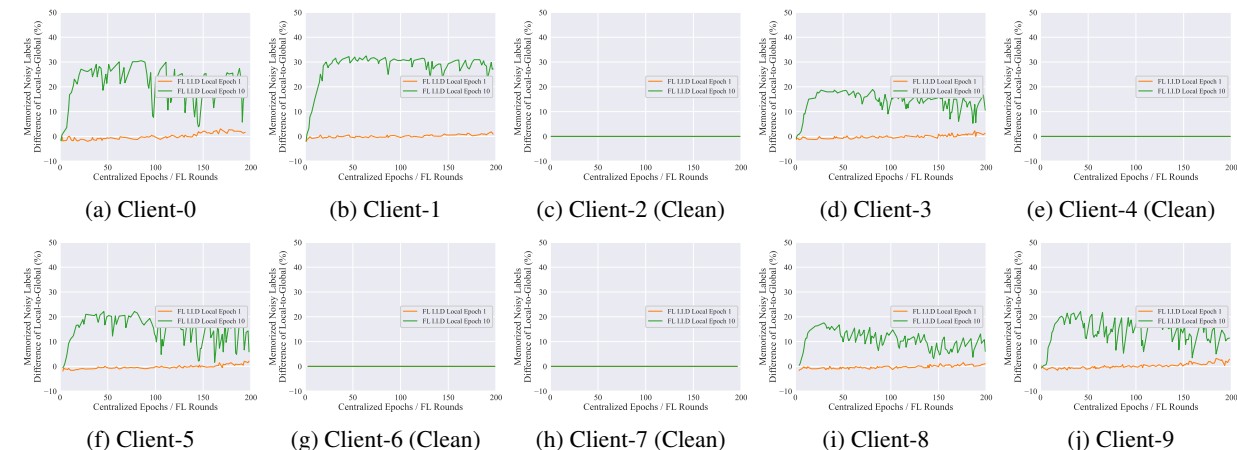

(a) Client-0    (b) Client-1    (c) Client-2 (Clean)    (d) Client-3    (e) Client-4 (Clean)

(f) Client-5    (g) Client-6 (Clean)    (h) Client-7 (Clean)    (i) Client-8    (j) Client-9

*Figure 15.* The difference between the overfitting degree of the client's local model on the client's noisy labels and that of the global model on the client's noisy labels. F-LNL setup: CIFAR-10, Non.I.I.D-Dirichlet (0.3), client sample ratio 0.5, Sym, $\phi = 0.6$, $\mathcal{U}(0.2, 0.4)$, and overall noise ratio=15.36%

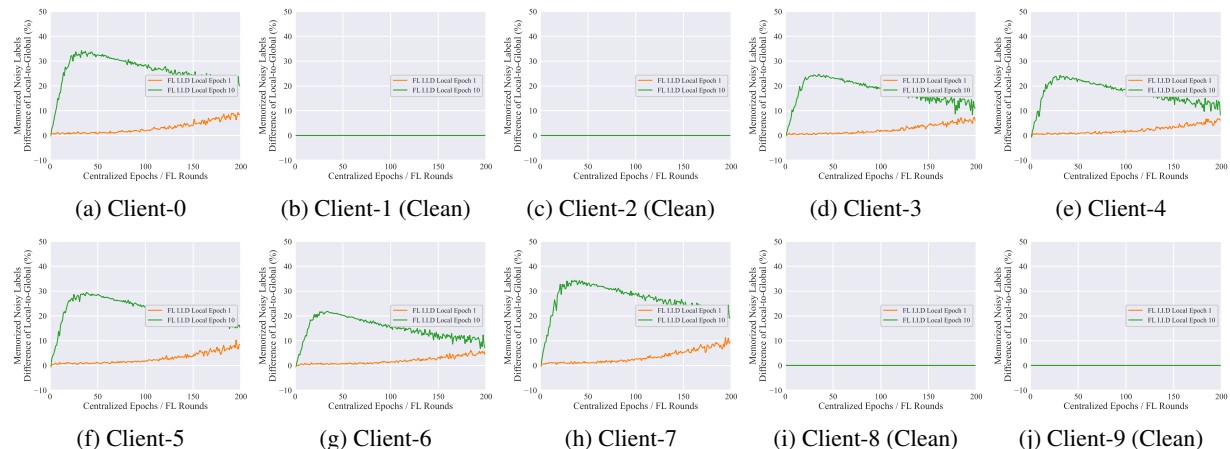

(a) Client-0    (b) Client-1 (Clean)    (c) Client-2 (Clean)    (d) Client-3    (e) Client-4

(f) Client-5    (g) Client-6    (h) Client-7    (i) Client-8 (Clean)    (j) Client-9 (Clean)

*Figure 16.* The difference between the overfitting degree of the client's local model on the client's noisy labels and that of the global model on the client's noisy labels. F-LNL setup: CIFAR-10, I.I.D, client sample ratio 1.0, Sym, $\phi = 0.6$, $\mathcal{U}(0.2, 0.4)$, and overall noise ratio=18.66%

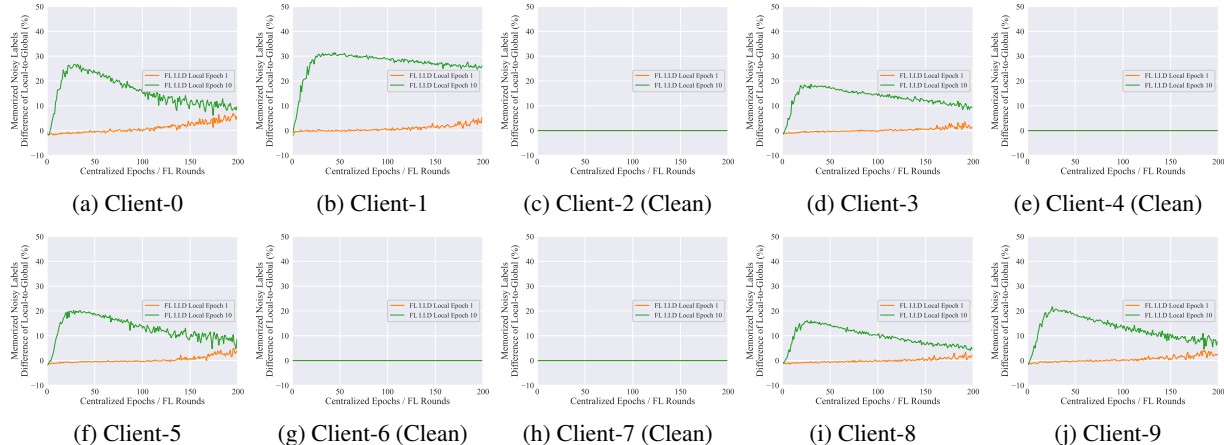

(a) Client-0    (b) Client-1    (c) Client-2 (Clean)    (d) Client-3    (e) Client-4 (Clean)

(f) Client-5    (g) Client-6 (Clean)    (h) Client-7 (Clean)    (i) Client-8    (j) Client-9

*Figure 17.* The difference between the overfitting degree of the client's local model on the client's noisy labels and that of the global model on the client's noisy labels. F-LNL setup: CIFAR-10, Non.I.I.D-Dirichlet (0.3), client sample ratio 1.0, Sym, $\phi = 0.6$, $\mathcal{U}(0.2, 0.4)$, and overall noise ratio=15.36%

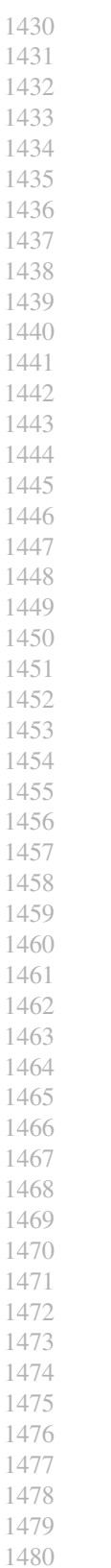
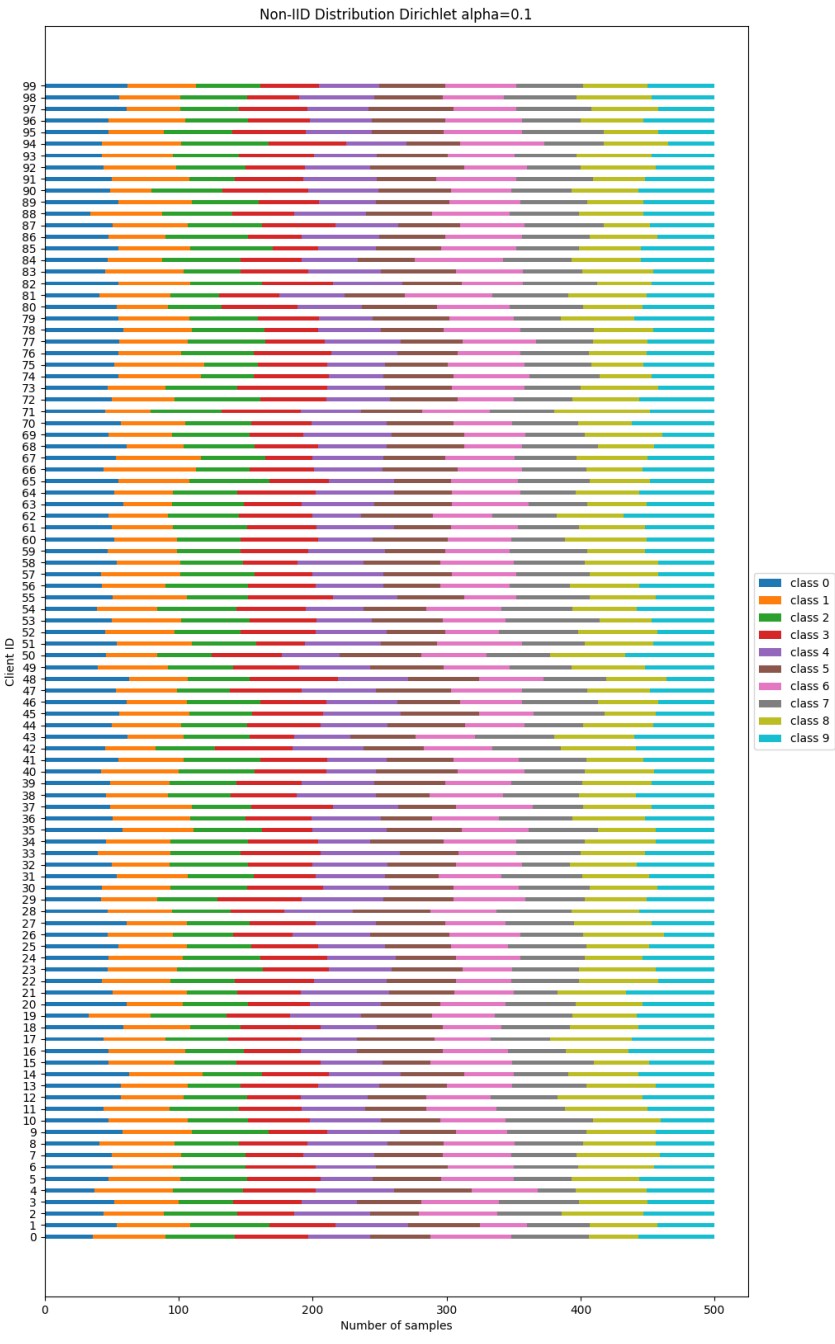

*Figure 18.* Visualization of the I.I.D data partition for FedGR.

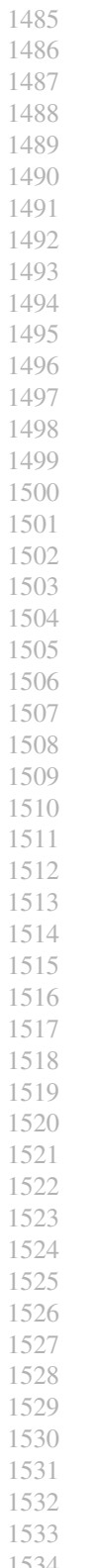
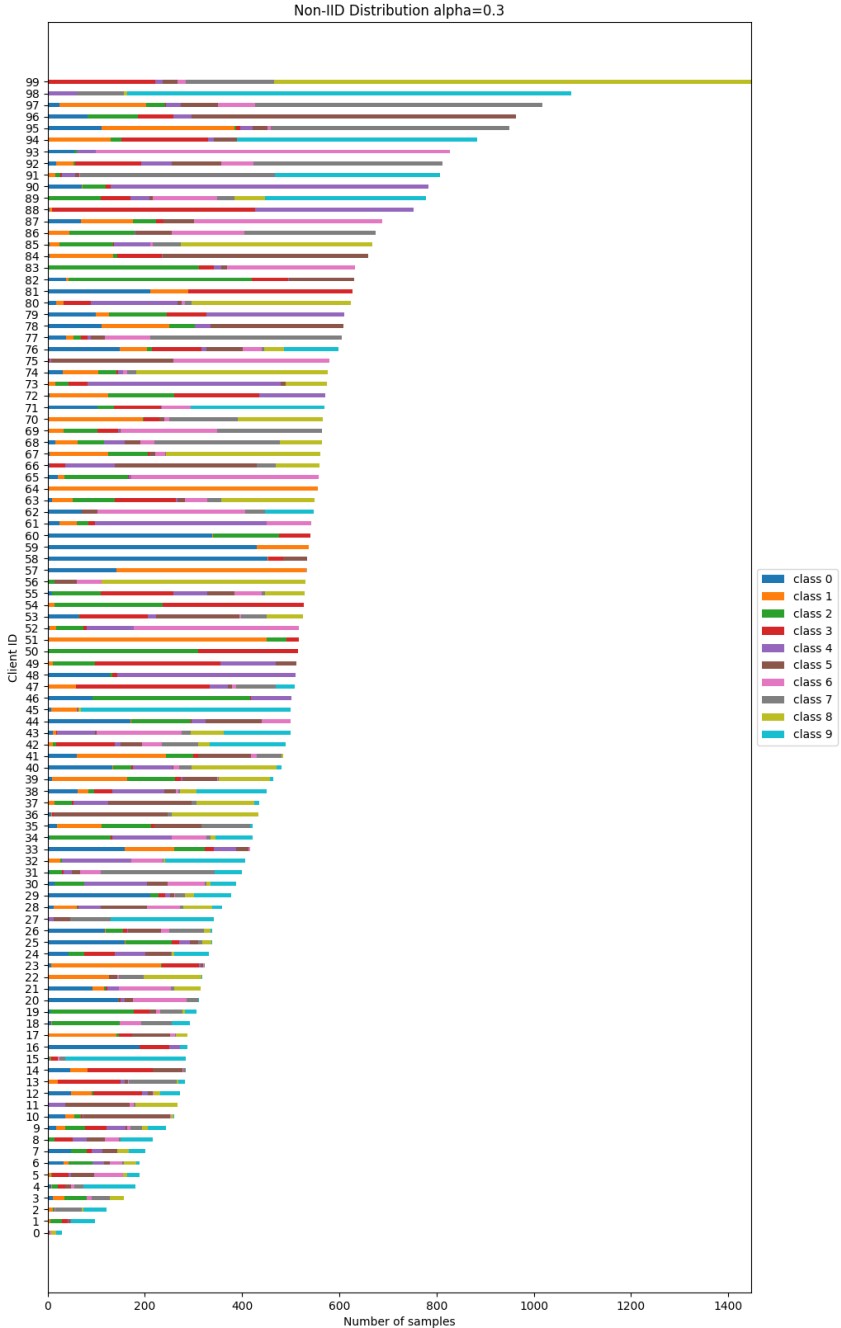

*Figure 19.* Visualization of the Non.I.I.D Dirichlet data partition for FedGR: $\alpha_{dirichlet} = 0.3$.

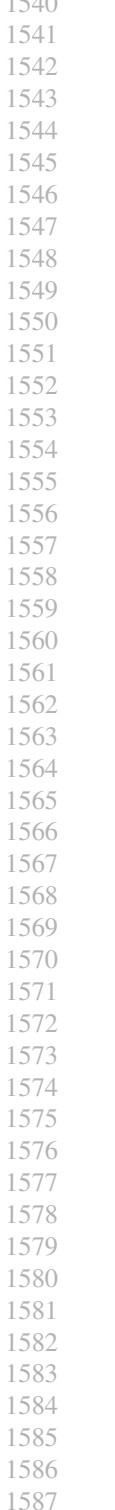
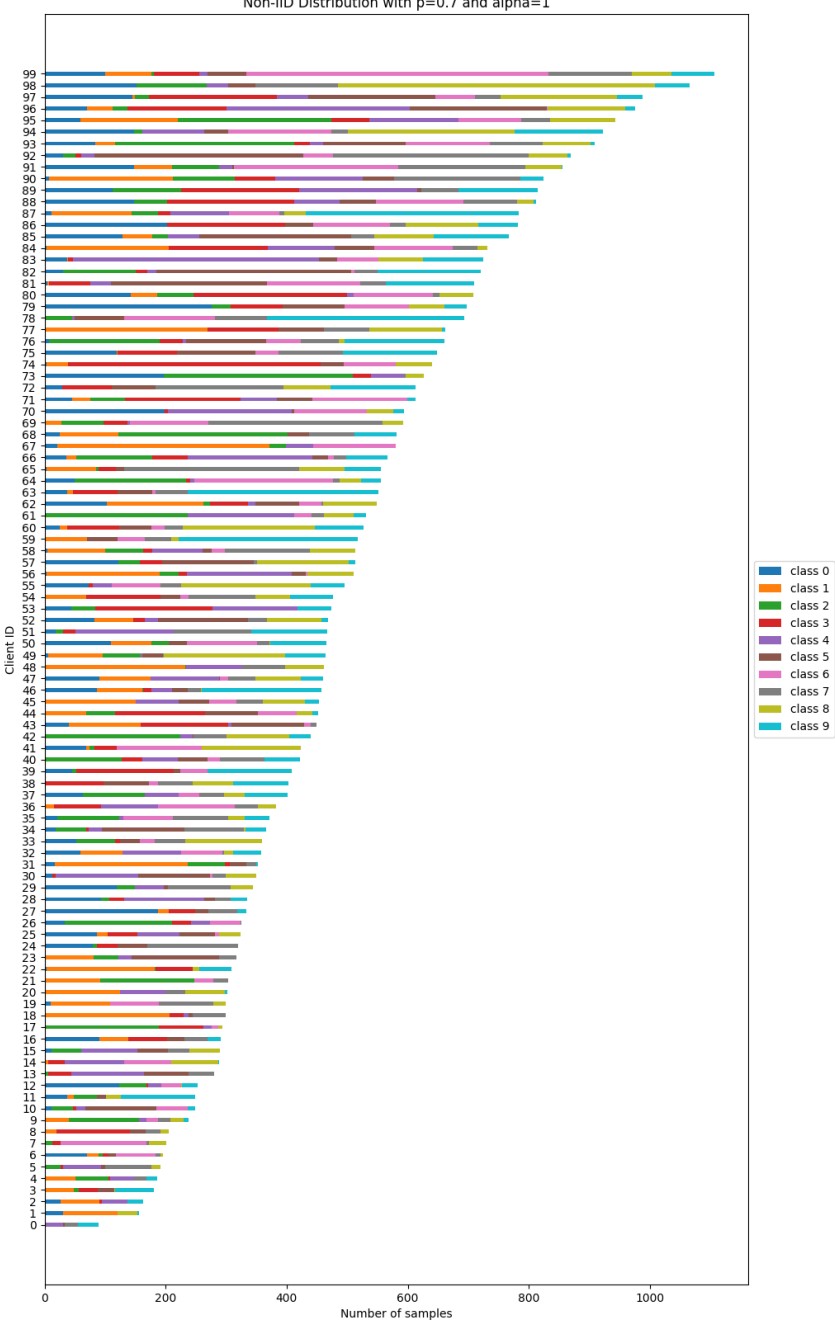

*Figure 20.* Visualization of the Non.I.I.D data partition used by FedCorr: $p = 0.7$ and $\alpha_{Dir} = 1$.

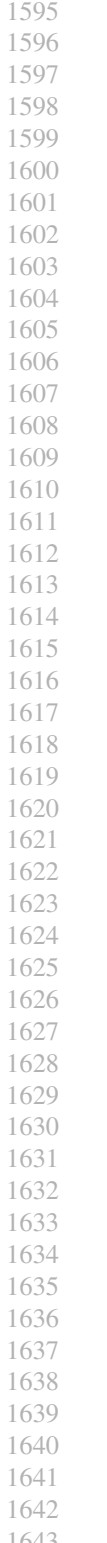
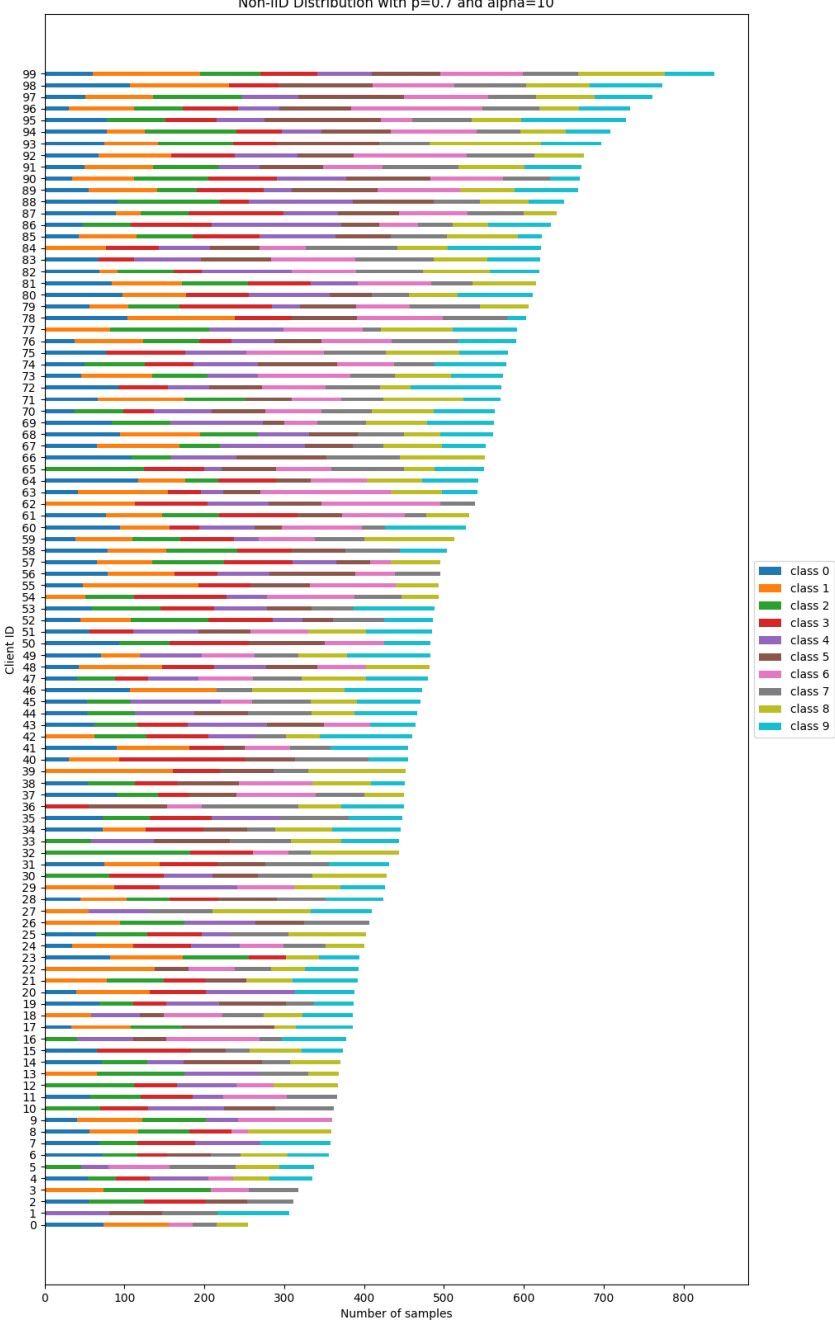

*Figure 21.* Visualization of the Non.I.I.D data partition used by FedCorr: $p = 0.7$ and $\alpha_{Dir} = 10$.

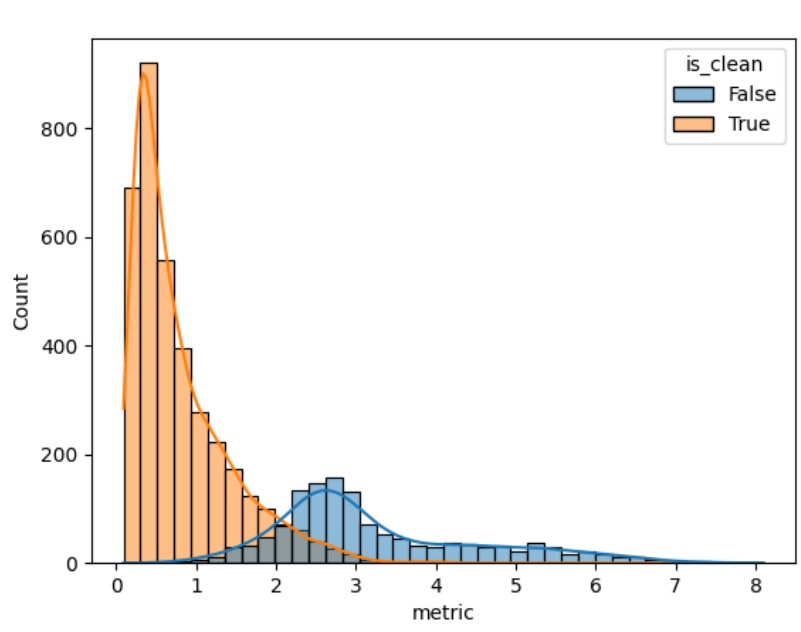

*Figure 22.* Visualization of the FS partition results of FedGR. The F-LNL setup: CIFAR-10, Mixed, $\phi = 1.0$, and $\mathcal{U}(0.2, 0.4)$.