# OpenReview forum: "Learning Locally, Revising Globally: Global Reviser for Federated Learning with Noisy Labels"
_ICML.cc/2026/Conference — ICML 2026 regular_

### Official Review · Reviewer_skjM · 2026-02-23

**Soundness:** 3
**Presentation:** 3
**Significance:** 2
**Originality:** 3
**Overall Recommendation:** 4
**Confidence:** 4

**Summary:**

This paper proposes FedGR, a new federated learning (FL) framework motivated by the intrinsic label-noise robustness of FL, to reduce the overfitting problem in FL training with noise labels. The authors report an underexplored phenomenon that FL models memorize noisy labels more slowly than centralized training models, and they leverage it to design a novel FL training framework with three components, i.e.,  sieving-and-refining, global EMA distillation, and global representation regularization. Extensive experimental results demonstrate that the proposed FedGR method can show substantial gains in both test accuracy and robustness under FL training with different noise levels.

**Compliance With Llm Reviewing Policy:**

Affirmed.

**Final Justification:**

My concern on convergence analysis, communication cost, and parameter analysis have been adequately addressed by the authors during the rebuttal stage. Therefore, I have increased my recommendation score to 4 accordingly.

**Key Questions For Authors:**

1. The mathematical expression in Eq. (15) is quite ambiguous. What is the meaning of $\gamma_g = \gamma_g, t_a \ge \alpha$?

2. What is the communication overhead of uploading instance-level data proxies to the server at every communication round, and how does it compare to the regular FL model upload/download communication overhead? Does FedGR adapt to compression or sampling strategies to reduce this additional communication overhead?

3. Is FedGR sensitive to label-noise ratio $\beta$, warm-up hyper-parameter $\alpha$, and proportion of the reliable labels $\mu$?

4. Does the intrinsic label-noise robustness of FL proposed in this paper still exist across different FL aggregation methods, different client participation rates, different local training epochs, and larger-scale dataset settings?

**Limitations:**

The authors do not discuss the limitations of this paper. Here are some of the suggestions for further improvement.

The intrinsic label-noise robustness phenomenon of FL has not been adequately analyzed in the paper. The authors claim that the proposed FedGR design is motivated by this intrinsic robustness of FL. However, without an in-depth investigation into the root causes of this robustness, the method design in FedGR lacks reasonable design rationale and seems like a combination of existing techniques. This weakens the technical soundness of this work. The authors should conduct a rigorous investigation into the mechanisms underlying label-noise robustness of FL. This can help bridge the gap between the observed phenomenon and the method design in FedGR. In addition, a thorough analysis of the underlying mechanisms of this robustness could help reviewers and researchers better understand the importance and potential application of their proposed FedGR, which further strengthens the technical soundness and contribution of this paper.

**Strengths And Weaknesses:**

### Strengths:
- The method for leveraging the intrinsic robustness of FL in label noise is motivating.
- The system design of FedGR is straightforward and easy to understand.
- The experimental results are comprehensive, comprising multiple datasets, noise types, and distribution settings.

### Weakness:
- **Unvalidated convergence analysis.** The author claims a convergence guarantee but does not present the corresponding analysis in the manuscript. In addition, the manuscript does not provide the underlying assumptions or proof sketch, making it difficult to judge its correctness and significance.

- **Insufficient investigation of FL robustness mechanisms.** While the paper presents a compelling observation regarding the intrinsic label-noise robustness of federated learning, the authors do not provide a deeper theoretical analysis or empirical investigation of this phenomenon. It remains unclear why FL can demonstrate superior robustness compared to centralized learning, and which factor (model architectures, optimizers, client participation rates, aggregation strategies) affects or determines this robustness.

- **Loose connection between the phenomenon and method design.** The authors claim that the proposed FedGR design is motivated by the intrinsic label-noise robustness of FL compared to centralized learning. However, without a rigorous investigation into the root causes of this robustness, the proposed three components in FedGR (Federated Sieving, EMA Distillation, and Global Regularization) lack reasonable design rationale and look like a combination of existing techniques. This weakens the technical soundness of this work.

- **Non-trivial and unreported communication cost.** The proposed FedGR method requires that all clients update instance-level data proxies to the server at every communication round. This could incur massive additional communication costs compared to the vanilla FedAvg, especially when trained on large-scale datasets. Though the authors claim that it introduces moderate computation and communication costs, they fail to provide the empirical or mathematical evidence in the manuscript, which limits the practical applicability of the proposed method.

- **Incomplete parameter analysis.** While the authors have presented hyperparameter analysis on several key parameters in FedGR. There remain three important threshold/parameters unexplored in FedGR: label-noise ratio $\beta$, warm-up hyper-parameter $\alpha$, and the threshold of the proportion of reliable labels $\mu$.

- **Typos and presentation issues.**

    - Missing impact statement.
    - Missing space in Line 98: “FedGRemploys”.
    - Missing footnote 1 in the caption of Figure 2: 'FedGR operates in three modules $^1$....'. However, there is no corresponding footnote in the text.
    - In Eq.(16), the first two notions $D_k^r$ and $D_k^c$ are datasets, but the last term is a set of labels (not (x,y) pairs), which is inconsistent.

---

> ### Author Rebuttal · Authors · 2026-03-31
>
> Thank you for your detailed review. We address your concerns below.
>
> **W1, Unvalidated convergence analysis**: The convergence of the proposed FedGR is theoretically guaranteed, and we present the analysis in Appendix A.7 (please see the supplementary material) due to the limited paper length, as noted in the main submission (Lines 162–163, right column). **In brief, we extend the convergence analysis of vanilla FedAvg to the modified client objective (Eq.4) and prove that FedGR achieves the standard $O(1/\sqrt{T})$ non-convex convergence rate.**
>
> **W2 & L1, Deeper Investigation on intrinsic label-noise robustness of FL**: First, we provided deeper empirical investigations in Appendix A.3 (please see the supplementary material), as mentioned in Fig. 1. In summary, we investigated this phenomenon under different client participation rates, label-noise rates, datasets, data heterogeneity, and client scales. **The results demonstrate that this phenomenon is not an artifact of specific experimental settings.** As for the optimizer and the model architecture, we follow the setup of vanilla FedAvg as they are widely-used in practical. Second, we agree that this phenomenon requires theoretical analysis, which we leave for future work, as mentioned in conclusion.
>
> **W3, Connection between phenomenon and method design**: Based on the investigations in the main text and appendix, we confirm that FL’s intrinsic label-noise robustness is non-trivial and not an artifact of specific experimental settings. And all FedGR modules leverage this property to regularize learning with noisy labels. Specifically, FedGR first uses it to sieve and refine each client’s noisy labels via the sieving-and-refining module. FedGR then regularizes local training through the globally revised EMA distillation module and the global representation regularization module, building on the intrinsic label-noise robustness of the global model. Hence, FedGR is designed around this phenomenon.
>
> **W4 & Q2, Communication cost**: Please refer to the response of W3 for reviewer cQRr.
>
> **W5 & Q3, Parameter analysis**: First, we would like to clarify that **almost all the hyerparameters are fixed for all the experiments in this study**. We only tune a $\lambda\mathcal{R}$ and $\alpha$ based on dataset difficulty. For the mentioned specific parameters, the label-noise ratio is not a tunable hyperparameter, which denotes the difficulty of the label-noise settings. And the warm-up round $\alpha$ is a standard parameter in C-LNL literature (e.g., DivideMix) and is fixed based on dataset difficulty (100 for CIFAR, and 50 for Clothing1M). $\mu$ is the hyperparameter to adjust the $\gamma_g$. For further clarity, we provide an extra hyperparameter analysis on $\alpha$ and $\mu$ (please refer to the response of W2 for reviewer cQRr). In brief, $\alpha$ and $\mu$ only exhibit sensitivity in exceptionally difficult F-LN settings—such as a high label-noise ratio (>0.5) or severe label skew ($\alpha_{dir}$ < 1)—which represent extreme corner cases in real-world scenarios.
>
> **Q1, Explanation of Eq.15**: The Eq.15 denotes the scheduling of the momentum decay $\gamma_g$ for the global revised EMA step. Specifically, 1) during the warmup phase, $\gamma_g$ is set to the zero, and 2) during the later phase, $\gamma_g$ is set zero if the estimated label-noise ratio is large than $\beta$ and the proportion of reliable samples is less $\mu$, otherwise  $\gamma_g$ is set to the user-defined values.
>
> **Q4, Compatibility of different FL setups**: First, we explicitly investigate these factors in our observation experiments detailed in Appendix A.3. As shown in Figures 5-11, such a property remains consistent across different datasets (CIFAR-10 and CIFAR-100), different local epochs (1 and 10), various client sampling ratios (0.2, 0.5, and 1.0) and different client scales (20 and 100). Second, the results on large-scale real-world label-noise dataset Clothing1M (Tables 3 and 16, and Figure 4) demonstrate the effectiveness of the FedGR. Third, while our primary investigation, experiments and theoretical analysis (Appendix A.7) are grounded in FedAvg, we agree that other FL aggregation methods should also be explored. However, we believe such phenomenon extends to similar aggregation strategies, such as FedProx.
>
> **W6, Typos and other minor issues**: Thanks for your suggestions. First, all the typos and missing contents will be added. Second, Eq.16 denotes a Kullback-Leibler divergence loss which is adopts to regularize the representation of clients’ model during the local training and it does not require labels. The $f(a;b)$ means the output of func $f$ with input $x$ and its parameters $b$.

---

> > ### Author Rebuttal · Reviewer_skjM · 2026-04-01
> >
> > Most of my questions have been adequately addressed by the authors. I have increased my recommendation score accordingly.

---

> > > ### Author Response · Authors · 2026-04-02
> > >
> > > Thank you for your continued engagement and we are glad to find that our responses have addressed your concerns. Thanks for your valueable comments!

---

### Official Review · Reviewer_cQRr · 2026-03-04

**Soundness:** 2
**Presentation:** 2
**Significance:** 2
**Originality:** 2
**Overall Recommendation:** 4
**Confidence:** 5

**Summary:**

This paper tackles the heterogeneous FL with noisy label (F-LN) problem, which is exacerbated by heterogeneity in label-noise types, ratios, and data distribution across different clients in FL. The authors find that the FL global model of FL exhibits a slow memorization of noisy labels. I think this is intuitive, since the global model is aggregated, so its sensitivity is relatively low. Then, this paper proposes the Federated Global Reviser (FedGR) framework, including 1) a sieving-and-refining module to partition clean and noisy samples for each client; 2) a globally revised exponential moving average distillation module to enforce consistency between the global representation of weakly augmented images and the local representation of strongly augmented images; and 3) a global representation regularization module to regularize the local training. Experiments on three datasets demonstrate that FedGR outperforms seven state-of-the-art baselines under various F-LN settings.

**Compliance With Llm Reviewing Policy:**

Affirmed.

**Final Justification:**

The author's final response addressed my main concern: they further explained why, in the paper's setting, the "common" direction naturally aligns with the "clean" direction. Even with some shortcomings in novelty and limitations in the experimental setup, I still decided to raise my score.

**Key Questions For Authors:**

Please see the five questions in Weaknesses.

**Limitations:**

No, the authors have not adequately discussed the limitations and potential negative societal impacts of their work in the provided main context. The only explicitly mentioned limitation is the deferral of the theoretical analysis of the memorization phenomenon to future work.

**Strengths And Weaknesses:**

Strengths:
1. The authors find that the FL global model of FL exhibits a slow memorization of noisy labels. I think this is intuitive, since the global model is aggregated, so its sensitivity is relatively low.
2. First, moving the samples sieving to the server mitigates the impact of local heterogeneity, and then the EMA distillation and global representation regularization module balance the globally robust knowledge with local distribution.
3. Experiments on three datasets demonstrate that FedGR outperforms seven state-of-the-art baselines under various F-LN settings.

However, I have some questions and concerns:
1. The proposed framework is predicated on the slower memorization effect of the global model, however, Figure 1 essentially only demonstrates that, under noise label settings, FL learns more slowly than centralized training and is less prone to overfitting to noise. So, the global model in Figure 1 "remembers noise more slowly" likely does so simply because it learns from all the data more slowly, rather than because it has some kind of "noise-resistant ability". If the global model simply acts as a delayed learner, relying heavily on it for server-side sieving and label refining might merely be exploiting an underfitting effect, which could collapse under different configurations (e.g., more aggressive learning rates) or prolonged training epochs. And also, I suggest the authors provide a foundational intuition or preliminary proof explaining why model aggregation inherently resists noise memorization.
2. The proposed framework introduces multiple moving parts and hyperparameters, including $\lambda_{\mathcal{B}}$, $\lambda_{\mathcal{R}}$, noise threshold $\beta$, EMA momentum weight $\gamma_{g}$, $\gamma_{l}$. How tune these balancing weights efficiently for different datasets?
3. What is the exact quantitative communication overhead introduced by uploading the instance-level data proxies?
4. For the globally revised EMA, what happens if the global model is temporarily poisoned by a cluster of highly noisy clients? Does the momentum decay mechanism sufficiently isolate the local EMA model from this?
5. Some small typo: in line 071: "In othe words" should be "In other words"; the author's appendix does not appear to be in the same PDF as the main context.

---

> ### Author Rebuttal · Authors · 2026-03-31
>
> Thank you for your patient review. We address your concerns below.
>
> **W1, Delayed learner vs. Noise-resistant ability**: First, comparing to the centralized model, the global model is not merely a delayed learner; it actively suppresses noise memorization. **As shown in Fig. 1(b), centralized learning (blue) and FedAvg (green and orange) initially improve at a similar rate on the label-noise dataset, but FedAvg does not degrade the way centralized learning does. Therefore, the global model is not simply exploiting an underfitting effect.** In addition, Figures 5–11 (Appendix) show that across different datasets and federated learning setups, the global model maintains its test accuracy and does not degrade quickly, unlike centralized models, which easily overfit. Second, comparing to the local model of each client, the global model is a delayed learner. But such a delayed learner enjoys a longer memory effect than that in centralized learning and this is meaningful for F-LNL. Third, we provide an intuitive explanation for this phenomenon in Appendix A.3. Clean data patterns are shared across clients, so their gradient updates align and reinforce one another during aggregation. Noisy labels, however, produce highly idiosyncratic, client-specific gradients that cancel out in expectation when averaged. We did not include this explanation in the main text because it is not sufficiently objective.
>
> **W2, Hyperparameter tuning**: We have provided the full hyperparameter setup in Table 17 (see the Appendix A.5 in supplementary material). **Most hyperparameters are shared across datasets, and we recommend using these defaults for unexplored datasets which could be a strong start point.** Second, we conduct experimental analysis of several core hyperparameters, including $\lambda_\mathcal{B}$, $\lambda_\mathcal{R}$, $\gamma_g$, and $\gamma_l$, in Table 6–7. According to these results, we provide a setup guide in our main text. Furthermore, we also provide extra hyperparameter analysis on warmup round $\alpha$ and reliable dataset size $\mu$ on CIFAR-10 under various FL setups.
> | Data Partition | I.I.D | I.I.D | I.I.D | Non.I.I.D-Dirichlet (0.3) | Non.I.I.D-Dirichlet (0.3) | Non.I.I.D-Dirichlet (0.3) |
> |---|---|---|---|---|---|---|
> | Noise Type | Sym | Asym | Mixed | Sym | Asym | Mixed |
> | $\Phi$ | 1.0 | 1.0 | 1.0 | 1.0 | 1.0 | 1.0 |
> | $\mathcal{U}(\rho_{min},\rho_{max})$ | 0.5-1.0 | 0.2-0.4 | 0.2-0.4 | 0.5-1.0 | 0.2-0.4 | 0.2-0.4 |
> | $\mu=0.2$ | 81.94 $\pm$ 1.46 | 91.07 $\pm$ 0.43 | 91.97 $\pm$ 0.21 | 60.35 $\pm$ 5.83 | 83.77 $\pm$ 2.84 | 83.53 $\pm$ 2.46 |
> | $\mu=0.5$ | 83.91 $\pm$ 1.32 | 91.64 $\pm$ 0.38 | 92.27 $\pm$ 0.18 | 63.64 $\pm$ 5.39 | 83.67 $\pm$ 5.02 | 84.65 $\pm$ 2.38 |
> | $\mu=0.8$ | 82.43 $\pm$ 1.07 | 91.13 $\pm$ 0.35 | 91.84 $\pm$ 0.19 | 61.20 $\pm$ 4.45 | 83.46 $\pm$ 2.58 | 84.15 $\pm$ 2.08 |
> | $\alpha=50$ | 82.52 $\pm$ 0.77 | 91.66 $\pm$ 0.30 | 92.18 $\pm$ 0.17 | 55.81 $\pm$ 2.39 | 83.80 $\pm$ 2.47 | 84.13 $\pm$ 2.47 |
> | $\alpha=100$ | 83.91 $\pm$ 1.32 | 91.64 $\pm$ 0.38 | 92.27 $\pm$ 0.18 | 63.64 $\pm$ 5.39 | 83.67 $\pm$ 5.02 | 84.65 $\pm$ 2.38 |
> | $\alpha=200$ | 76.02 $\pm$ 2.27 | 91.50 $\pm$ 0.19 | 91.76 $\pm$ 0.22 | 60.05 $\pm$ 1.38 | 81.44 $\pm$ 2.96 | 80.97 $\pm$ 2.43 |
>
> **W3, Communication overhead**: First, as noted in the 2-nd footnote in page 2, we provide an analysis of computation and communication overhead in Table 18 (see Appendix A.8 in supplementary material). Compared with vanilla FedAvg, the additional communication overhead per round consists of two scalars: the sample-wise loss value and its corresponding identifier. The total communication overhead depends on the dataset scale. The additional computation overhead includes three extra forward passes during local training, as well as the server-side GMM. Second, compression is not necessary to reduce the communication overhead introduced by these two small scalars.
>
> **W4, Poisoned global model**: First, the global model can be easily poisoned by a cluster of highly noisy clients only during the warm-up phase, since it uses vanilla supervised learning on noisy labels. In later phases, poisoning becomes much harder because noisy samples are filtered and rectified by the Sieving-and-Refining module. To be specifically, as Fig.3(b) shown, the high sample selection quality of FedGR remedy such potential risk. Second, the reported results (Table 1–3) show that under extreme label-noise ratios (0.5–1.0), the proposed FedGR still outperforms other baselines. Therefore, **poisoning the global model is therefore difficult without strong assumptions**. Third, under the local EMA update rule (Eq.12), $\gamma_g$ is sufficiently large, ensuring the local EMA teacher does not instantly collapse to a poisoned global state (as shown in Tabel.7).
>
> **W5, Typos and Appendix**: Thank you for catching the typos and the organization of appendix; we will fix it. Furthermore, we will add the more limitation discussion about hyperparameter and so on.

---

> > ### Author Rebuttal · Reviewer_cQRr · 2026-04-01
> >
> > Thank you for this rebuttal.
> >
> > My main concern is still the paper's central premise: the global model exhibits an intrinsic noise-resistant property. The rebuttal suggests that FL does not degrade in the same way as centralized training, and gives an explanation based on gradient alignment across clients. But the rebuttal still does not establish that aggregation (global model) itself inherently resists noise memorization under controlled comparisons. The gradient alignment explanation clarified two points: 1) The clean-label gradients are more likely to be consistent across clients, so aggregation reinforces them. 2) The noisy-label gradients are more idiosyncratic across clients, so aggregation tends to average them out. However, **aggregation does not know which directions are "clean" and which are "noisy", it only reinforces directions that are common across clients and attenuates directions that disagree**. So the noisy gradients need not cancel under aggregation, and may instead align if the corruption is correlated across clients. Therefore, I still feel that the current evidence supports the statement that FL is slower to memorize noise in the settings, but not yet the claim that the global model has an inherent noise-resistant ability.
> >
> > In addition, the response to the communication overhead is also still incomplete to me: FedGR uploads per-instance proxies, i.e., $(d\_{i,k}, \bar{\ell}\_{i}^{t})$, so the extra communication scales with the number of local samples. But the rebuttal does not quantify the exact overhead (e.g., practice transmission volume) relative to FedAvg.
> >
> > Overall, I appreciate the authors' clarifications. But due to concerns about the paper's core claim, I have decided to maintain my score.

---

> > > ### Author Response · Authors · 2026-04-02
> > >
> > > Thank you for your continued engagement and comment.
> > >
> > > Our clarification and the review comment are discussing the same thing: the aggregation of FedAvg "reinforces directions that are common... and attenuates directions that disagree”. We agree that "aggregation does not know which directions are 'clean' and which are 'noisy'". However, we would like to clarify why the "common" directions naturally align with the "clean" ones in our setting. First, at the local level, **updates are dominated by clean gradients because clean data constitutes the majority of the dataset (i.e., the noise transition matrix is diagonally dominant).** This is empirically supported by the rise in the test accuracy curve, as the model would otherwise be unable to achieve such high test accuracy (>60%). Second, at the global level, the F-LN)problem is characterized by dual-heterogeneity: clients possess diverse non-IID data distributions alongside varying noise types and ratios. Because clients do not share the same underlying noise patterns (i.e., noise transition matrix), their noisy gradients point in diverse, conflicting directions. **Consequently, the gradients with respect to noisy samples fail to form a cohesive, joint force capable of biasing the global model.** Instead, the aggregation process is naturally dominated by the common, clean model updates, allowing the global model to optimize along the clean direction even in the presence of noisy labels.
> > >
> > > As for the concern about the communication overhead, we provide a communication volume comparison here. For FedGR, the total extra payload per sample are 8 bytes:
> > > - $d_{i,k}$ (Sample ID): 1 `int32` = 4 bytes.
> > > - $\bar{\ell}_{i}^{t}$ (Loss value): 1 `float32` = 4 bytes.
> > >
> > > For the I.I.D settings of Table 1, the model payloads of FedAvg and FedGR could be $\approx 44.68$ MB ($\approx 11.17$ million float32 parameters) and the communication volume comparison could be the following:
> > > | Method | Model payload volume per round | Extra payload volume per round | Relative overhead |
> > > |---|---|---|---|
> > > | FedAvg | $\approx 44.68$ MB | 0 | 0 |
> > > | FedGR | $\approx 44.68$ MB | **500*8=4000 B** | **0.0087%** |
> > >
> > > For the I.I.D settings of Table 3, the model payloads of FedAvg and FedGR could be $\approx 94.0$ MB ($\approx 23.5$ million float32 parameters) and the communication volume comparison could be the following:
> > > | Method | Model payload volume per round | Extra payload volume per round | Relative overhead |
> > > |---|---|---|---|
> > > | FedAvg | $\approx 94.0$ MB | 0 | 0 |
> > > | FedGR | $\approx 94.0$ MB | **2000*8=16000 B** | **0.016%** |
> > >
> > > We hope these clarifications address your concerns. We would be deeply grateful if you might consider raising your score.

---

### Official Review · Reviewer_Uopp · 2026-03-11

**Soundness:** 3
**Presentation:** 4
**Significance:** 3
**Originality:** 3
**Overall Recommendation:** 4
**Confidence:** 4

**Summary:**

This submission studies federated learning under heterogeneous label noise (F-LN) and provides an empirical observation that the global model in FL exhibits slower memorization of noisy labels than centralized training, maintaining more reliable predictions during training. Building on this insight, the authors propose FedGR, a global-model-centric framework with three components: 1) a server-side federated sieving (FS) plus client-side label refining (LR) that uses global-model-inferred proxies to partition and correct noisy labels; 2) a globally revised EMA distillation that initializes a client EMA teacher by blending with the global model and distills to the local model; and 3) a global representation regularization aligning global and local representations under weak–strong augmentations. Extensive experiments on CIFAR-10/100 and Clothing1M with varied noise and distribution heterogeneity demonstrate strong gains over seven baselines. Furthermore, this submission provides theoretical convergence analysis on the proposed FedGR.

**Compliance With Llm Reviewing Policy:**

Affirmed.

**Final Justification:**

I have carefully read the authors' rebuttal. Given that my original assessment was favorable, I keep my initial rating unchanged.

**Key Questions For Authors:**

1. How robust is FS to adversarial/malicious clients manipulating uploaded proxies (e.g., inflating/deflating losses)? Have you considered robust aggregation or outlier filtering at the server?

2. What is the computational overhead of server-side GMM fitting as the number of clients and samples grows?

**Limitations:**

The observed intrinsic label-noise robustness of the global model in FL is an intriguing empirical finding. However, a more formal analysis could help the community understand why server aggregation offer noise robustness.

**Strengths And Weaknesses:**

Strengths
- Exploits an interesting phenomenon: the global model in FL overfits noisy labels more slowly than centralized training, and leverages it systematically throughout the pipeline.
- Addresses an important problem—federated learning with heterogeneous label noise—where many centralized LNL assumptions (class balance, homogeneous noise) break down.
- Strong experimental results: Experiments on CIFAR-10/100 and a real-world noisy dataset (Clothing1M), across IID and Non-IID (Dirichlet) partitions, multiple noise types (symmetric, asymmetric, mixed), and varying noisy-client proportions demonstrate the superior performance of the proposed FedGR.
- Offering a theoretical analysis on the convergence of the proposed FedGR
- This submission is well-organized with clear presentation.

Weaknesses
- The “intrinsic label-noise robustness of FL” claim seems to be strong. The observation is interesting, but it is not fully clear whether this is an intrinsic property of FL or a consequence of specific choices like local epochs, sampling ratio, aggregation, and optimization budget.
- Real-world evidence is limited. Though Clothing1M is a large-scale real-world noisy-label dataset here, the appendix itself notes that Clothing1M is “less faithful” to realistic F-LN.

---

> ### Author Rebuttal · Authors · 2026-03-31
>
> We sincerely thank you for your positive evaluation and for recognizing the practical relevance, strong empirical performance, and coherent design of our work We address your concerns below.
>
> **W1, Intrinsic property vs. consequence of specific choices:** We argue it is an intrinsic property of the FedAvg rather than a FL setup. According to the discussion in Appendix A.3, FedAvg acts as a weight-space ensemble. Gradients driven by shared clean data are coherent across clients and are systematically reinforced during server aggregation. Conversely, gradients induced by idiosyncratic noisy labels are highly client-specific and tend to cancel out in expectation. Furthermore, Figures 5-11 in the Appendix demonstrate that this slower memorization holds consistently across different local epochs and sampling ratios, confirming it is not merely an artifact of a FL configuration.
>
> **W2, Real-world Empirical Evaluation:** We note Clothing1M is "less faithful" to F-LN simply because its noise is naturally centralized and feature-dependent, whereas F-LN specifically studies client-level heterogeneity. However, to the best of our knowledge, there is no benchmark that meet the native dual heterogeneity. We will add this to our limitations and future work discussion.
>
> **Q1, Robustness to malicious clients:** The proposed FedGR assumes "honest-but-noisy" clients, which is the standard assumption in the F-LNL literature. If malicious clients intentionally inflate/deflate losses, the server-side GMM could be skewed. However, FedGR's server aggregation is orthogonal to Byzantine-robust techniques. Integrating robust aggregation rules (e.g., Krum, Median) or outlier filtering before GMM fitting is a promising extension to secure FS against adversarial attacks. We will add this to our limitations and future work discussion.
>
> **Q2,Computational overhead of server-side GMM**: The server fits a 1D GMM on scalar loss values. For a dataset of $D$ samples (e.g., 50,000 for CIFAR-10), fitting a 2-component 1D GMM takes a fraction of a second on a standard CPU. The time complexity is $O(D \cdot iter)$, which scales linearly and is orders of magnitude faster than the neural network parameter aggregation step.

---

> > ### Author Rebuttal · Reviewer_Uopp · 2026-04-05
> >
> > I have carefully read the authors' rebuttal. Given that my original assessment was favorable, I keep my initial rating unchanged.

---

> > > ### Author Response · Authors · 2026-04-06
> > >
> > > Thanks for your patient and valuable review comments!

---

### Official Review · Reviewer_mycA · 2026-03-11

**Soundness:** 3
**Presentation:** 3
**Significance:** 3
**Originality:** 3
**Overall Recommendation:** 5
**Confidence:** 4

**Summary:**

This paper proposes FedGR to address the federated learning with noisy labels under both label-noise heterogeneity and data heterogeneity. The main motivation is an observation that the global FL model appears to memorize noisy labels more slowly than centralized training and than local client models. The proposed FedGR, which combines with following components: Federated Sieving (FS): the server fits a GMM over per-example running loss proxies uploaded by clients to separate likely clean/noisy examples; Label Refining (LR): clients refine noisy labels using the global model and confidence-weighted pseudo-labeling; Globally Revised EMA Distillation + representation regularization: local EMA teachers are revised toward the global model, and feature consistency to the global model is enforced. The paper reports experiment results on CIFAR-10, CIFAR-100, and Clothing1M with iid and Dirichlet non-iid partitions, multiple noise types, and varying noise prevalence across clients. Results show consistent improvements over several federated noisy-label baselines, especially in harder settings.

**Compliance With Llm Reviewing Policy:**

Affirmed.

**Final Justification:**

The rebuttal resolved my main concerns.

**Key Questions For Authors:**

Please refer to weaknesses.

**Limitations:**

- The authors mention the communication and computational costs in the appendix. However, the reliance on a suite of hyperparameters could be discussed more explicitly as a practical limitation.
- The computational overhead on the client side is very high. As shown in Table 18, FedGR requires (E+3) forward passes and E backward passes per round, plus maintaining an EMA model. For resource-constrained edge devices, this is a bottleneck.

**Strengths And Weaknesses:**

Strengths
- Important and practical problem: Noisy labels in federated learning are practically relevant, and the joint treatment of label-noise heterogeneity and non-iid data is meaningful.
- The evaluation spans multiple datasets and multiple noise setups, including mixed noise types and non-iid partitions. And the experimental performance are strong, especially in the more difficult high-noise and non-iid settings.
- The main intuition—the global model can be a better denoising signal than local models—is sensible and practically appealing, which provides a reusable insight for the community.
- The proposed method is conceptually coherent. The overall design—use the global model for denoising and local regularization—is easy to follow and reasonably well aligned with the motivating observation.
- The presentation is good and easy to follow.

Weaknesses
- While the authors state this preserves privacy (and discuss it in Appendix A.6), could the authors comment on whether tracking the trajectory of individual loss values over time might expose clients to advanced membership inference attacks?
- In Eq. 11, the refinement strategy relies heavily on the estimated noise ratio. If the GMM in the Federated Sieving step poorly estimates it (e.g., due to extreme class imbalance in the Non-IID setting), how does this error propagate through the EMA and regularization modules?
- Missing of the most recent related works, e.g., FedClean and so on.

---

> ### Author Rebuttal · Authors · 2026-03-29
>
> We sincerely thank you for your positive evaluation and for recognizing the practical relevance, strong empirical performance, and coherent design of our work. We address your concerns below.
>
> **W1, Privacy and advanced membership inference attacks (MIAs)**: The proposed FedGR remains highly resistant to MIAs, even tracking individual loss trajectories. Specifically, the data proxies uploaded are 1D scalar cross-entropy loss values attached to opaque identifiers ($d_{i,k}$). Because the cross-entropy function is many-to-one, it is mathematically impossible to invert a scalar loss to reconstruct the high-dimensional input image or its exact label. Hence, it is hard to perform membership inference attacks on FedGR. We will add this discussion on MIAs to Appendix A.6.
>
> **W2, Error propagation of poor GMM estimation**: If the GMM poorly estimates the noise ratio (e.g., due to extreme Non-I.I.D.), the errors will be mitigated by our downstream modules. Specifically, the Global Representation Regularization ($\mathcal{R}_k$) enforces feature-level consistency with the global model, which operates independently of the GMM's hard label assignments, thus buffering against error propagation. Furthermore, the Globally Revised EMA Distillation also acts as safety net. By revising the local EMA teacher with the robust global model (Eq. 12), we prevent the local model from fully absorbing the GMM's partitioning errors.
>
> **W3, Missing related works**: Thank you for pointing this out. We will include a discussion of FedClean. FedClean is local-first, whose sample selection quality strongly depends on the strength of the local CNLL method. It trusts that each client can run some local noisy-label learner well enough to produce useful inferred labels. While FedGR is global-first, which adopts the global model to denoise the client. That makes FedGR feel more tailored to federated heterogeneity, especially when local client datasets are small or skewed.
>
> **L1, Hyperparameter**: Thanks for your suggestion. FedGR uses several hyperparameters, including $\lambda_\mathcal{B}$, $\lambda_\mathcal{R}$, $\gamma_g$, $\gamma_l$, and etc. To facilitate follow-up research, we provide a hyperparameter analysis (Table 6–7) to show their impact and summarize a setup guide. The remaining hyperparameters follow prior F-LNL and C-LNL literature and are shared across datasets. For an unexplored dataset, some tuning is still required, but our recommended settings provide a strong starting point.
>
> **L2, Computation overhead for local trianing**: We agree that FedGR introduces client-side overhead compared to vanilla FedAvg. However, as shown in Table 18 (Appendix), compared to typical F-LNL baselines like FL-DivideMix (which requires $4m+2$ forward passes and 2 GMMs per client), FedGR is significantly more computationally efficient.

---

> > ### Author Rebuttal · Reviewer_mycA · 2026-04-03
> >
> > Thank you for the detailed response. Overall, the rebuttal addresses my concerns.

---

> > > ### Author Response · Authors · 2026-04-05
> > >
> > > Thanks for your valueable review comments!

---

### Decision · Program_Chairs · 2026-04-30

**Decision:**

Accept (regular)

**Comment:**

This paper tackles a practical problem of federated learning with noisy labels under dual heterogeneity, i.e., label-noise&data heterogeneity, by leveraging the insightful observation that FL global models memorize noise more slowly than centralized models. The reviewers universally commend the submission for its reusable observation, coherent algorithmic framework, significant empirical evaluations, and the significant practical importance to the FL community. During the rebuttal, the authors successfully resolved the primary concerns by providing a convincing gradient-alignment explanation for the intrinsic noise robustness, clarifying the minimal communication overhead, and supplying a rigorous convergence analysis. Given the originality of the core observation, the solid technical evaluations, and the thoroughness of the authors' responses, I recommend for acceptance.